# New insights into the use of stable water isotopes at the northern Antarctic Peninsula as a tool for regional climate studies

Francisco Fernandoy[1], Dieter Tetzner[2], Hanno Meyer[3], Guisella Gacitúa[4], Kirstin Hoffmann[3], Ulrike Falk[5], Fabrice Lambert[6], Shelley MacDonell[7]

[1]Facultad de Ingenieria, Universidad Andres Bello, Viña del Mar, 2531015, Chile
[2]Center for Climate and Resilience Research, Universidad de Chile, Santiago, 8370361, Chile
[3]Alfred Wegener Institute Helmholtz Centre for Polar and Marine Research, Research Unit Potsdam, Telegrafenberg A43, 14473 Potsdam, Germany.
[4]Programa GAIA-Antártica, Universidad de Magallanes, Punta Arenas, 6210427, Chile
[5]Climate Lab, Geography Department, University Bremen, 28334 Bremen, Germany
[6]Department of Physical Geography, Pontificia Universidad Católica de Chile
[7]Centro de Estudios Avanzados en Zonas Áridas (CEAZA), La Serena, Chile

*Correspondence to*: Francisco Fernandoy (francisco.fernandoy@unab.cl)

**Abstract.** Due to recent atmospheric and oceanic warming, the Antarctic Peninsula is one of the most challenging regions of Antarctica to understand both local– and regional–scale climate signals. Steep topography and a lack of long–term and in situ meteorological observations complicate the extrapolation of existing climate models to the sub–regional scale. Therefore, new techniques must be developed to better understand processes operating in the region. Isotope signals are traditionally related mainly to atmospheric conditions, but a detailed analysis of individual components can give new insight into oceanic and atmospheric processes. This paper aims to use new isotopic records collected from snow and firn cores in conjunction with existing meteorological and oceanic datasets to determine changes at the climatic scale in the northern extent of the Antarctic Peninsula. In particular, a discernible effect of sea ice cover on local temperatures and the expression of climatic modes, especially the Southern Annular Mode (SAM), is demonstrated. In years with a large sea ice extension in winter (negative SAM anomaly), an inversion layer in the lower troposphere develops at the coastal zone. Therefore, an isotope–temperature relationship (δ–T) valid for all periods cannot be obtained, and instead, the δ–T depends on the seasonal variability of oceanic conditions. Comparatively, transitional seasons (autumn and spring) have a consistent isotope–temperature gradient of +0.69‰ °C$^{-1}$. As shown by firn core analysis, the near–surface temperature in the northern-most portion of the Antarctic Peninsula shows a decreasing trend (-0.33°C y$^{-1}$) between 2008 and 2014. In addition, the deuterium excess ($d_{excess}$) is demonstrated to be a reliable indicator of seasonal oceanic conditions, and therefore suitable to improve a firn age model based on seasonal $d_{excess}$ variability. The annual accumulation rate in this region is highly variable, ranging between 1060 kg m$^{-2}$ y$^{-1}$ and 2470 kg m$^{-2}$ y$^{-1}$ from 2008 to 2014. The combination of isotopic and meteorological data in areas where data exist is key to reconstruct climatic conditions with a high temporal resolution in Polar Regions where no direct observations exist.

# 1 Introduction

West Antarctica, especially the Antarctic Peninsula (AP), has received increasing attention from the scientific community due to the notable effects of recent warming on the atmosphere, cryosphere, biosphere and ocean. The increase of air temperatures along the West Antarctic Peninsula coast (Carrasco, 2013) displays signs of a shifting climate system since the early 20th century (Thomas et al., 2009). Recently, rapid warming of both atmosphere and ocean has caused ice shelf instability in West Antarctica, especially in some regions of the AP (Pritchard et al., 2012). Instability leading to ice shelf collapse has triggered accelerated ice–mass flow and discharge from land–based glaciers into the ocean, as the ice shelves' buttressing function is lost. Accelerated rates of ice mass loss (Pritchard and Vaughan, 2007; Rignot et al., 2005; Pritchard et al., 2012), in combination with increased surface snow melt, has contributed to a negative surface mass balance especially in the northern part of the AP region (Harig and Simons, 2015; Seehaus et al., 2015; Dutrieux et al., 2014; Shepherd et al., 2012).

The glaciers of the AP have lost ice mass at a rate of approximately 27 ($\pm$2) Gt $y^{-1}$ between 2002 and 2014. This mass loss combined with the mass loss over the West Antarctic ice sheet (121($\pm$8) Gt $y^{-1}$), surpassed the mean positive mass balance of +62 ($\pm$4) Gt $y^{-1}$ observed in East Antarctica, of which most of the positive balance relates to the Dronning Maud Laud region whereas the mass balance of the rest of the EAIS is at equilibrium (Harig and Simons, 2015). This demonstrates how vulnerable the coastal region of West Antarctica is to increased air and sea surface temperatures (Bromwich et al., 2013; Meredith and King, 2005).

Surface snow and ice melt on the AP represents up to 20% of the total surface melt area (extent) and 66% of the melt volume from Antarctica for at least the last three decades (Trusel et al., 2012; Kuipers Munneke et al., 2012). Regional positive temperatures detected by remote–sensing techniques and ice–core data reveal that melt events have been temporally more wide–spread since the mid–20th century (Abram et al., 2013; Trusel et al., 2015), with some severe melt events during the first decade of the 21st century (Trusel et al., 2012). Increased surface melt and glacier calving are likely to have freshened upper ocean layers and therefore impacted biological activity in the coastal zone (Meredith et al., 2016; Dierssen et al., 2002). The most significant warming trend detected at the AP coast occurred during the winter season, especially on the western side of the Peninsula, where a positive trend of >0.5°C decade$^{-1}$ for the period 1960–2000 has been reported at several stations (Turner et al., 2005; Carrasco, 2013). For example, winter warming is especially evident in daily minimum and monthly mean temperature increases, as described by Falk and Sala (2015) for the meteorological record of the Bellingshausen Station at King George Island (KGI) at the northern AP during the last 40 years. In KGI the daily mean temperature during winter increased at about 0.4°C decade$^{-1}$, with a marked warming during August (austral winter) at a rate of +1.37($\pm$0.3)°C decade$^{-1}$. Positive temperatures even in winter are more commonly observed, leading to more frequent and extensive surface melting year–round especially for the northern AP, which is dominated by maritime climate conditions (Falk and Sala, 2015).

The mechanisms causing increasing atmosphere and ocean temperatures are still not completely understood but can be linked to perturbations of regular (pre–industrial period) atmospheric circulation patterns (Pritchard et al., 2012; Dutrieux et al., 2014). Most heat advection to the southern ocean and atmosphere has been related to the poleward movement of the Southern Annular Mode (SAM) and to some extent to the El Niño Southern Oscillation (ENSO) (Gille, 2008; Dutrieux et al., 2014; Fyfe et al., 2007). During the last decades, SAM has been shifting into a positive phase, implying lower than normal (atmospheric) pressures at coastal Antarctic regions (latitude 65°S) and higher (atmospheric) pressures over the mid–latitudes (latitude 40°S) (Marshall, 2003). As a result of lower pressures around Antarctica, the circumpolar westerly winds increase in intensity (Marshall et al., 2006). As a consequence, air masses transported by intensified westerlies overcome the topography of the AP more frequently, especially in summer, bringing warmer air to the east side of the AP (van Lipzig et al., 2008; Orr et al., 2008). The correlation between the SAM and surface air temperature is generally positive for the AP, explaining a large part (~50%) of near–surface temperature increase for the last half century (Marshall et al., 2006; Marshall, 2007; Carrasco, 2013; Thompson and Solomon, 2002). An enhanced circulation enables more humidity to be transported to and trapped at the west coast of the AP due to the orographic barrier of the central mountain chain. This has resulted in the consistent increase of accumulation across the entire AP during the 20th century, thereby doubling the accumulation rate from the 19th century in the southern AP region (Thomas et al., 2008; Goodwin et al., 2015; Dalla Rosa, 2013).

The increase of greenhouse gas concentrations and the stratospheric depletion of the ozone layer, both linked to anthropogenic activity, are thought to be the main forcing factors of the climate shift that has affected the ocean–atmosphere–cryosphere system for at least the last half century (Fyfe and Saenko, 2005; Sigmond et al., 2011; Fyfe et al., 2007).

The lack of long–term meteorological records limits accurate determination of the onset and regional extent of this climate shift. Therefore, climate models are needed to extend the scarce climate data both spatially and temporally. One major challenge is to correctly integrate the steep and rough topography of the AP into climate models. To facilitate this, more detailed information of surface temperatures, melting events, accumulation rates, humidity sources and transport pathways are urgently needed. As direct measurements of these parameters are often not available, the reconstruction of the environmental variability, basically relies on proxy data such as the stable water isotope composition of precipitation, firn and ice (e.g.: Thomas and Bracegirdle, 2009; Thomas et al., 2009; Abram et al., 2013).

In this study, we focus on a stable water isotope–based, high temporal resolution assessment (seasonal resolution between austral autumn 2008 and austral summer 2015) of climate variables including accumulation rates, temperatures and melt events on the AP and their relationship with atmospheric and oceanic conditions i.e. sea surface temperature, humidity and sea ice extent. We investigate the effects of the orographic barrier of the AP on air mass and moisture transport, with increasing precipitation rates from the coast to the mountain range on the Peninsula divide at ca. 1100 m a.s.l. (Fernandoy et al., 2012), where the ice thickness reaches ca. 350 m at maximum (Cárdenas et al., 2014).

## 2 Glaciological setting and previous work

Since 2008 we have undertaken several field campaigns to the northernmost region of the AP where we have retrieved a number of firn cores of up to 20 m depth. The present investigation is the first of its kind for this sector of the AP. Other studies have been carried out further south at Detroit Plateau (Dalla Rosa, 2013) and Bruce Plateau (Goodwin et al., 2015),

approximately 100 and 400 km southwest of the northern AP. Nonetheless, not much is known about the glaciological conditions at the northern tip of the AP and very few ice cores have been retrieved from this area despite the high number of scientific stations in the region (Aristarain et al., 2004; Simões et al., 2004; Goodwin et al., 2015; Fernandoy et al., 2012; Dalla Rosa, 2013). The AP and Sub-Antarctic islands are principally characterized by mountain glaciers or small ice caps, which flow into the Bellingshausen and Weddell Sea to the west and east, respectively (Turner et al., 2009). Rückamp et al.

(2010) noted that the ice cap covering King George Island, South Shetlands (62.6°S, 60.9°W) is characterized by polythermal conditions and temperate ice at the surface (>-0.5°C), and is therefore sensitive to small changes in climatic conditions. Further south, Zagorodnov et al. (2012) showed that temperatures from boreholes reach a minimum at 173 m depth (-15.8°C) at Bruce Plateau (66.1°S, 64.1°W, 1975.5 m a.s.l.). Similar glaciological conditions were reported on the east side of the AP at James Ross Island (64.2°S, 57.8°W, 1640 m a.s.l.) (Aristarain et al., 2004). Accumulation rates at the

northern AP are directly related to the westerly atmospheric circulation and maritime conditions, with values close to 2000 kg m$^{-2}$ y$^{-1}$ on the west side (Goodwin et al., 2015; Potocki et al., 2016) and lower values (~400 kg m$^{-2}$ y$^{-1}$) on the eastern side (Aristarain et al., 2004); ice thickness from all coring–sites reported is <500 m to the bedrock.

## 3 Methodology

### 3.1 Field work and sample processing

During five austral summer campaigns (2008–2010, 2014, 2015), an altitudinal profile was completed from sea level near O'Higgins Station (OH) to 1130 m a.s.l at the Laclavere Plateau (LCL) (Fig. 1). In total, five firn cores are included in this paper: OH-4, OH-5, OH-6, OH-9, OH-10 (Fig. 1); coordinates and further details of the firn cores are given in Table 1. Two hundred and eight daily precipitation samples were gathered at the meteorological observation site of the O'Higgins Station (57.90°W, 63.32°S, 13 m a.s.l.) during 2008–2009 (Fernandoy et al., 2012) and 2014 (Table 2). The overwintering crew at

O'Higgins Station collected daily precipitation samples from pluviometers installed at the meteorological observation site. Each daily sample comprised of a filling a narrow neck HDPE type bottle with a 30 ml composite sample of the precipitation (both liquid and solid) that fell in the previous 24 hours. The bottles were tightly closed and stored frozen year–long to ensure correct storage and to facilitate the subsequent transport to the laboratory at the end of each year. From these samples, approximately 6% (13 samples) were discarded from the analysis due to improper storage causing leakage from the bottles.

Improper storage was assessed using a statistical outlier test (modified Thompson tau technique) that indicated unusual values of stable water isotope analyses.

Cores from the O'Higgins Station site (OH-4, OH-5, OH-6 and OH-9) were retrieved between 2008 and 2010 and analyzed for their stable water isotope composition and physical properties as described by Fernandoy et al. (2012) and Meyer et al. (2000). Additionally, a density profile of OH-9 was obtained using an X–ray microfocus computer tomograph at the ice–core processing facilities of the Alfred Wegener Institute, Helmholtz Centre for Polar and Marine Research in Bremerhaven, Germany (Linow et al., 2012). X–ray tomography provides a very high–resolution (1 mm) density profile of the physical properties of the ice. The OH-10 core was retrieved in 2015 using an electric drilling device with a 5.7 cm inner diameter (Icedrill.ch AG). The retrieved core was first stored under controlled temperature conditions (-20°C) at the Chilean scientific station Prof. Julio Escudero (King George Island) and later transported and stored at -20°C in a commercial cold store in Viña del Mar, Chile. The core sections were measured and weighted for density–profile construction and then sub–sampled to a 5 cm resolution for stable water isotope analysis. A visual log and description of each core was carried out to identify possible melt layers and their thicknesses. Subsequently, the samples were melted overnight at 4°C in a refrigerator at the Stable Isotope Laboratory of the Universidad Nacional Andrés Bello (UNAB), Viña del Mar, Chile. To avoid any evaporation, the 5 cm samples were placed in sealed bags (Whirl-pak) and agitated to homogenize the samples before isotopic analysis. Firn and precipitation samples collected from OH in 2014 (Table 1) were analyzed using a liquid water stable isotope analyzer from Los Gatos Research (TLWIA 45EP), located at the UNAB facilities. Measurement precision was higher than 0.1 ‰ for oxygen and 0.8 ‰ for hydrogen isotopes for all analyzed samples. All oxygen and hydrogen stable water isotope data from precipitation and firn core samples are presented in relation to the Vienna Standard Mean Ocean Water Standard (VSMOW) in ‰, as $\delta^{18}O$ and $\delta D$ for oxygen and hydrogen isotopes, respectively.

**3.2 Database and time series analysis**

Stable water isotope data were compared to major meteorological parameters from the region (Fig. 2). For this purpose, the following data sets were incorporated into our analysis: daily and monthly near–surface air temperature ($T_{air}$), precipitation (Pp) and sea–level pressure (SLP) measurements recorded at the Bellingshausen Station (BE) (58.96°W, 62.19°S, 15.8 m a.s.l.) and the O'Higgins Station (OH). These datasets were downloaded from the Global Summary of the Day (GSOD) from the National Climatic Data Center (NCDC, available at: www.ncdc.noaa.gov) and the SCAR Reference Antarctic Data for Environmental Research (READER, available at: https://legacy.bas.ac.uk/met/READER/) (Turner et al., 2004).

The temperature record from OH contains several large data gaps, and so the available data from 1968 to 2015 were compared with those measured at BE to evaluate the possibility of lapsing data from BE to the site due to the data continuity available (uninterrupted record since 1968). The BE and OH data are highly correlated (R=0.97, p<0.01), and so a correction of -1.4°C was applied to the BE data based on linear regression analysis. Other nearby stations such as Esperanza (63.40°S, 57.00°W), were not considered because of a slightly lower correlation (R= 0.96, p<0.01) and the possibility of a higher continental influence on the temperature record.

Sea surface temperature (SST) time series were extracted from the Hadley Centre observation datasets (HadSST3, available at: http://www.metoffice.gov.uk/hadobs/hadsst3/). The HadSST3 provides SST monthly means on a global 5° by 5° grid from 1850 to present (Kennedy et al., 2011a, b). Mean monthly SSTs were extracted from a quadrant limited by 60–65°S and 65–55°W. Missing data or outliers were interpolated from measurements taken in neighboring quadrants.

Relative humidity (*rh*) time series were extracted from data obtained by the calculation of 3 day air parcel backward trajectories under isobaric conditions using the freely–accessible Hybrid single-particle Lagrangian integrated trajectory (HYSPLIT) model (http://ready.arl.noaa.gov/HYSPLIT.php). This three–dimensional model was run using the global data assimilation system (GDAS) archives from NOAA/NCEP (Kanamitsu, 1989) at a 1° latitude–longitude spatial resolution with a 1 hour temporal resolution and is available from 2006 to present (for more details visit:
http://ready.arl.noaa.gov/gdas1.php). For studying the characteristics of air parcels approaching the AP, *rh* time series were obtained from backward trajectories arriving under isobaric conditions (850 hPa) at the OH station. SST and *rh* datasets were resampled to a regional scale defined by high–density trajectory paths (Bellingshausen and Weddell Seas). The resampled fields were defined by the spatial coverage of 1 day backward trajectories. The limits of the resulting quadrant extends from 98° W to 34° W longitude and from 47° S to 76° S latitude. The covered area is representative of the study site because it
includes the region affected by westerly winds and sea ice front during winter time, both factors that exert a high influence on approaching air parcels. A field horizontal mean of resampled *rh* values between sea level and 150 m a.s.l. was computed for this area to construct the *rh* time series used throughout this study.

Altitudinal temperature profiles were obtained from radiosonde measurements carried out at BE between 1979 and 1996 (SCAR Reference Antarctic Data for Environmental Research). Lapse rates were calculated from the temperature difference
between sea level and the 850 hPa level. SAM index time series were obtained from the British Antarctic Survey (BAS, available at: http://legacy.bas.ac.uk/met/gjma/sam.html) (Marshall, 2003). Mean monthly sea ice extent around the AP (between 1979 and 2014) was obtained from the Sea Ice Index from the National Sea & Ice Data Center (NSIDC, available at http://nsidc.org). The measurements of sea–ice extension incorporated in this study considered as a starting point the coastal location of OH, and the sea–ice front in the direction towards KGI as an end point.

**3.3 Stable Isotope time series analysis**

Firn and ice core ages are often dated by analyzing the seasonality of stable water isotope values. In the firn cores analyzed in this study, there was a significant difference in the standard deviation (Sdev > 1.0) of high resolution (5 cm) oxygen isotopes values between firn cores from lower altitudes (OH-4, $\delta^{18}O$ Sdev = 1.2) versus cores from higher altitudes (OH-10, $\delta^{18}O$ Sdev = 2.6) (Table 1). However, within each individual core, the raw datasets obtained from stable water isotope
analysis (Section 3.1) produced low oscillation variance in the isotope–depth profile. Whilst the measured isotope signals were noisy, the values do not fluctuate far from each core's mean. This low variance, added to the fact that the patterns described in the isotope–depth profiles do not correspond to seasonal cycles, means that dating each core using traditional

annual layer counting is complicated. Difficulties from using conventional dating methodology for these firn cores led us to search for other ways to define the time scale of our signals.

We first analysed the $d_{excess}$ data, because $d_{excess}$ is related to seasonal oceanic conditions, and therefore displays an annual signal in this region (Fernandoy et al., 2012). Whilst the stable water isotope results did not display a regular pattern, the

$d_{excess}$ is characterized by a noisy, low frequency oscillation. We use this low–frequency periodic signal to date the core (see below).

We calculated theoretical $d_{excess}$ values at our site using the relationship between $rh$ and SST computed by Uemura et al. (2008): $d_{excess\ meteo} = -0.42 * rh + 0.45 * SST + 37.9$. The suitability of using this relationship for the AP region was assessed by comparing the $d_{excess}$ measurements of the daily precipitation samples taken at the OH station with the corresponding

theoretical values. For each day that a precipitation sample was collected at OH, 3 day air parcel backward trajectories were calculated using the HYSPLIT model. We identified frequent air parcel paths and calculated monthly mean values of $rh$ and SST from re–analysis data (GDAS) along these paths. We found a very good agreement between the measured and our theoretical $d_{excess}$ values, with a correlation of R=0.86 (p<0.01). This high correlation allows us to directly compare a synthetic $d_{excess\ meteo}$ time series and the observational $d_{excess}$ record obtained from each firn core. For the method to be

successful, the resultant depth–age model should maximize the common variability between the two time series.

The $d_{excess}$ signal obtained from stable isotope analysis of firn cores is measured with respect to depth (i.e.: in the space domain). To extract the low frequency seasonal signal, we first computed the Fast Fourier Transform (FFT) of the $d_{excess}$ data, which identifies all the frequencies in the record. For each of the signals, we defined a cut–off frequency for each core using the peak with the second lowest frequency identified in the amplitude spectrum (Table 4). We then reconstructed the

low–frequency $d_{excess}$ signal by calculating the Inverse Fast Fourier Transform (IFFT) from the lowest two identified frequency peaks..

We applied the same procedure to the monthly means of the synthetic $d_{excess\ meteo}$ time series, thus obtaining two low-frequency signals that should show the same seasonal variability due to their dependency on the same variables (i.e.: environmental condition of the moisture source region). We then chose a linear depth–age model that visually matched the

variability in the low–frequency observational $d_{excess}$ data with the variability in the low-frequency synthetic $d_{excess\ meteo}$ data. A single linear stretching factor was calculated using that relationship and applied to the complete firn cores datasets. We used the same depth–age model to put the firn core $\delta^{18}O$ records on a time axis for further analysis using monthly means.

## 4. Results

### 4.1 Precipitation samples

Table 2 and Fig. 4 show the stable isotope results, basic statistics and annual distribution of the precipitation samples collected at the OH station. Combining $\delta D$ and $\delta^{18}O$ values from all precipitation samples enables the definition of a Local

Mean Water Line (LMWL): $\delta D = 7.83 * \delta^{18}O - 0.12$. Backward trajectory analysis of precipitation events reveals high–frequency transport across the Bellingshausen Sea during the 24 hours before the air parcels reach the AP (Fig. 5).

### 4.1.1 Isotope–Temperature relationship

The relationship between the stable water isotope composition of daily precipitation events and daily near–surface temperature ($T_{daily}$) at OH was assessed using linear regression analysis of a sample set of measurements. The sample population consisted of the months with the largest number of precipitation samples (namely December 2008, March 2008 and 2009, June 2008 and October 2014). Only selected months were analyzed to ensure the most complete set of data within a relatively short timeframe to improve the derivation of the calculated relationship. Outliers, given by anomalous $d_{excess}$ values (see Section 3.1, $d_{excess} < -9$ ‰), were filtered out in order to avoid disturbances in the model, as the quality of these samples was likely compromised during storage and transport. It should be noted that this relationship will be compared to firn core time series in Section 4.2.2, in order to reconstruct the air surface temperature at Laclavere Plateau.

Additionally, to investigate the relationship at a monthly scale, a correlation was performed for monthly means calculated from daily events ($\delta^{18}O_{monthly}$; $T_{monthly}$) over the 24 month long precipitation dataset (February – 2008 to March – 2009, and April – November, 2014) (Table3). Considerable differences were identified between the daily and monthly $\delta^{18}O$–T relationships (Table 3; Fig. 4a), which indicated that the isotope–temperature relationship is seasonally–dependent. The seasonality of this relationship was determined using the linear regression slope ($s$) of the daily $\delta^{18}O$–T relationship (see Table 3 for statistical details). The seasonal linear regression between $\delta^{18}O$ and $T_{daily}$ based on 208 precipitation events revealed correlation coefficients (R) higher than 0.6 and a statistical significance ($p$) lower than 0.03. To facilitate the evaluation of seasonal signals, the daily datasets were categorized by season, such that: austral summer (December-January-February, DJF) was characterized using December 2008 data; austral autumn (March-April-May, MAM) was characterized using March 2008 and 2009 data; austral winter (June-July-August, JJA) by June 2008 data; and austral spring (September-October-November, SON) by October 2014 data. If MAM and SON are combined together, considering they are both shoulder seasons, the $\delta^{18}O$–T relationship is defined by the linear regression: $\delta^{18}O = 0.79* T_{daily}-7.76$ (R= 0.74, $p<0.01$). For austral summer, the $\delta^{18}O$–T relationship can be expressed as: $\delta^{18}O = 1.17* T_{daily}-8.19$ (R=0.81, p=0.01); and austral winter as: $\delta^{18}O = 0.35* T_{daily}-8.66$ (R=0.63, p=0.01).

### 4.1.2 Deuterium excess – Temperature relationship

Deuterium excess ($d_{excess}$) was calculated for each precipitation sample from stable water isotope data obtained at OH (see Table 2 for descriptive statistics). To evaluate the proximity to the original evaporation source for each precipitation sample, we examined both the isotope–temperature relationship as well as the near surface temperature and $d_{excess}$ relationship using linear regression analysis. As for the evaluation of the isotope–temperature relationship (Section 4.1.1), daily $d_{excess}$ values for December 2008, March 2008 and 2009, June 2008 and October 2014 were compared with daily mean temperatures, however, correlation coefficients were not significant (Table 3). For the 2008-09 datasets, the $d_{excess}$–T correlation for

monthly averages (calculated from daily events as outlined in Section 4.1.1) was significant (Table 3; Fig. 4b), and the associated linear regression was calculated to be: $d_{excess}$ = -0.60 * $T_{monthly}$+2.12 (R=-0.77, p<0.01). For the 2014 dataset the correlation is not significant (R=0.33, p>0.05).

### 4.1.3 Moisture source of precipitation

Three day air parcel backward trajectories from precipitation events exhibit a wide distribution, probably explaining in part the variability of the isotope–temperature relationship presented in Sections 4.1.1 and 4.1.2. Most of the pathways originate in the Southern Pacific Ocean and the Amundsen–Bellingshausen Seas. The trajectories are primarily derived from the Bellingshausen Sea, the Bransfield Straight and the Drake Passage, Tierra del Fuego and South America's southern tip. In addition, some trajectories (<15%) originate from AP's eastern side. Precipitation trajectories show an almost elliptically distributed pattern with a N40°W orientation, and most follow pathways between 60°S and 67°S. The correlation between monthly mean values of $d_{excess}$ (from precipitation samples) and $d_{excess\ meteo}$ (constructed from the meteorological parameters $rh$ and SST of the high density precipitation pathways) had a significant correlation coefficient of R= 0.86 (*p*<0.01) (Fig. 6), demonstrating that oceanic conditions control most of the precipitation variability.

### 4.2 Firn core samples from the AP

Table 1 shows the stable isotope results and descriptive statistics for firn cores retrieved at the northern AP. The co–isotopic relationship δD–δ$^{18}$O for each single firn core retrieved from LCL is related to the global meteoric water line (GMWL) and the local meteoric water line (LMWL) (Rozanski et al., 1993), with a mean slope of s= 7.91 and an intercept of 3.64 (Fig. 7). These values are very close to those of the LMWL, although with a slightly higher intercept.

### 4.2.1 Age model based on stable water isotopes

Stable water isotope results from each firn core allow the derivation of individual depth profiles of δD, δ$^{18}$O and $d_{excess}$ for each firn core. Lowest noise values and the clearest seasonal patterns were found in $d_{excess}$ profiles ($d_{excess\ core}$) (Fig. 3), similar to findings published by Fernandoy et al. (2012). In Section 4.1.3 it was shown that $\underline{d}_{excess}$ from precipitation samples and $d_{excess\ meteo}$ at sea level are significantly correlated, which suggests a nearby moisture source precipitating in this area. The significant correlation indicates that moisture source conditions of a coastal and oceanic–proximal origin will be represented and preserved in the $d_{excess\ core}$ record and could be used as a chronological marker. The $d_{excess\ core}$ signals were first filtered for their high frequency oscillation patterns and then the remnant signals were compared with the high frequency filtered $d_{excess\ meteo}$ monthly means (See Section 3.3). The comparison between $d_{excess\ meteo}$ and $d_{excess\ core}$ resulted in a close similarity between them. Main peak–valley fitting between both signals led to a monthly mean $d_{excess\ core}$ signal represented on a defined depth–time scale (Fig. 8). The comparison between time series of monthly mean $d_{excess\ core}$ and $d_{excess\ meteo}$ data reveals correlation coefficients of R≥0.67 (*p*<0.01; degree of freedom (df) > 21, see Table 4) for each individual firn core analyzed and obtained from 2006 to 2015. Table 4 summarizes correlation coefficients, statistical significances and time intervals for

each firn core. From the firn cores retrieved from LCL, a single time series was constructed and then compared to the $d_{excess\ meteo}$ time series in order to analyze the isotopic signal for the whole time interval. A higher correlation coefficient of R= 0.75 ($p<0.01$; df= 81) was obtained between the two signals ($d_{excess\ meteo}$ and $d_{excess\ core}$). For the overlapping time interval in OH-9 and OH-10 (February 2012 to January 2014), we only considered data from OH-9, as these samples consist of fresher snow layers with density between 350 and approximately 410 kg m$^{-3}$ (at core depth between 0 and 1 m), and firn with densities between approximately 410 and 530 kg m$^{-3}$ (at core depth between 1 and 7 m) than the corresponding interval in OH-10. This in turn helps to avoid attenuation of the isotopic signal. Although we only considered OH-9 data for the overlapping time interval, we studied the changes in the standard deviation of the isotopic signal ($\delta^{18}$O and $\delta$D) from both firn cores in the common time span. The standard deviation shows a decrease of 16% after one year of deposition in core OH-10 with respect to the same time interval in OH-9; and for 2 to 3 years after deposition, the standard deviation of the signal decreases by 18%.

During visual firn core logging and density measurement, few thin elevated density layers were identified. Melt layers were characterized by regular lateral extension and thickness 10 mm or higher. A 40 mm thick layer was observed in the core OH-10 at depth 8.5m, which was the maximum observed thickness. Melt represented a total of 5% of the accumulated water equivalent column of the cores OH-6, OH-9 and OH-10. The melt layers do not show clear evidence of infiltration nor have a clear pattern of distribution with depth (i.e. an association with summer layers, as there are more or less homogenously distributed along the cores). Other firn high density structures, such as thinner crust–like layers (<10 mm), do not correspond to melt events and are mostly related to wind scour processes and liquid precipitation. Melt and crust layers do not show a clear seasonal pattern with relation to their time equivalent with depth. Around 70% of melt layers counted have a width <10 mm.

### 4.2.2 Seasonal temperature reconstruction from stable water isotopes

The age model developed using the $d_{excess\ core}$ oscillation was later applied to construct a $\delta^{18}$O time series (Fig. 3). From this time series a periodical two–year pattern was identified. This pattern is characterized by positive monthly mean $\delta^{18}$O standardized anomaly values (z = observation - mean * Std. dev.$^{-1}$) between May and November in the years 2008 (z = 0.6), 2010, 2012 and 2014 (z > 1). Additionally, they exhibit a negative correlation to temperatures at BE (z = -0.7 for 2010, and < -1 for 2008, 2012 and 2014). Between June and July in the years 2009, 2011 and 2013, $\delta^{18}$O values (z < -1) are lower than the mean and exhibit a positive correlation to temperature at BE (z < -1) (Fig. 9). Therefore, the two–year periodical pattern mentioned above is represented by even numbered years with austral winter $\delta^{18}$O values higher than the mean, followed by odd numbered years with austral winter $\delta^{18}$O values lower than the mean. Monthly mean $\delta^{18}$O values were transformed to their temperature equivalent using the $\delta^{18}$O–T relationship obtained in Section 4.1.1 from precipitation samples (Fig. 10) to investigate their seasonal behavior.

Calendar seasons in these latitudes do not follow regular patterns (i.e. DJF, MAM, JJA, SON), as seasonality largely depends on the sea ice cover during winter, often extending beyond calendar limits. Large sea ice extent (SIE) leads to a delayed on–

set of spring conditions. In this case winter–like conditions will be extended beyond August. Restricted sea ice extent on the contrary will lead to earlier spring–like conditions (before August). Depending on such conditions, we defined three seasons with their corresponding $\delta^{18}$O–T relationship. These seasons are: (1) an austral transitional season which considers the months from March to May and October–November (MAM-ON) (using precipitation datasets from Table 3, March 2008–2009 and October 2014 for the $\delta^{18}$O–T relationship), (2) an austral winter season which considers the months from June to September (JJAS) (using precipitation datasets from June 2008 for the $\delta^{18}$O–T relationship, Table 3), and (3) an austral summer season which considers months from December to February (DJF) (considers precipitation datasets from December 2008 for the $\delta^{18}$O–T relationship). This basic classification of the seasons to describe the $\delta^{18}$O–T relationship does not explain all of the variability observed in the data, as some particular seasons showed variable behavior when compared to the mean seasonal behavior in the time span covered in this study. In those cases, the seasonal behavior was extended or contracted beyond the boundaries of the main season classification depending on the SIE.

### 4.2.3 Air temperature trends at Laclavere Plateau

In the AP region, lapse rates change seasonally primarily due to variations in the presence and extent of sea ice cover which impacts thermal stability on the lower atmosphere. Mean seasonal lapse rates obtained in this region show a clear seasonal dependency, with the highest rates during DJF (-5.31 °C km$^{-1}$), similar values during MAM and SON (-4.43°C km$^{-1}$, and -4.06°C km$^{-1}$ respectively) and the lowest rates during JJA (-2.73° km$^{-1}$) (Fig. 11).

Monthly near surface temperatures at LCL were estimated using a linear regression analysis based on monthly lapse rates in BE, winter SAM index and SIE from OH (Fig. 12). The resultant equation is: $T_{LCL}= (T_{BE}$ -1.4)+1.13 $(M_{month}*SIE_{OH}+N_{month})$, during the months when sea ice is developed (from May to September) and where $M_{month}$ and $N_{Month}$ represent the slope and intercept of the monthly lapse rate–SIE relationship, respectively. During the months when there is no sea ice (from October to April) the monthly temperature can be calculated from $T_{LCL}= (T_{BE}$ -1.4)+1.13 * H(t), where H(t) is the monthly mean lapse rate value of the month t measured in BE between 1978-1996. Considering these variables, a mean annual air temperature of -7.5°C with a trend of -0.18°C year$^{-1}$ (statistically not significant at $p$=0.05) was estimated for LCL for 2009-2014. Comparatively, if only the $\delta^{18}$O time series data and the isotope–T relationship are considered, a mean annual air temperature of -6.5°C with a trend of -0.33°C year$^{-1}$ (statistically not significant at $p$=0.05) is estimated for LCL for the same period. The correlation between monthly mean temperature at LCL, estimated using the $\delta^{18}$O signal from firn cores and $T_{LCL}$ estimated using the coupled effect of the latitude–corrected temperature record from BE, $SIE_{OH}$ and lapse rates from BE, have a correlation coefficient of R = 0.70 ($p$<0.01). Both signals show a synchronous behavior, also with respect to the air temperature recorded at OH station. No statistically significant correlation was observed between coastal stations (OH and BE) temperature records and the stable water isotope composition of firn cores.

### 4.2.4 Accumulation rates

Density measurements from firn cores were used to construct density–depth profiles. Along these profiles a significant increase of density with depth was obtained. Linear regressions across different sections represent normal firn compaction processes reaching the snow–firn–density boundary (550 kg m$^{-3}$) at the 15.2 m depth. Using these linear regressions and considering the depth intervals delimited in Section 4.2.1 as monthly values, we were able to estimate accumulation rates for different years. A mean accumulation rate of 1770 kg m$^{-2}$ y$^{-1}$ was estimated at LCL for 2008 to 2015. The highest accumulation total was found in 2008 (>2470 kg m$^{-2}$), and the amount noticeably reduced until 2015 (1600 kg m$^{-2}$) with an absolute minimum recorded in 2010 (1060 kg m$^{-2}$) (Fig. 13a). A marked decrease was observed in accumulation during JJA and SON between 2008 and 2015, which could be in part responsible for the overall decreasing rates. However, this trend was found to be statistically non–significant ($p>0.1$).

The highest accumulation occurs during the MAM and SON seasons (Table 5). Accumulation rate estimations derived from the OH-9 and OH-10 cores for the common period 2012 – 2013 only differ by approximately 3%. Other cores from the western flank of the Peninsula (OH-4, OH-5 and OH-6) show that the accumulation in 2008 (common period) depends on altitude, with increasing values from the lower region to the highest point on the LCL (Fig. 13b). The rate of increase was approximately 1500 kg m$^{-2}$ km$^{-1}$ y$^{-1}$ from 350 m a.s.l. to 1130 m a.s.l. in 2008.

## 5 Discussion

### 5.1 Stable water isotope fractionation and post–depositional processes

The stable water isotope composition of precipitation samples from the 2008 and 2014 datasets are very similar to each other, and to firn cores from the western flank and from LCL Plateau (OH-4 to OH-10). Comparing the $\delta^{18}$O signal from OH-6 with data from precipitation samples at OH and with two other cores from the western side of the AP (OH-4 and OH-5) during a common period (March 2008 – August 2008), a $\delta^{18}$O decrease of -0.085‰ km$^{-1}$ was found with increasing distance from the coast (Fig. 14a). The same data set was used to study the $\delta^{18}$O–altitude relationship. The $\delta^{18}$O seasonal means show an altitude dependency that yields seasonal $\delta^{18}$O–altitude patterns. Between OH (0 m a.s.l.) and LCL (1130 m a.s.l.) during MAM, a clear decrease of $\delta^{18}$O with height is observed (-2.4‰ km$^{-1}$ with R=0.97 at p level<0.05), whereas during JJA no significant decreasing $\delta^{18}$O trend is observed (Fig. 14b).

Backward trajectory analysis revealed that the most frequent pathways for air parcels that reach the northern part of AP derive from the Bellingshausen Sea, between 55°S and 60°S throughout the year (Fig. 5). In contrast, localities further south on the AP and in West Antarctica, Ellsworth Land and coastal Ross Sea, respectively, exhibit a stronger continental influence on the precipitation source, depending on seasonal and synoptic scale conditions (Thomas and Bracegirdle, 2015; Sinclair et al., 2012). The LMWL obtained from precipitation samples at OH (m= 7.83) is similar to the Antarctic meteoric water line obtained by Masson-Delmotte et al. (2008) (m= 7.75), and to the GMWL as presented by Rozanski et al. (1993)

(m= 8.13). The similarity between the slope of LMWL and GMWL indicates that the fractionation processes during condensation mostly take place under thermodynamic equilibrium (Moser and Stichler, 1980). These results are consistent with those obtained by other authors for King George Island (Simões et al., 2004; Jiahong et al., 1998). Combining the stable water isotope signature of OH precipitation with time series of meteorological data representative for the conditions prevailing on the ocean near the OH station, a strong relationship with *rh* and SST at the moisture source can be derived. This relationship has been well established, especially for the coastal Antarctic region where moisture transport from the source is generally of short–range (Jouzel et al., 2013). The correlation between the $d_{excess}$ of precipitation and a theoretical $d_{excess\ meteo}$ derived from time series of meteorological data from the surrounding region has shown that both datasets are highly correlated (R= 0.86). Based on this evidence, we propose that the Bellingshausen Sea constitutes the most important source of water vapor for precipitation for the study region at the northern AP. A similar conclusion was drawn for regions further south at the Peninsula (Thomas and Bracegirdle, 2015), however, with an increase of contributions from other local sources (e.g.: Amundsen Sea and continental conditions) to the local precipitation. This has also been observed at the northern AP, where some precipitation events that exhibited a stable water isotope composition beyond the normal range for the region (e.g. 20 August 2009, $\delta^{18}O$ = -19.4‰), were associated with uncommon sources of humidity as shown by the backward trajectory analysis.

In firn cores obtained from the AP, average values from both $\delta^{18}O$ and $\delta D$ decrease as elevation increases to LCL (1130 m a.s.l.), which supports the altitudinal isotope effect identified by Fernandoy et al. (2012) for the region. In addition, standard deviations of seasonal (monthly mean) $\delta D$ and $\delta^{18}O$ values of firn cores from LCL are low and similar to those of firn cores from lower altitudes. Despite the variations in isotopic composition with height, in all firn cores the $\delta D$–$\delta^{18}O$ co–isotopic correlation is very similar to the LMWL obtained from precipitation samples at OH. This provides evidence of the uniformity of the fractionation conditions during the condensation process. Although a slight isotopic smoothing effect was distinguished between the cores (16% after one year of deposition), the distortions caused by post–depositional effects that may alter or homogenize the isotopic signal at this site, such as diffusion, can be considered as limited. The latter indication is well supported by the high accumulation rate in the region that does not allow a prolonged exposition of the freshly fallen snow to the atmosphere. Furthermore, the absence of significant infiltration and percolation associated with melting and refreezing events and the lack of a relationship between ice layers and seasons as well as with the stable water isotope record implies that the isotopic composition is not altered by surface melt infiltration and percolation. Thus, this reassures that post–depositional processes in the LCL region are negligible in the time period analyzed. High firn density peaks, mostly thinner than 10 mm, are represented by discontinuous or non–regular layers and counts for 25% of the total layers. These layers developed by wind ablation on wind–scouring processes, when the air and drifted snow flows against surface irregularities (like Sastrugies); and also by solidification of super–cooled droplets. In both cases, a thin ice crust was formed, as observed during the field seasons and in the core stratigraphy. Melt events, recognized by more regular and thicker firn and ice layers (>10 mm), represent approximately 75% of total high density layers (See Section 4.2.1). Even though these observations are in agreement with the results obtained in this region by Fernandoy et al. (2012) and Aristarain et al. (1990), several studies

(Fernandoy et al., 2012; Simões et al., 2004; Travassos and Simoes, 2004; Jiahong et al., 1998) have identified a significant melt layers in firn cores, mainly from KGI and from the western side of the AP at altitudes below 700 m a.s.l. The limited effect of post–depositional processes due to the high accumulation rates and to the ice layers reducing diffusion (Stichler et al., 2001), along with the high correlation between $d_{excess\,meteo}$ and $d_{excess\,cores}$, confirm that the isotopic variations observed in firn core isotope records are mostly related to isotopic fractionation occurring during condensation and to $rh$ and SST conditions in the vapor source regions.

## 5.2 Stable water isotope and the local temperature relationship

The changing seasonal $\delta^{18}O$–T relationship obtained from precipitation samples shows that the relationship between air temperature and condensation temperature varies throughout the year. The strong similarity in the $\delta^{18}O$–T relationship during MAM and SON contrasts with the pronounced difference of this relationship between DJF and JJA. This highlights the variability of the $\delta^{18}O$–T relationship along the whole year at the northern AP. However, the $\delta^{18}O$–T correlations presented in this study were calculated from precipitation samples of particular months and years, which can induce bias. However, it can be assumed that these datasets give an idea of the variations that can be seen in between seasons in this region. Furthermore, the $\delta^{18}O$–T correlations obtained for MAM and SON (0.77‰ °C$^{-1}$ and 0.61‰ °C$^{-1}$, respectively) are similar to the values obtained by other authors for the AP (Aristarain et al., 1986; Peel et al., 1988). Even though the dataset is capable of representing variations within the time span covered by this study, it is too short to build a consistent baseline for the region. Despite the reduction of the seasonal temperature difference in coastal sites, the difference in the seasonal $\delta^{18}O$–T relationship suggests the existence of processes that disrupt the direct linkage between condensation temperature and surface air temperature. The negative correlation between the $\delta^{18}O$ signal from LCL ice cores and BE (and OH) monthly mean temperatures (Fig. 9 and 10), which is noticeable in some years during JJA, contrasts with the commonly accepted seasonal behavior characterized by a positive correlation between $\delta^{18}O$ and surface air temperatures (Clark and Fritz, 1997). This particular behavior could be related to strong variations in meteorological conditions in the area between BE (OH) and LCL throughout the whole year. Therefore, air temperature on LCL was estimated by two independent methods: lapse rates (vertical temperature gradients) and $\delta^{18}O$–T equivalents. The best correlation between both LCL temperatures estimations was obtained when an extended seasonal behavior was considered (R= 0.70; p<0.01). This result is in agreement with the natural seasonal variability in high latitudes, where the effects of some seasons extend beyond the calendar seasonal temporal limits related to the SIE, as previously explained. Without taking this seasonal variability into account would lead to a misinterpretation of the air temperature reconstruction for LCL, since the $\delta^{18}O$–T correlation would then be rather poor (R= 0.42) and not reflecting the true seasonality in this region. The high similarity in the $\delta^{18}O$–T relationship during MAM and SON can be explained by the seasonal transition between summer and winter, when oceans surrounding the northern AP pass from ice–free to fully ice–covered conditions (or vice versa), respectively. Ice–free ocean conditions are related to seasonal oscillations, which are highly dependent on atmospheric circulation patterns. In this sense, years with a marked negative SAM anomaly are associated with ice–covered sea conditions, whereas positive SAM phases are associated with

ice–free sea conditions (Fig. 12). Other studies (Turner et al., 2016) point to a similar interaction between surface air temperature and SIE at AP and recognized that the SIE's inter–annual variability is related to atmospheric modes. This supports our own observations that the sea ice is important for regulation of surface air temperatures in the region.

### 5.3 Firn age model and accumulation rates

The stable water isotope signal obtained from firn cores shows no regularity in its seasonal behavior and lacks a clear annual oscillation pattern, likely due to the strong maritime influence (Clark and Fritz, 1997). These two criteria prevent the development of an age model by conventional annual layer counting in the isotope record (Legrand and Mayewski, 1997). In this context, the $d_{excess}$ parameter represents a robust time indicator, as it has shown to be principally dependent on $rh$ and SST conditions prevailing in the eastern Bellingshausen Sea where these variables are relatively stable (Jouzel et al., 2013).

The high correlation coefficients (and high statistical significance) obtained for the relationship between $d_{excess}$ and $d_{excess\ meteo}$, as shown in Section 4.2.1, demonstrate that the method used to construct a time series is effective in dating isotope records of firn cores from the northern AP, even at a monthly resolution.

The most frequent $d_{excess}$ values found in the firn cores (3‰ – 6‰) are in agreement with a strong coastal influence scenario as determined by Petit et al. (1991), implying that the $d_{excess}$ relates to $rh$ and SST of the humidity source and not to surface

air temperature (Jouzel et al., 2013). Saigne and Legrand (1987) postulated that $rh$ conditions prevailing at the sea surface have an important effect on the $d_{excess}$ signal of precipitation below 2000 m a.s.l in the study region. The stable water isotope results, in combination with the meteorological datasets, show that precipitation on LCL is highly correlated with $rh$ and SST conditions in the Bellingshausen Sea near the AP.

The irrelevance of post–depositional effects along with the flat topography on LCL suggests that the estimation of

accumulation rates from firn cores is representative of the amount of snow originally precipitated. Moreover, the slight smoothing of the isotope signal after deposition, as well as the small differences in the accumulation rate observed for the common time period of firn cores OH-9 and OH-10, demonstrates that our age model is reliable, as two different data sets yield similar estimations for a common period. The results obtained enable LCL to be classified as a high annual snow accumulation site (Table 5), closely following the estimations of other authors on King George Island dome (Bintanja, 1995;

Zamoruyev, 1972; Jiahong et al., 1998) and on the AP further south of LCL (Dalla Rosa, 2013; Goodwin, 2013; van Wessem et al., 2015), of around 2000 – 2500 kg m$^{-2}$ y$^{-1}$, but differs from the accumulation rate obtained by Simões et al. (2004) and Jiankang et al. (1994) on King George Island dome (600 kg m$^{-2}$ y$^{-1}$). A seasonal accumulation bias was noted, with more favorable conditions for accumulation (i.e. higher precipitation amount) during autumn resulting from more frontal systems approaching the AP (Table 5).

### 5.4 Seasonal variability and disruption of atmospheric conditions

The depletion of δ$^{18}$O with increasing height (altitude effect) and the simultaneous increase in accumulation along the western side of AP at the LCL latitude can be explained with the help of an orographic precipitation model as proposed by

Martin and Peel (1978). This model states that moist air parcels from the Southern Ocean are forced to ascend and cool down when approaching the AP due to the steep topography forming an orographic barrier to westerly winds. The depletion observed in $\delta^{18}O$ reflects the strength of the fractionation process taking place within a short distance and in a low temperature environment (Fig. 15a). Therefore, the isotopic fractionation process occurring at the AP and the direct linear

relationship between $\delta^{18}O$ and the condensation temperature enable us to study temperature behaviour with respect to altitude increase on the basis of $\delta^{18}O$ variations (Craig, 1961). However, whereas MAM air temperatures show a clear decrease with increasing height (atmospheric instability of the lower troposphere), JJA air temperatures exhibit an increase from sea level to 350 m a.s.l (atmospheric stability). At higher altitudes, a decreasing temperature trend is observed (atmospheric instability). The break at 350 m a.s.l during JJA could indicate the existence of a strong stratification within the

lower troposphere on the western side of the AP. In addition, the variations in monthly mean lapse rates measured by radiosondes in BE throughout the year, provide evidence for the existence of a process that modifies the behaviour of the lower troposphere, decreasing the lapse rate (between sea level and 850 hPa) during JJA and considerably increasing it during DJF (Fig. 16).

The close linear correlation identified between lapse rate magnitude and SIE indicates that SIE is an important factor for the

development of these variations, especially between May and September.

The phenomenon previously described is likely linked to the development of an inversion layer in the lower troposphere on the western side of the AP mainly during JJA, which in turn is related to a strong radiative imbalance. During JJA, solar radiation diminishes until it reaches a minimum at the winter solstice. The lack of solar radiation leads to considerable cooling that favors the formation of sea ice and in turn, causes differential cooling between the sea ice surface and the air

above it. As the sea ice surface cools faster than the air above it, a near–surface altitudinal pattern of increasing temperature develops where local atmospheric stability prevails. The layer of atmospheric stability extends from sea level up to at least 350 m a.s.l, where it turns into an atmospheric instability regime. Both regimes together favor the decrease of the overall lapse rate, as temperature first increases and then decreases with height. Conversely, no inversion layer is formed during DJF due to the absence of sea ice and hence, atmospheric instability prevails, which is related to high lapse rates (Fig. 15b and

15c).

The existence of an inversion layer during the months with sea ice coverage might explain the low oscillation of monthly mean temperatures estimated at LCL compared to monthly mean air temperatures at BE (OH). The negative correlation between SAM index and SIE also seems to play an important role, as SAM positive phases enhance the transport of warm and moist air towards the western side of the AP, thus inhibiting the formation of sea ice. This has a direct impact on the

lapse rate as the development of an inversion layer is hindered and therefore air temperatures on LCL are regulated. The interaction between SAM and SIE plays a key role as sea–ice–covered conditions temper the maritime system, favoring continental–like conditions and reducing annual mean air temperature, implying a higher temperature amplitude in BE and OH throughout the year.

The temperature time series estimated from the stable water isotope record ($\delta^{18}O$ and $d_{excess}$) from LCL firn cores exhibits a periodic (biannual) pattern, which can be linked to a similar periodical behavior observed in SAM index and in SIE. The relatively constant temperatures observed during MAM, JJA and SON in years with a positive SAM phase provide evidence that during these seasons condensation is taking place at similar temperatures. Under such conditions (positive SAM), the low variations in the lapse rate throughout the year, along with the low thermal oscillation in BE (OH) explain the presence of a constant condensation temperature, which does not differ much from air temperature during DJF. Conversely, the stronger annual temperature oscillation observed on LCL during negative SAM phases indicates marked variations in condensation temperature throughout the year.

Finally, the proposed inversion layer model (Fig. 16) explains the seasonal variations observed in the $\delta^{18}O$–T relationship of precipitation samples from OH. The distortion of the direct relationship between condensation temperature and surface air temperature by an inversion layer makes it necessary to differentiate $\delta^{18}O$–T relationship according to the lapse rate evolution throughout the year. In this context, MAM and SON were identified as transitional periods in the formation of the inversion layer, mainly because of the sea–ice formation and retreat during these seasons. The seasonal adjustment considered to estimate LCL temperatures must be applied, because the sea–ice cover varies inter–annually in its duration and extension, which in turn produces the inter–annually variable inversion layer. The proposed model for the coastal region on the western side of the AP at OH latitude, is consistent with the observations of Yaorong et al. (2003) on KGI (South Shetland Islands), where several inversion layers developed extending beyond 400 m a.s.l.

## 6 Conclusions

In this study, we examined one of the most complete records of recent precipitation from the northern AP, with a total of 208 single precipitation events and more than 60 m of firn cores. The firn cores retrieved in this work include the accumulation at the northwestern AP region between 2008 and 2014. Precipitation and firn stable water isotope compositions have been compared to different meteorological data sets to determine their representativeness as climate proxies for the region.

The results of our study reveal significant seasonal changes in the $\delta^{18}O$–T relationship throughout the year. For autumn and spring a $\delta^{18}O$–T ratio of 0.69‰ °C$^{-1}$ (R= 0.74) was found to be most representative, whereas for winter and summer the $\delta^{18}O$–T ratio appears to be highly dependent on SIE conditions. The apparent moisture source for air parcels precipitating at the northern AP is mainly located in the Bellingshausen Sea and in the southern Pacific Ocean. The transport of water vapor along these oceanic and coastal pathways exerts a strong impact on the $d_{excess}$ signal of precipitation. The comparison between the $d_{excess}$ signal from the moisture source and the $d_{excess}$ signal from firn cores has been used successfully to date the firn cores from the northern AP, yielding a seven–year isotopic time series in high temporal resolution for LCL.

Based on our dating method we could define LCL as a high snow accumulation site, with a mean annual accumulation rate of 1770 kg m$^{-2}$ y$^{-1}$ for the period 2006–2014. Accumulation is highly variable from year to year, with a maximum and minimum of 2470 kg m$^{-2}$ (in 2008) and 1060 kg m$^{-2}$ (in 2010), respectively. In addition, we identified the presence of a

strong orographic precipitation effect along the western side of the AP reflected by an accumulation increase with altitude (1500 kg m$^{-2}$ y$^{-1}$ km$^{-1}$), as well as by the isotopic depletion of precipitation from sea level up to LCL (-2.40‰ km$^{-1}$ for autumn) and from the coast line up to the ice divide (-0.08 ‰ km$^{-1}$).

The maritime regime present on the western side of the AP has a strong control on air temperatures, observed as restricted summer/winter oscillation, and is reflected in a poor seasonality of the $\delta^{18}$O and $\delta$D profiles in firn cores. Recent climatic conditions can be only reconstructed from $\delta^{18}$O time series obtained from LCL firn cores when considering an inversion layer model during winter season. The strength of the inversion layer likely depends on SIE and SAM index values. Taking into account the effect of the inversion layer on the isotope–temperature relationship, we observe a slight cooling trend of mean annual air temperature at LCL with an approximate rate of -0.33°C y$^{-1}$ for the period sampled by the examined firn cores (2009-2014). This finding is in line with evidence from stacked meteorological record of the nearby research stations as determined by Turner et al. (2016).

Our results demonstrate that the stable water isotope composition of firn cores retrieved from LCL is capable of reproducing the meteorological signal present in this region, validating it as a valuable proxy for paleo–climate reconstructions in the northern AP region. Environmental (atmosphere and ocean) and glaciological conditions present at LCL, a ~350 m thick ice cap, together with an almost undisturbed isotopic record are optimal prerequisites for the preservation of a climate proxy record with a high temporal resolution. Consequently, LCL is a suitable site for recovering a medium–depth ice core to investigate climate variations during the last centuries in the northern AP region.

## 6 Acknowledgements

The present work was funded by the FONDECYT project 11121551 and supported by the Chilean Antarctic Institute (INACH), the Chilean Air Force and Army logistic facilities. We want to show our gratitude to the Universidad Nacional Andres Bello for supporting this study. We also greatly thank our colleagues, who made field work conditions less severe, especially to Daniel Rutllant for his support on the logistic and field safety. We would like to thank all people involved on the laboratory work, especially to Ivonne Quintanilla, who carried out the sample processing at UNAB and to Dr. Johannes Freitag, for his support with the X–Ray tomography processing at AWI. Tracy Wormwood is greatly thanked for her support editing this manuscript. We highly appreciate the comments of two anonymous reviewers, who provided helpful observations that greatly contributed to improve this manuscript. Finally, we thank the dedicated work of the editor of this article Benjamin Smith.

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

**Figure 1:** Study area and location of the firn cores presented in this work. (a) Detail of the study zone: the green point shows the Chilean Station O'Higgins (OH) at the west coast of the Antarctic Peninsula. Firn cores retrieved between 2008 and 2015 are shown by red dots.

(b) Location of O'Higgins and Bellingshausen Station and Laclavere Plateau, which are mentioned through the text. Satellite image (Landsat ETM+) and digital elevation model (RADARSAT) available from the Landsat Image Mosaic of Antarctica (LIMA) (http://lima.usgs.gov/).

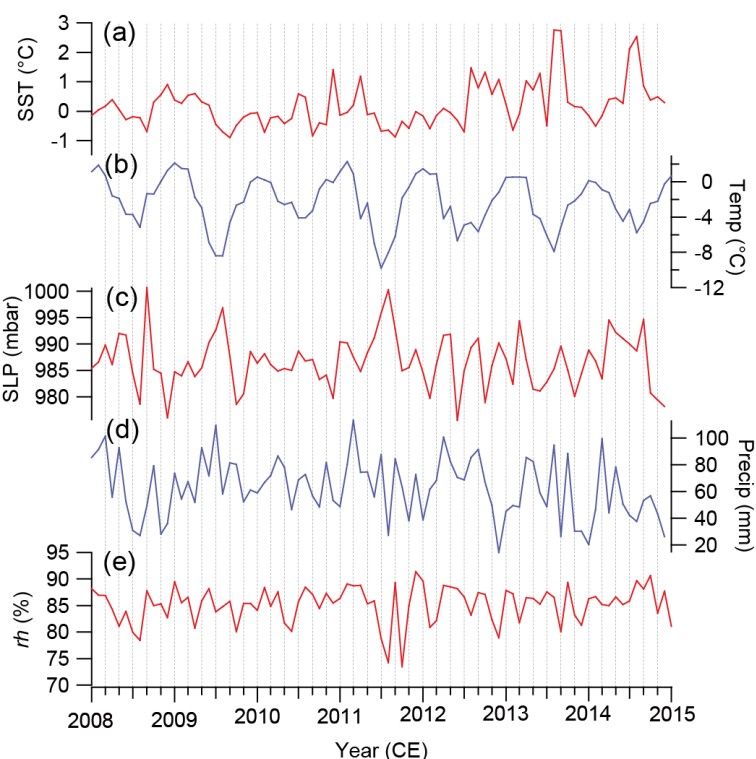

**Figure 2:** Monthly meteorological data sets used in this study (a) Sea surface temperature (SST), (b) Air temperature (Temp), (c) Sea level pressure (SLP) and (d) Precipitation amount (Precip) from Bellingshausen Station (BE) on King George Island and (e) Relative humidity (*rh*) from the Southern Ocean surrounding the northern Antarctic Peninsula (AP) region. Data shown in the figure is available from the READER dataset (https://legacy.bas.ac.uk/met/READER/) (Turner et al., 2004).

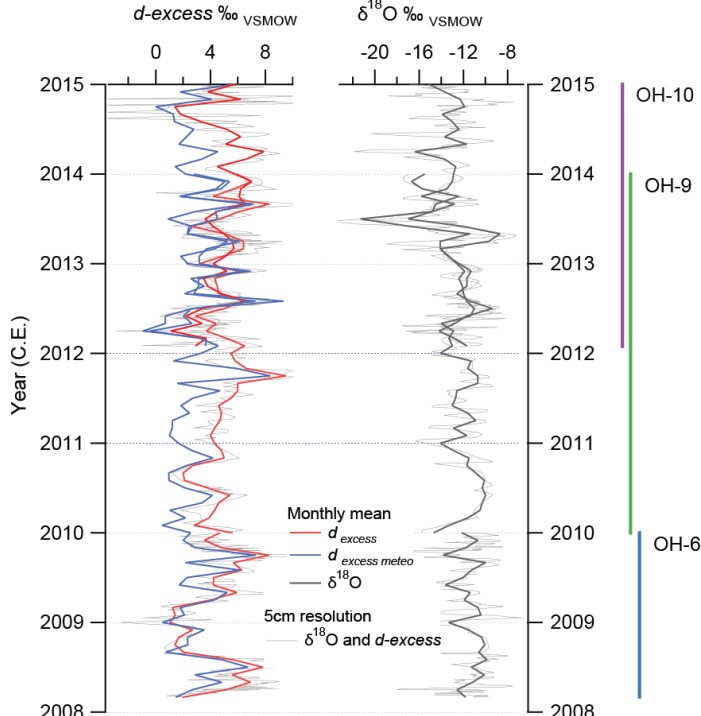

**Figure 3:** Time series for firn cores OH-6 (light blue line right), OH-9 (green light right) and OH-10 (purple line right) derived for $\delta^{18}O$ (bold black line middle left panel) and $d_{excess}$ (red line left panel) records using a theoretical $d_{excess}$ ($d_{excess\ meteo}$) value (blue line left panel). The $d_{excess\ meteo}$ is calculated from Sea Surface Temperature (SST) and Relative Humidity ($rh$) according to Uemura et al. (2008). All three cores are located at the same location within the GPS navigator horizontal error (<10m).

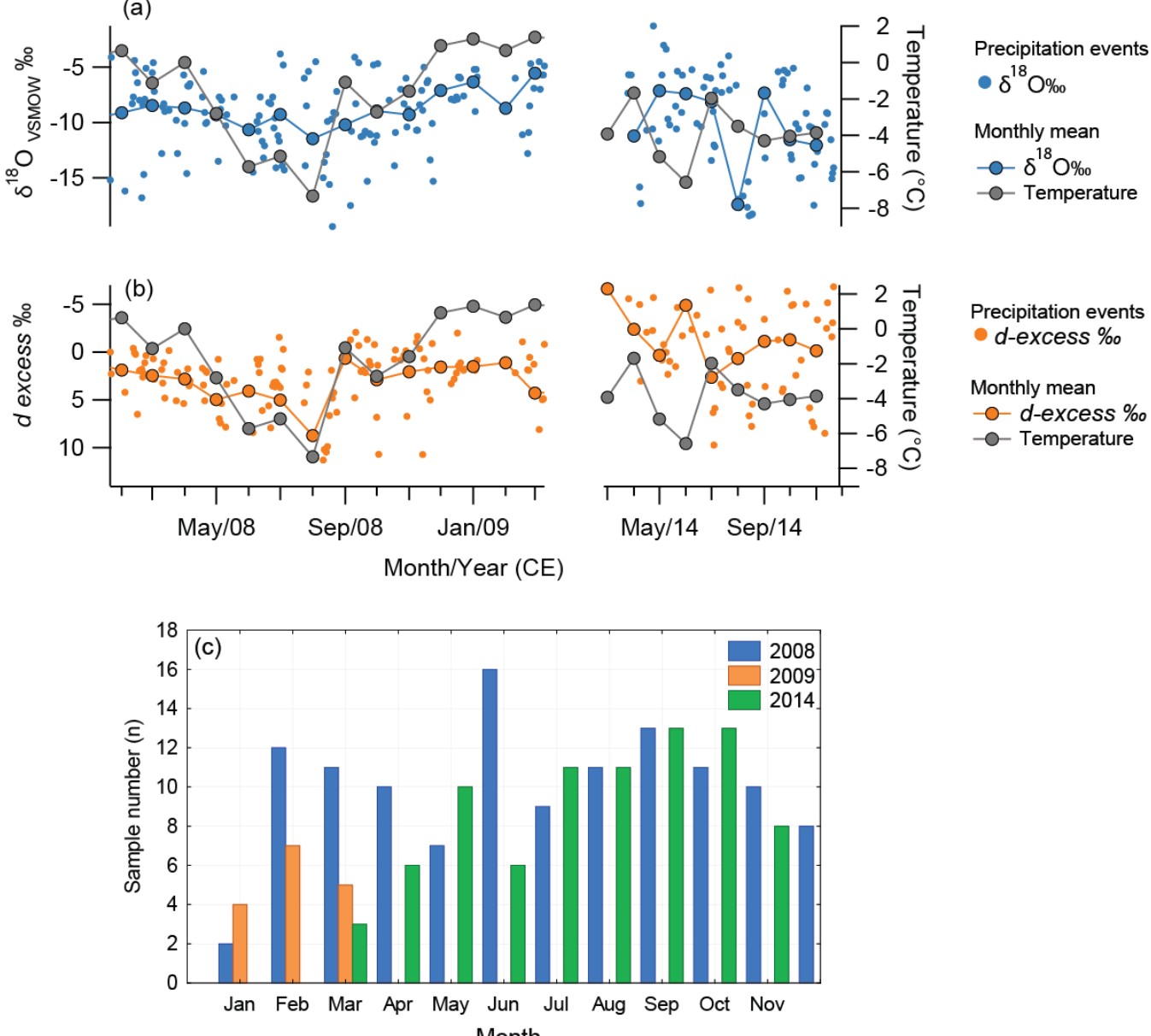

**Figure 4:** Stable water isotope composition of precipitation events and air temperature at O'Higgins Station. (a) shows the $\delta^{18}O$ composition of precipitation of single daily events (small solid blue dots) and monthly means (big solid blue dots and line) and (b) deuterium excess ($d_{excess}$) of single daily events (small orange dots) and monthly means (big orange dots and line). In both (a) and (b) monthly mean air temperature is also shown (grey solid dots and line). (c) Histogram showing the monthly distribution of precipitation samples (n) collected at O'Higgins Station in 2008, 2009 and 2014.

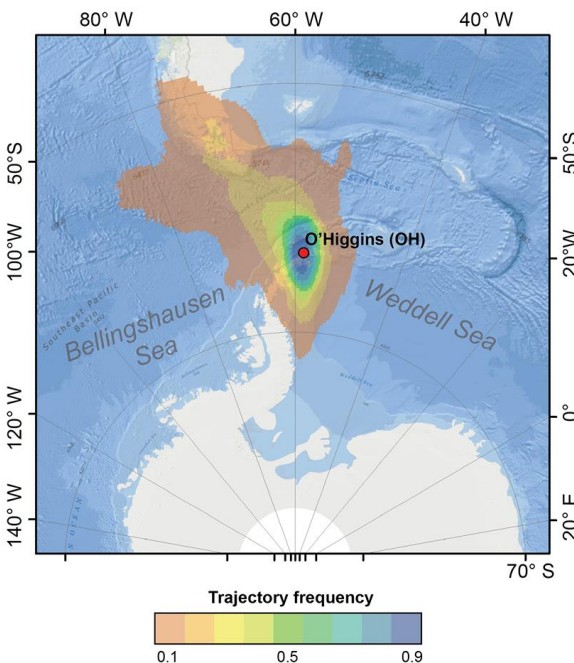

**Figure 5:** Frequency distribution map of the main transport paths of air masses approaching the northern Antarctic Peninsula (AP). Translucent red colours represent the lowest frequency and blue colours the higher frequency. In general, most of the air masses arriving at the AP are coming from the Bellingshausen Sea and the South Pacific Ocean.

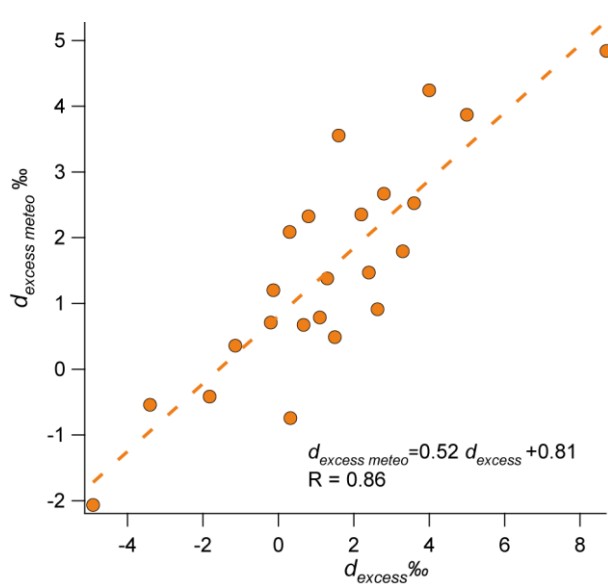

$$d_{excess\ meteo}=0.52\ d_{excess} +0.81$$
$$R = 0.86$$

**Figure 6:** Correlation between monthly mean deuterium excess values ($d_{excess}$) from precipitation samples and theoretical deuterium excess values ($d_{excess\ meteo}$) calculated from meteorological parameters of the moisture source region according to Uemura et al. (2008).

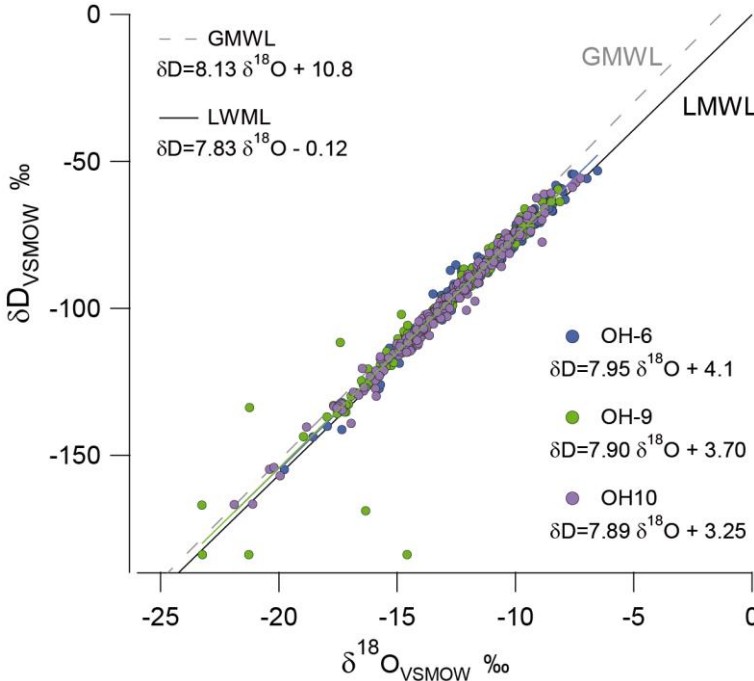

**Figure 7:** Co–isotopic regression of firn cores OH-6 (solid blue dots), OH-9 (solid green dots) and OH-10 (solid purple dots). All slopes and intercepts are very close to each other as well as to the global and local meteoric water line (GMWL – grey dashed line, and LMWL – black solid line, respectively). Stable water isotope analysis for each firn core was made at 5 cm resolution, representing 630 samples in total.

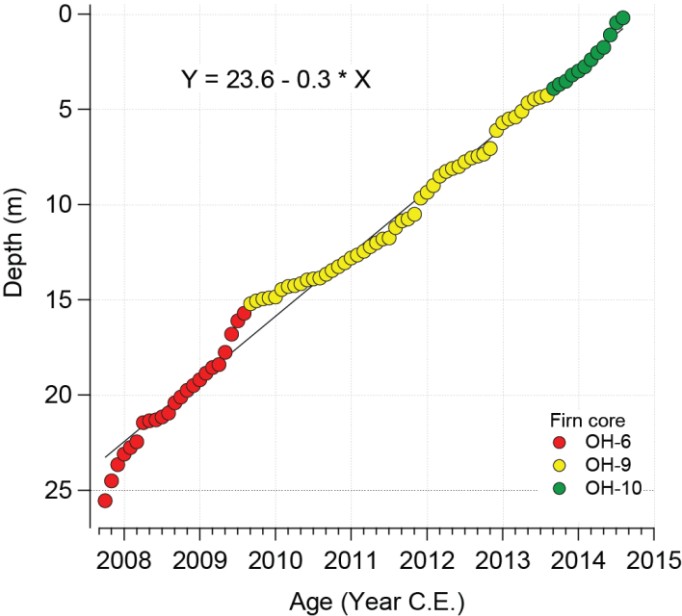

**Figure 8:** Depth–Age model for firn cores OH-6, OH-9 and OH-10 retrieved from Laclavere Plateau. The linear relationship between depth and time was constructed based on the cores measured $d_{excess}$ oscillation and a synthetically constructed $d_{excess}$ from meteorological observations. Note that the Depth axis was intentionally inverted to visualize the surface (0 m) on top.

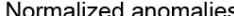

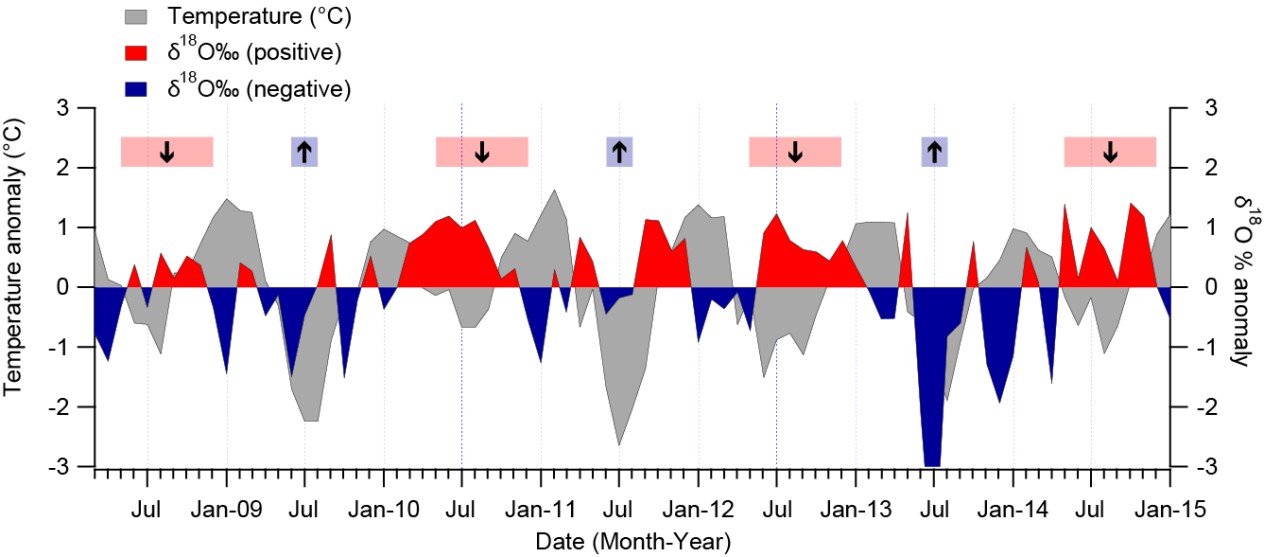

**Figure 9:** Standardized anomalies for air (monthly mean) temperatures (solid grey colours) registered at Bellingshausen Station (BE) on King George Island and a composite δ¹⁸O time series derived from firn cores OH-6, OH-9 and OH-10 from Laclavere Plateau. Upper translucent red (blue) boxes show period of positive (negative) anomalies, down (up) arrow shows the negative (positive) stable water isotope – temperature relationship. Both time series were detrended prior to constructing the time series of anomalies.

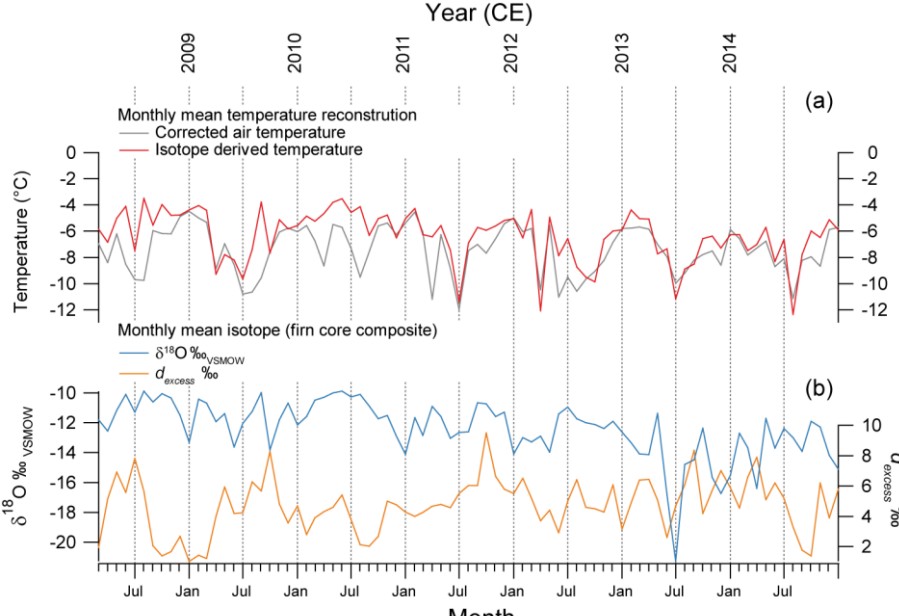

**Figure 10:** (a) Monthly mean air temperature reconstruction for LCL between March 2008 and January 2015 based on air temperature corrected by a seasonal factor and altitudinal gradient (grey line) and based on a δ¹⁸O composite time series derived from firn cores from LCL corrected by a seasonal factor (red line), respectively. (b) δ¹⁸O and $d_{excess}$ monthly mean composite time series of LCL firn cores used for the temperature reconstruction of the upper panel.

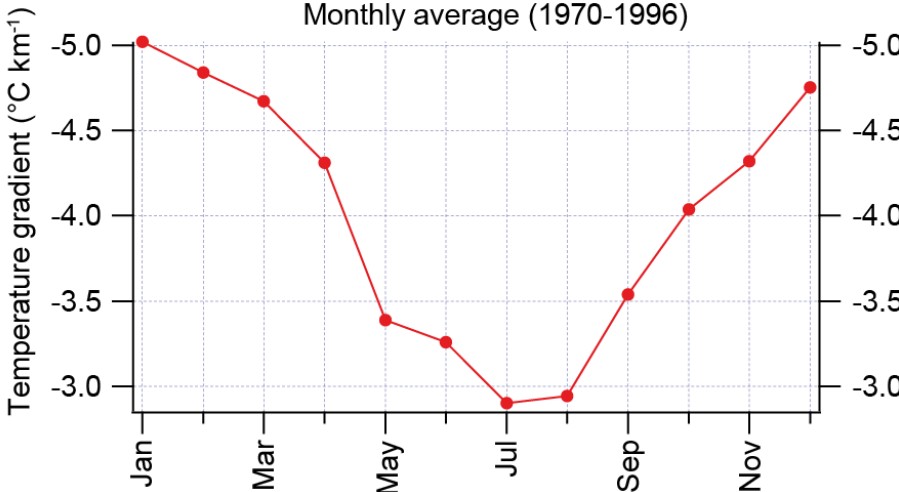

**Figure 11:** Temperature lapse rate from sea level to 850 hPa level at Bellingshausen station (BE), King George Island, Antarctica. The data is shown as the monthly mean value of observation between 1979 and 1996 (SCAR Reference Antarctic Data for Environmental Research).

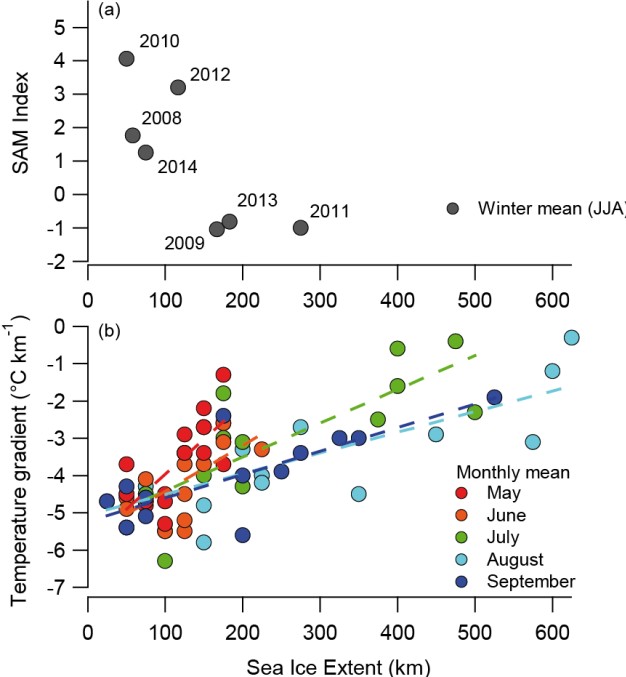

**Figure 12:** Sea ice extent (SIE) from O'Higgins Station (OH) and its relationship to (a) the Southern Annular Mode and (b) to the temperature gradient between sea level and 1100 m a.s.l. at the Laclavere Plateau (LCL). SIE data is from the National Snow & Ice Data Center data set (NSIDC). Sea ice extent, defined as the extension of the oceanic region covered by at least 15% ice, exhibits a negative relationship to the Southern Annular Mode between 2008 and 2014. The relationship to the temperature gradient is positive. A decreasing
10     seasonal pattern of the temperature gradient can be observed from May to September (1979 – 1996).

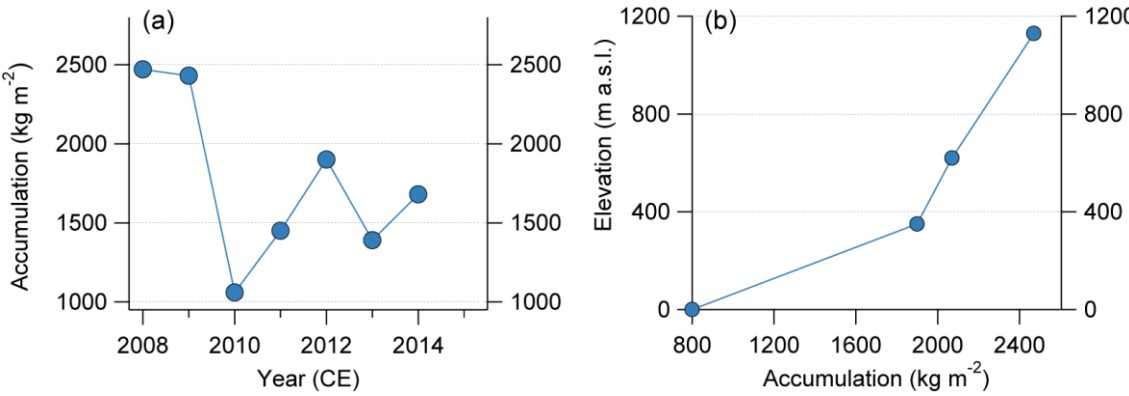

**Figure 13:** (a) Accumulation rates for Plateau Laclavere during 2008 – 2014 estimated from the stable water isotope composition of firn cores OH-6, OH-9 and OH-10 and their respective density profiles. (b) Accumulation variability for the west flank of the northern Antarctic Peninsula from the coast to Laclavere Plateau. Accumulation rates were derived from precipitation at O'Higgins Station at sea level and firn cores (OH-4, OH-5 and OH-6) for higher altitudes.

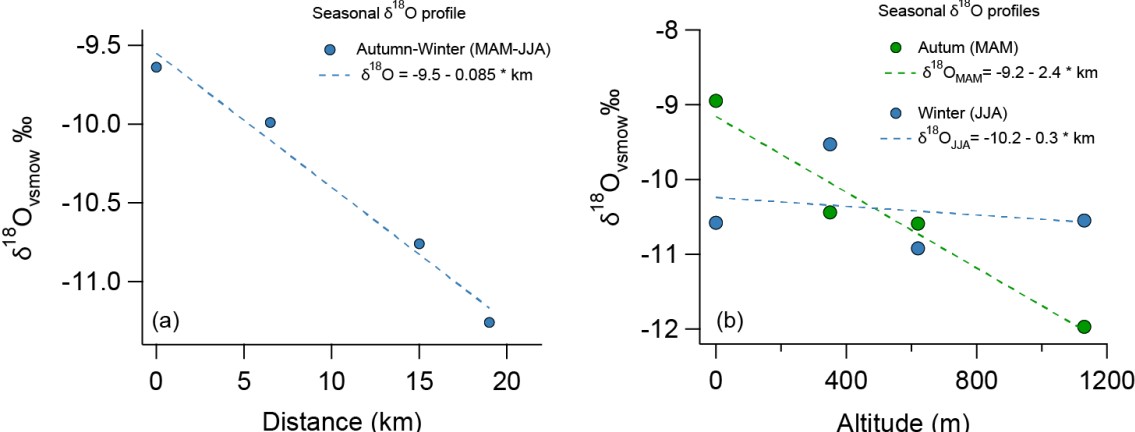

**Figure 14:** $\delta^{18}O$ profile with relation to (a) the distance from the coast at O'Higgins Station (OH) and at different points on the west flank of the AP (6.5 km (OH-4), 15 km (OH-5) and 19 km (OH-6)) and (b) altitude at 350 m (OH-4), 620 m (OH-5) and 1130 m a.s.l. (OH-6) during autumn (MAM) (green solid dots) and winter (JJA) (blue solid dots).

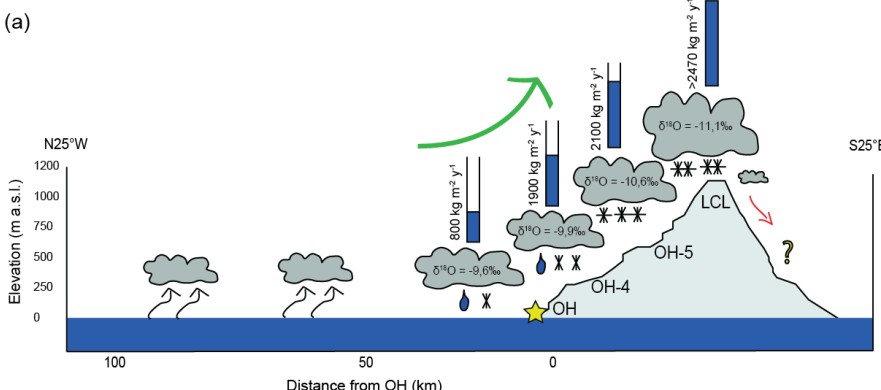

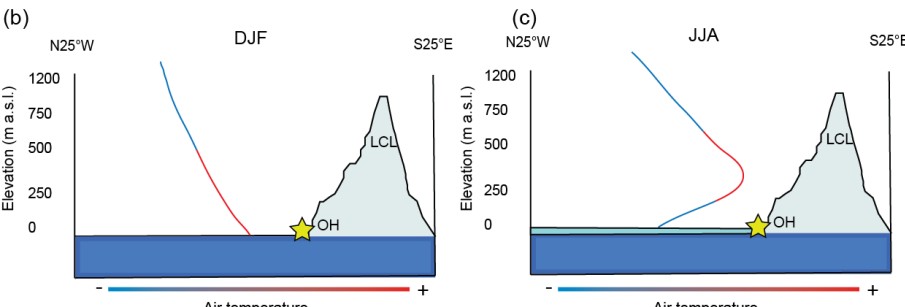

**Figure 15:** (a) Schematic chart showing the orographic barrier effect of the AP on the stable water isotope depletion and accumulation rate at different altitudes, firn core locations (OH-4, OH-5 and OH-6) and distances from the coast (OH); (b) temperature gradient (adiabatic cooling) during DJF (summer) and sea–ice–free conditions; (c) inversion layer in the lower troposhere during sea–ice–covered conditions in JJA (winter).

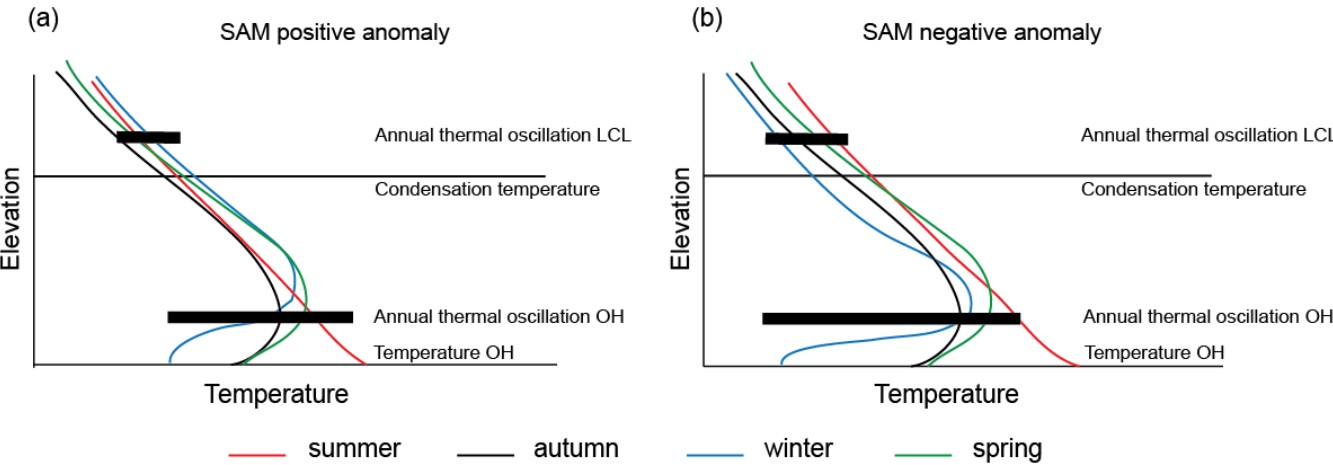

**Figure 16:** Sea level to Laclavere Plateau temperature oscillation scheme during summer (DJF), autumn (MAM), winter (JJA) and spring under: (a) positive SAM anomaly conditions and (b) negative SAM anomaly conditions.

**Table 1:** Statistical summary of the geographical location and water stable isotope composition of all firn cores examined in this work. OH-4 and OH-5 correspond to cores retrieved on the west side of the AP, whereas OH-6, OH-9 and OH-10 were retrieved at LCL on the east–west divide. All cores were analyzed in a 5 cm resolution.

| Core | OH-4 | OH-5 | OH-6 | OH-9 | OH-10 |
|---|---|---|---|---|---|
| Coordinates | 57.80°W, 63.36°S | 57.62°W, 63.38°S | 57.76°W, 63.45°S | 57.76°W, 63.45°S | 57.76°W, 63.45°S |
| Altitude (m a.s.l.) | 350 | 620 | 1130 | 1130 | 1130 |
| Depth (m) | 15.8 | 10.6 | 11.0 | 11.7 | 10.2 |
| Drilling date | Jan 2009 | Jan 2009 | Jan 2010 | Jan 2014 | Jan 2015 |
| $\delta^{18}O$ (‰) | | | | | |
| Mean | -10.4 | -10.2 | -12.0 | -12.8 | -12.9 |
| Sdev | 1.2 | 1.5 | 2.5 | 2.5 | 2.6 |
| Min | -14.1 | -14.2 | -19.8 | -23.3 | -21.9 |
| Max | -7.0 | -7.2 | -6.5 | -8.1 | -7.3 |
| $\delta D$ (‰) | | | | | |
| Mean | -78.9 | -78.1 | -91.4 | -97.5 | -98.8 |
| Sdev | 9.7 | 12.0 | 19.4 | 21.0 | 20.5 |
| Min | -108.2 | -111.2 | -154.9 | -183.8 | -166.8 |
| Max | -54.0 | -52.1 | -53.2 | -59.6 | -55.8 |
| $d_{excess}$ (‰) | | | | | |
| Mean | 4.0 | 3.9 | 4.4 | 5.1 | 4.7 |
| Sdev | 1.5 | 1.7 | 2.8 | 1.9 | 2.7 |
| Min | 0.5 | -0.6 | -2.6 | 0.0 | -6.5 |
| Max | 8.6 | 8.2 | 15.0 | 11.0 | 11.3 |
| n (samples) | 318 | 213 | 208 | 232 | 190 |

**Table 2:** Statistics of the stable water isotope composition of precipitation samples collected at OH Station on the AP 2008–2009 and 2014.

| Station | O'Higgins | O'Higgins |
|---|---|---|
| Sampling interval | Feb 2008 – Mar 2009 | Apr – Nov 2014 |
| Coordinates | 63.32°S, 57.90°W | 63.32°S, 57.90°W |
| $\delta^{18}O$ (‰) | | |
| Mean | -9.2 | -10.1 |
| Sdev | 3.3 | 4.4 |
| Min | -19.4 | -18.4 |
| Max | -3.8 | -1.3 |
| $\delta D$ (‰) | | |
| Mean | -70.5 | -81.9 |
| Sdev | 26.4 | 34.2 |
| Min | -150.6 | -148.4 |
| Max | -21.8 | -16.0 |
| $d_{excess}$ (‰) | | |
| Mean | 2.7 | 3.8 |
| Sdev | 4.2 | 4.7 |
| Min | -6.6 | -1.8 |
| Max | 22.3 | 14.7 |
| n (samples) | 139 | 69 |

**Table 3:** Statistical correlations and linear regression significance of the $\delta^{18}O$ and $d_{excess}$–Temperature relationship for precipitation samples. The Slope and Standard error are related to the linear regression constructed from the isotope–Temperature regressions at daily and monthly scales.

| | Corr. coef. (R) | Slope (s) | Std. Error | p-value |
|---|---|---|---|---|
| **Precipitacion $\delta^{18}O_{daily}$–$T_{daily}$** | | | | |
| DJF | 0.81 | 1.17 | 0.62 | 0.01 |
| MAM | 0.65 | 0.77 | 2.08 | <0.01 |
| JJA | 0.63 | 0.35 | 1.74 | 0.01 |
| SON | 0.6 | 0.61 | 2.88 | 0.03 |
| **Precipitacion $\delta^{18}O_{monthly}$– $T_{montly}$** | | | | |
| All data–set | 0.30 | 0.28 | 2.36 | >0.05 |
| 2008-2009 | 0.74 | 0.41 | 1.03 | <0.01 |
| 2014 | -0.32 | -0.77 | 3.60 | >0.05 |
| **Precipitacion $d_{excess}$–$T_{daily}$** | | | | |
| DJF | -0.37 | -0.86 | 1.66 | >0.1 |
| MAM | 0.09 | 0.10 | 2.61 | >0.1 |
| JJA | -0.28 | -0.29 | 3.61 | >0.1 |
| SON | -0.42 | -0.74 | 5.97 | >0.1 |
| **Precipitacion $d_{excess}$–$T_{montly}$** | | | | |
| All data–set | 0.06 | 0.07 | 2.98 | >0.05 |
| 2008–2009 | -0.77 | -0.60 | 1.43 | <0.01 |
| 2014 | 0.33 | 0.65 | 2.35 | >0.05 |

**Table 4:** Correlation between deuterium excess ($d_{excess\ meteo}$) values calculated from monthly mean meteorological data (SST and $rh$) and water stable isotope monthly means for all cores used in this study. Degrees of freedom (df) defined as: n-2 for each correlation and Cut-off frequencies for the FFT analysis are $cm^{-1}$, expressing the lower and upper boundary of the integrated spectrum, respectively.

| Core | OH-4 | OH-5 | OH-6 | OH-9 | OH-10 |
|---|---|---|---|---|---|
| Time interval | Jan 2006 – Jan 2009 | Mar 2007 – Jan 2009 | Mar 2008 – Jan 2010 | Feb 2010 – Jan 2014 | Feb 2012 – Jan 2015 |
| Corr. coefficient | 0.72 | 0.79 | 0.81 | 0.78 | 0.67 |
| p-value | <0.01 | <0.01 | <0.01 | <0.01 | <0.01 |
| df | 34 | 21 | 21 | 47 | 34 |
| Cut-off freq. $cm^{-1}$ | 3.14 25.16 | 4.69 37.56 | 4.81 43.27 | 4.31 43.10 | 5.26 78.95 |

**Table 5:** Accumulation rates calculated for all firn cores used in this study. All rates are shown as seasonal and annual mean values with respect to the time interval covered by each core.

| AP Accumulation (kg m$^{-2}$) | | | | | |
|---|---|---|---|---|---|
| | Western Flank | | LCL | | |
| | OH-4 | OH-5 | OH-6 | OH-9 | OH-10 |
| DJF-MAM | 1121 | | | | |
| JJA-SON | 1300 | | | | |
| **2006** | **2510** | | | | |
| DJF-MAM | 1650 | >1380 | | | |
| JJA-SON | 1300 | 1150 | | | |
| **2007** | **2950** | **>2530** | | | |
| DJF-MAM | 1130 | 1020 | >1530 | | |
| JJA-SON | 770 | 1050 | 940 | | |
| **2008** | **1900** | **2070** | **>2470** | | |
| DJF-MAM | | | 1090 | | |
| JJA-SON | | | 1340 | | |
| **2009** | | | **2430** | | |
| DJF-MAM | | | | 700 | |
| JJA-SON | | | | 360 | |
| **2010** | | | | **1060** | |
| DJF-MAM | | | | 680 | |
| JJA-SON | | | | 770 | |
| **2011** | | | | **1450** | |
| DJF-MAM | | | | 1170 | 1080 |
| JJA-SON | | | | 730 | 690 |
| **2012** | | | | **1900** | **1770** |

