# Peer review of "New insights into the use of stable water isotopes at the northern Antarctic Peninsula as a tool for regional climate studies"

_The Cryosphere, 2016_

## Referee Comment (RC1) · Anonymous Referee #1 · 17 Feb 2017

Review of TCD manuscript: New insights into the climatic signal from firn cores at the northern Antarctic Peninsula, Fernandoy et al. 2017

Notation: Page-line(s)

The paper presents valuable glacio-chemical data for the Antarctic Peninsula. The authors present water stable isotope data sets of precipitation and firn cores collected near O'Higgins station. The authors present an innovative method to obtain the time scale of the the firn cores which, the authors claim, cannot be dated by traditional methods such as annual cycles counting of d18O variations. The authors then discuss the isotope-temperature relationship at different seasons and conclude that an isotope-temperature relationship cannot be valid for all seasons, but rather depends on seasonal variability of oceanic conditions. The paper is well written, and structured.

The data quality is good and the figures are well executed. However, I consider that the title of the manuscript is misleading because climatic signals per se cannot be obtained from such short cores, but rather the usefulness of such cores can be assessed in terms of possible tools for climate studies, given longer cores are retrieved at the site. I consider that the manuscript can be accepted for publication at The Cryosphere after the authors address the major comments described in this review.

Major comments

Title I found the title misleading since the firn cores temporal extent is too short as to infer climatic signals from them. A suggested title would be: Water stable isotope and deuterium excess records from precipitation and firn cores from northern Antarctic Peninsula as tools for climate studies in the region.

Database I am concerned about the author's use of the Tair and altitudinal T profiles from BE station. Why the authors would expect this temperature to be representative for the core sites? Later in section 2.3 the authors mention that daily mean temperatures are available at OH, then why not to use that data set instead of the BE-data? Please elaborate and justify the use of the BE data set.

Stable isotope time series analysis In section 2.3, page 5-25, the authors state that the d-excess signal obtained from the firn cores was depicted against depth and filtered using IFFT to be compared with d-excessmeteo, but the authors never showed the original raw core isotope data, nor discussed the quality of it, e.g. amplitude of the signal, seasonality, possible melt, signal differences /similitude at the different core sites, etc. The authors must describe the raw isotope data before applying any further statistical method to compare it with either instrumental or modelled data. Before assessing the quality and representativeness of the raw data it is not possible to carry on with other comparisons. Further, the authors obtained the time scale of core d-excess based on the "the strong similarities between both signals (d-excess and d-excess meteo)"; even thought this method is quite interesting and innovative, and both profiles clearly agree

(Fig. 3), the authors must indicate why a more traditional dating was not preformed, e.g., annual layer counting, which would be the obvious first attempt on dating the cores. The authors must address this issue and justify the dating method used in order to show its value over more traditional approaches. The authors must also indicate the error in the time scale obtained. This is shortly introduced in the text in section 4.2 but it should be mentioned already in section 2.3.

Seasonality The authors define seasons and sample sets to each season. However, the authors have a limited number of years (2008-2014) as to construct a representative seasonal signal of d18O in precipitation at the study site to use as a baseline to compare with particular monthly means of a given year. The authors use 1 month of a particular year to describe a seasonal signal without discussing its representativeness. Therefore, the results must be explained as results for a particular year rather than as to a "seasonality", mentioning the results limitations.

Glaciological setting The authors must provide a more detailed description of the study site, e.g. glaciological setting, meteorology of the study site, earlier studies of in the OH area, if available. This could be included in an additional section as "study site"

Minor comments

1-23: Be more specific when given the results. The results presented here are a snapshot of a region situated at Antarctic Peninsula but they don't necessarily reflect the whole Antarctic Peninsula situation. Unless a geographical significance study is done, please clearly state to which region of the Antarctic Peninsula are your results representative.

2-2: I have seen Antarctic Peninsula being referred as AP or APIS (when talking about Antarctic Peninsula Ice sheet), I wonder why the authors chose API.

2-24: at several stations

2-27: with a marked warming

3-22: the authors only mention the "high temporal resolution" of their records but did not talk about the temporal extent of their records. Since the title of the manuscript involves climate, the temporal extent of the firn cores is as important as the resolution and should be introduced together.

4-3: austral summer campaigns

4-3: please remove "several shallow–depth firn cores (totalling more than 60 m) were retrieved from the northern part of the API". There is no need to add this vague information if the authors are going to give more details of the cores in the next sentences.

4-4: Add link to Figure 1. Label O'Higgins station as "OH", also add info on image source and contours details.

4-8: Please be more specific about how many samples were discarded and why.

10-13: ...profile of density and physical properties of the ice

10-13: add info of the drill, handling, storage, and sampling for all cores. This can be summarized in a table.

4-16: To which institution/facility in Viña del Mar? Please add information about sample melting and storage during melting (type of vial). As mentioned by the authors in page 20-21, secondary processes during storage and transport (and also melting of the sample) can perturb the isotopic signal.

4-18: Indicate where the water stable isotopes were analysed (instrument, method, etc) and the accuracy for all the cores. This is depicted in Table 1 but it is not clearly stated in the text. Cite references to values shown in Table 1.

4-18: Please indicate where the firn and precipitation samples were collected at OH.

In Fig. 2, show source of the data.

Please add the BE station in Figure 1.

5-1: Please indicate the arrival point of the HYSPLIT trajectories. Was 1-backtrajectory estimated or a cluster of backtrajectories?

5-22: "for the whole region". The author's refer to the whole Antarctic Peninsula? If, yes, the authors might reconsider limit the interpretation to the nearby study area. It is unclear to me if the HYSPLIT backtrajectories were set to end at OH or to other sites in the Peninsula. Please clarify this.

Figure 3: it is not clear to me if cores OH-10, OH-9, and OH-6 were drilled at the exact point, how close were there drilled? This info cannot be inferred from Table 1 which shows the exact coordinates for all three cores. The core sites could be shown as a zoomed-in section in Fig. 1.

6-25,26: this should be stated earlier in the text, e.g. in section 2.3 when figure 3 was shown.

6-27: Remove OH

6-28: Please mention how many samples were rejected and why. This is mentioned earlier in the text but is not discussed.

Figure 6: please add labels for the study site, and Bellingshausen Sea in the map.

7-6: please mention how the authors defined the seasons, e.g. DJF-→ summer. Also, please explain better how you selected the set of samples representative of each season; as it is written in the text, it appears that the set for each season was selected upon the number of samples of arbitrary months which might cause bias, especially in section 3.1.1 where the seasonal regression slopes are discussed. Please consider using all samples available for each season or limit your discussion to the represented months but not to seasonal scales. Please discuss the annual precipitation distribution at the study site if available and put your results in that context. Also discuss how precipitation samples were taken, is a precipitation event identified as one precipitation sample or are the samples taken and identified on a daily (hourly) basis? Indicate in a

figure the number of samples per month and also the volume per sample/event.

7-9: Please discuss how outliers were removed in section 3.1.1 (when first referred) .

7-11: please include the error and significance of the regression slopes.

7-13: the authors must justify why they believe a particular month is representative of a season. If the authors have data for all months, why to assume only one month as representative of a season? This issue has been addressed in a previous comment (7-6). This is very important to clarify as the authors are attempting to link their results to climatic features.

7-16: An inverse behaviour between July and June or between July-June compared to MAM and SON? Please explain.

7-22: do not use "weak", use instead: correlation coefficients for these comparisons are not significant (indicate level of confidence).

7:3.1.3 Please indicate geographical sites in a figure (in Fig. 6 for example).

Figure 8 can be removed and the equation of the regression line can be given in section 3.1.3.

8-5: define GMWL and LMWL in the text (now is only defined in Fig. 9)

8-6: previously you defined the slope as "s", now it is mean slope as "m" but not defined, either use s again or define mean slope as "m".

8-9: Please remove the first sentence as it is unnecessary.

8-10: as mentioned in a previous comment, a discussion about the quality of the isotope raw data must be addressed earlier in the text.

8-26: how the authors could explain the melt layers then if there is no signal of infiltration or connection with summer melt? Could the authors include the percentage of melt per m w.e.?

8-31: higher than the annual, monthly mean?, please specify.

9-5: "monthly d18O-T relationship was considered to reflect seasonal behaviour" based on? Please add a discussion to explain the authors' assumption.

5-9: what is the general trend? The data sets are too short as to describe or assume they represent a general trend.

9-15: please indicate the basis to the -1.4C latitudinal correction. Also indicate lapse rates used.

9-16: Indicate the significance of the trend, this is important due to the short period covered (only 5 years).

Figure 10: indicate the resolution of the temp. data.

9-20: indicate significance of the trend.

9-25: of which station? Please specify.

10-30: replace "clear" with significant or not significant.

10-3.2.3 Is there any evidence of wind redistribution of snow that could be operating at lower elevations? How is the amplitude of the seasonal d18O cycles at different elevations? Is there any sign of melt at lower elevations?

11-27: compositions

11-14: I would be cautious to extrapolate the results to the whole Peninsula region and rather specify that the result is valid for the study site. Reference to data from additional sites at the Peninsula is needed as to assure what the authors claim. The authors also need to address that the time extent of the cores prevent to robustly interpret their results into a climatic scenario. The results presented in the study are a snapshot representing and must be carefully put into a climatic context in order to avoid speculative interpretations.

11-18-20: The authors need to show evidence of similar findings elsewhere in the Peninsula as to support their claim, otherwise the claim is highly speculative when extrapolating the study results to the whole Peninsula area.

Figure 12: please correct x-axis label to "sea ice extent". Indicate the SIE data source. Indicate also the definition of the SIE index used.

12-8: it is important to address if the trend is significant or not.

12-20: "ice layers likely developed by wind ablation on wind–scouring processes at the plateau." Could the authors explain how wind could create ice layers?

I would suggest the d2H notation instead of the dD notation throughout the text to be in line with IUPAC guidelines as much as possible.

16-1: Given the accumulation rates at LCL, the low compaction at the top core meters, and the GPR data (accounting for 16 years in 41 m of snow/firn, section 3.2.4) could it be possible to have a record for the last centuries with a medium-depth ice core retrieved there? Do the authors have any preliminary age-model for the ice cap considering such high accumulations and low compaction rates?

---

## Referee Comment (RC2) · Anonymous Referee #2 · 28 Feb 2017

**Review Summary:**

This manuscript examines the relationships between stable water isotopes (D and 18O) measured in northern Antarctic Peninsula firn cores and precipitation samples and regional climate variables (SST, sea ice extent, air temperature, relative humidity) to evaluate the nature of climatic information captured in firn and snow samples from this region. Sea ice extent is found to impact local temperature and atmospheric variability, whereby extensive sea ice under negative phases of SAM cause surface inversion and a skewing of the isotope-temperature relationship in some seasons. Transitional seasons (spring and autumn) lack this oceanic forcing and the resulting stable boundary layer conditions permit deriving temperature from isotopes. Though the cores presented in this manuscript only extend back to 2008, the analysis could prove useful for interpreting longer ice cores extracted from this region.

This is an important paper because it directly evaluates the isotopic record with respect to observed atmospheric and oceanic parameters. As the observational record from this region (and Antarctica more broadly) is quite limited, analyses such as this one represent a necessary foundation before interpretation of longer firn and ice core records can be attempted. Though the work presented here will certainly be of scientific interest and the discussion and conclusion sections are interesting, I find there to be some gaps between these sections and the methods and analysis performed and presented. Overall, I believe the authors could better present the methods employed in the text and figures to better support the conclusions. Below, I have included broad and specific comments to this regard.

**General comments:**

The authors present an interesting method to date their firn cores based on measured firn-core dexcess and "dexcess meteo" estimated from probabilistic air parcel origins (from HYSPLIT) and associated gridded SST and relative humidity datasets. This seems to work out nicely, but I do have some questions about the data and methods (see below and see specific comments).

There is a need to be more explicit about what datasets were used and exactly how. I would also like to see a better justification of their inclusion over perhaps more suitable datasets. This relates to observations (e.g., why use Bellingshausen station observations over the more proximal O'Higgins?) and the gridded datasets (why use HadISST versus a higher resolution observed SST dataset, and why not use the sea ice data from this same gridded dataset as opposed to using the NSIDC sea ice extent index). Also, how sea ice extent was measured is not explained in the manuscript.

To estimate dexcess meteo from the air parcel source region, SST seems to be from a fixed region whereas relative humidity was determined based on the HYSPLIT trajectories. Is this correct? And if so, why not just pull SST time series from the same geographic area as the RH reanalysis data? The regressions between observed temperature and isotopes as presented in sections 3.1.1 and 3.1.2 are not clear. Adding these regression scatter plots to Figure 7 would help.

The regression between firn core derived dexcess and that derived from the gridded datasets (dexcess meteo) seems circular given that the gridded dataset-derived dexcess meteo was used to date the firn cores. There are multiple instances where the correlation between the core dexcess and the dexcess meteo is used to validate various parameters and interpretations of the core (including the dating), and I don't think this is supported because the firn cores were dated by peak matching with the dexcess meteo time series. If I am understanding this correctly, I believe the authors should revise the use of correlations between the two time series to support their analyses. I have documented some of these instances my comments below, but there are several other instances in the discussion that I have not mentioned.

The lack of melt in the cores is surprising given warm summers in this region and the literature cited in the introduction.

**Specific comments:**

**Page 2**

Line 4: Bromwich et al 2013 focuses on the central West Antarctic Ice Sheet air temperatures. I would suggest changing this reference to one that focuses specifically on Antarctic Peninsula air temperature trends.

Line 8: Change to "have recently lost mass"

Line 9: Most modeling studies have shown that surface melt, though accelerated in places regionally, plays little direct role in the mass balance today. Only on the northernmost AP does it impact SMB, and only indirectly via ice shelf stability forcing, does it impact mass balance elsewhere – and today this is limited to the AP.

Line 12: Change "is losing" to lost

Line 13: Change "surpasses" to surpassed

Line 14-15: Change to "This demonstrates how sensitive the coastal region of West Antarctica is to increased..."

Line 16-17: I believe Trusel et al 2012 and Kuipers Munneke et al 2012 did not report significant positive trends in AP surface melt as whole. However, Abram et al 2013 and Trusel et al 2015 (Nature Geoscience) both note positive surface melt trends on the northeast Antarctic Peninsula from an ice core and from climate models and observations, respectively. I would suggest revising this sentence.

Line 19: The first part of this sentence needs a citation indicating what studies show more wide-spread surface melt since the mid 20th century.

Line 31: Do the authors mean "tropical" ?

Line 32-33: The "southern oscillation" is not another term for the SAM, but rather ENSO. Please revise.

**Page 3**

Line 5: I would suggest citing Orr et al. 2008 (J. Climate) in reference to summer airflow over the AP owing to westerly wind increases.

Line 15: Please change to "hampers accurately determining"

Line 16: Please change to "Therefore, climate models are necessary to extend the scarce climate data both spatially and temporally."

**Page 4**

Line 8: Could you please expand upon what you mean by "improper storing"?

Line 28: Please indicate in the text how far Bellingshausen station is from the firn core sites and O'Higgins. Also, why were observations from O'Higgins not used?

Line 31-32: HadISST is actually on a 1° grid. Did you use a different version of the

data product? Also, why was HadISST chosen over a more strictly observational SST dataset (e.g., AVHRR, AMSR-E) or the 0.25° NOAA OISST v2 product? Given the cores only go back to 2008, I would think that using observations would be the best route. The use of HadISST (and the actual resolution used) should be further justified.

**Page 5**

Line 7: Did you use 1 day back trajectories or 3 day? If only 1 day as specified here, why on line 1 do you state 3 day? The methods here are a bit unclear. For example, did you calculate the RH only across the areas with >50pct parcel frequency (or some other threshold)? Also, could you reference Figure 6 here?

Line 14-16: It is unclear what sea ice metric was used to define sea ice extent "around the API". Was total Antarctic sea ice extent provided by the NSIDC Sea Ice Index used? If so, that dataset is certainly not suitable for the more regional/local analysis of this manuscript. This also raises the question of why not use the sea ice concentration data that are also part of the SST dataset (whether that is HadISST, if justified, or one of the higher resolution datasets)?

Line 18-20: This information is repetitive with the previous paragraph.

Line 24: Please change "obtained" to "derived" or "estimated"

**Page 6**

Line 6: Change "has been proved" to "has been proven" or similar. Section 2.4 more broadly: Was a constant wave velocity chosen to convert two way time to depth? Certainly the firn here is quite heterogeneous given high surface melt rates. Also, was the surface actively melting during the January fieldwork?

**Page 7**

Line 6: Change "seasons'" to "season's"

Line 10-11: Considering the "considerable differences" between daily and monthly

mean isotopic values, could you please show standard deviation error bars on your monthly mean time series in figure 7?

Line 11-16: The regression analysis presented here is quite unclear. Is the regression slope derived from 3 points each for MAM and SON? Or, are you regressing daily values? Please consider revising the text here and adding a figure to show these regressions. This would be very helpful. Also, why are only fall and spring values being regressed (or were the other seasons regressed individually, but the results were insignificant?)? Please expand on this here.

**Page 8**

Line 2: This correlation testing seems circular to me. The d excess (meteo) was used to date the ice cores by aligning the ice core d excess vs depth profiles with the d excess (meteo) vs time. So, we should clearly expect a high degree of correlation to result since these time series are already manually aligned.

Line 8: This paragraph seems better suited for the methods section.

Line 9: Please change to "allowed us to derive"

Line 13: Again, the methods need to be more clear about the time frame analyzed using the back trajectories. Here it is stated 2 days, but elsewhere it says 3 days and 1 day.

**Page 9**

Lines 1-3 / Figure 10: The stated relationships in the text are quite difficult to see on the plot. Could you plot this instead as a scatter plot, or perhaps highlight these areas on the existing line plot? Only January is labeled on the plot, with one other tick at July (?), so it's hard to understand. Please consider revising Figure 10 to improve clarity.

Line 5-7: The methods used here for extending or contracting the relationship is unclear. Please revise.

Line 15: Methods for determining a latitude temperature correction unclear. Please clarify.

Line 16-17: Figure 12b does not show the mean annual air temperature or a negative trend over time. Please revise figure citation or consider adding this information in a figure.

Line 17-20: Methods for determining temperature using the meteorological observations and sea ice extent (how was this measured?) is unclear. Please revise and consider adding a figure showing these monthly correlations.

Line 28: Please show the linear regression showing the -0.008 per mil slope on Figure 13a as opposed to the linked dots. Please also show for Figure 13b.

**Page 10**

Line 6: Figure 14a only shows accumulation through 2014.

**Page 11**

Line 11-13: Again, these datasets were aligned, so the correlation reporting is circular.

Line 15-16: This could be interesting – where did the anomalous humidity air parcel originate according to the back trajectory analysis?

**Page 12**

Line 2: I would suggest revising the use of "natural" here when referring to SAM, given the anthropogenic forcing on SAM (ozone and GHG), which is appropriately acknowledged earlier in the manuscript.

Line 17-19: I think this should be stated in reverse – that the isotopic composition is not altered by surface melt infiltration and percolation.

Line 20-21: I find the lack of ice layers in the firn cores due to surface melt refreezing to be unexpected. The mean monthly temperature at O'Higgins is often at or above 0°C

during summer months. Do the lower elevation cores (300-600 m elevation) not have significant melt? And are smaller ice layers not present at the plateau cores? Even at -4-7°C mean monthly temperature (assumed using lapse rate for the 1100m plateau), I would expect melt each year. These layers may not coincide with summer seasons based on your age-depth scales, which is common due to melt percolation into deeper layers.

**Page 13**

Line 24: Should this say "altitude"?

**Page 15**

Line 1: Should this say "manually"?

Line 8: Should this say between 2008 and 2014? I don't believe any data before 2008 are presented.

Line 9: Change to "proxies".

**Page 16**

Line 3: I look forward to seeing longer records from this area!

**Figures**

Figure 1

It would be helpful to have an inset of the Antarcitc Peninsula (or perhaps even just the northern Antarctic Peninsula). In particular, having Bellingshausen station located on this map would be helpful given that some of the meteorological data analyzed are from this site. Also, a box showing the area where SST data were extracted would be helpful.

Figure 7

The small dots plotted differ from the legends and caption. Notably the "small orange

dots" in (b) appear the same as the small dots in (a). Please revise this figure (also see comment from page 7 about adding error bars).

---

## Author Comment (AC1) · 20 Apr 2017

Major comments

OBS1. Title I found the title misleading since the firn cores temporal extent is too short as to infer climatic signals from them. A suggested title would be: Water stable isotope and deuterium excess records from precipitation and firn cores from northern Antarctic Peninsula as tools for climate studies in the region.

ANS.1. The reviewer has a valid point here, as the word "climatic" could mislead the readers. Instead we propose the following title: New regional insights into the stable water isotope signal at the northern Antarctic Peninsula as tools for climate studies

OBS2. Database I am concerned about the author's use of the Tair and altitudinal T

profiles from BE station. Why the authors would expect this temperature to be representative for the core sites? Later in section 2.3 the authors mention that daily mean temperatures are available at OH, then why not to use that data set instead of the BE-data? Please elaborate and justify the use of the BE data set.

ANS.2. The BE dataset was used instead of OH, since the latest has numerous gaps of validated data (see: https://legacy.bas.ac.uk/met/READER/surface/O_Higgins.All.temperature.html). This is especially important for the most recent years. Some years (e.g.: 2015) even with 50% of the monthly data validated under 80% of daily temperature records. On the other hand, BE has an uninterrupted validated record from 1968 to the present. Both data sets (OH-BE) correlate (for validated months) with a r-value higher than 97% (p<0.01). This correlation is even higher than the correlation between the OH Station data and Esperanza (ESP) Station (96%), which is located less than 100 km away. However, ESP is located at the east coast of the Antarctic Peninsula and therefore, is partially influenced by continental conditions. The Fig.1 attached to this comment shows the linear regression and correlation for the OH-BE and OH-ESP datasets.

OBS3. Stable isotope time series analysis In section 2.3, page 5-25, the authors state that the d-excess signal obtained from the firn cores was depicted against depth and filtered using IFFT to be compared with d-excessmeteo, but the authors never showed the original raw core isotope data, nor discussed the quality of it, e.g. amplitude of the signal, seasonality, possible melt, signal differences /similitude at the different core sites, etc. The authors must describe the raw isotope data before applying any further statistical method to compare it with either instrumental or modelled data. Before assessing the quality and representativeness of the raw data it is not possible to carry on with other comparisons. Further, the authors obtained the time scale of core d-excess based on the "the strong similarities between both signals (d-excess and d-excess meteo)"; even thought this method is quite interesting and innovative, and both profiles clearly agree (Fig. 3), the authors must indicate why a more traditional dating was not

preformed, e.g., annual layer counting, which would be the obvious first attempt on dating the cores. The authors must address this issue and justify the dating method used in order to show its value over more traditional approaches. The authors must also indicate the error in the time scale obtained. This is shortly introduced in the text in section 4.2 but it should be mentioned already in section 2.3.

ANS.3. The reviewer is right by addressing this comment, we certainly did not discuss properly the quality either the representativeness of the RAW data. We will add a new paragraph at the beginning of section 2.3, which will include a general description of the RAW data, a discussion about the quality and representativeness and we will also refer to the issues that we had attempting to carry out a traditional dating method. Furthermore, as it is suggested, we will highlight the value of the new method proposed over other more traditional approaches. In relation to the displaying the RAW data, this is available from the assets to this publication: https://doi.pangaea.de/10.1594/PANGAEA.871083 https://doi.pangaea.de/10.1594/PANGAEA.871080 Figure 3 already includes this data, nevertheless, the color that we have chosen was probably not the best one. We will change the color in this figure in order to represent the RAW data in a better way. Since our dating was carried out using a signal-tuning method, the error associated with this method is usually dependent on the error that the reference signal has and also to the differences between the two signals that are compared. In our context, as both signals should be dependent on almost the same variables which have seasonal behaviors, the error associated with the signal matching can be estimated as to be of +/- 1 month.

OBS.4. Seasonality The authors define seasons and sample sets to each season. However, the authors have a limited number of years (2008-2014) as to construct a representative seasonal signal of d18O in precipitation at the study site to use as a baseline to compare with particular monthly means of a given year. The authors use 1 month of a particular year to describe a seasonal signal without discussing its representativeness. Therefore, the results must be explained as results for a particular year

rather than as to a "seasonality", mentioning the results limitations.

ANS.4. This comment is right in relation to the seasonal long-term approach that we are not able to achieve with our limited dataset. A paragraph will be added in section 4.1 specifying the limitations and representativeness of our dataset. We will highlight the fact that the information presented in this study is intending to give a rough idea about changes identified in the isotopic signal throughout the year in a short time span, rather than creating a regional baseline.

OBS.5. Glaciological setting The authors must provide a more detailed description of the study site, e.g. glaciological setting, meteorology of the study site, earlier studies of in the OH area, if available. This could be included in an additional section as "study site".

ANS.5.The authors believe that a new section is not absolutely needed. The introduction already describes most available (rather scarce) glaciological information for the study region. Others like meteorological and geographical references are widely discussed along this manuscript and in a previous work of this group (Fernandoy et al., 2012), also published in the journal. Nonetheless, we will add more details to the last part of the introduction section.

Minor comments

1-23. Be more specific when given the results. The results presented here are a snapshot of a region situated at Antarctic Peninsula but they don't necessarily reflect the whole Antarctic Peninsula situation. Unless a geographical significance study is done, please clearly state to which region of the Antarctic Peninsula are your results representative.

ANS.1-23. This was corrected along the whole manuscript to emphasize that our discussion is valid for the most northern portion of the Antarctic Peninsula, i.e.: Study region close to the Laclavere Plateau and nearby west flank.

2-2. I have seen Antarctic Peninsula being referred as AP or APIS (when talking about Antarctic Peninsula Ice sheet), I wonder why the authors chose API.

ANS.2-2. The abbreviation API was taken for simplification reason from the sample codes analyzed here and used also on our previous publication. In order to keep a consistency with other authors from the region, we changed the abbreviation API to AP for this manuscript.

2-24. at several stations

ANS.2-24. Corrected as suggested

2-27. with a marked warming

ANS.2-27. Corrected as suggested

3-22. the authors only mention the "high temporal resolution" of their records but did not talk about the temporal extent of their records. Since the title of the manuscript involves climate, the temporal extent of the firn cores is as important as the resolution and should be introduced together.

ANS.3-22. Corrected as suggested

4-3. austral summer campaigns

ANS.4-3. Corrected as suggested

4-3: please remove "several shallow–depth firn cores (totalling more than 60 m) were retrieved from the northern part of the API". There is no need to add this vague information if the authors are going to give more details of the cores in the next sentences.

ANS.4-3. Corrected as suggested

4-4: Add link to Figure 1. Label O'Higgins station as "OH", also add info on image source and contours details.

ANS.4-4. Corrected as suggested

4-8: Please be more specific about how many samples were discarded and why.

ANS.4-8. Additional information was added to this section. The samples were discarded using a statistical outlier test (modified Thompson tau technique) (Thompson, 1985). Using this criterion all samples with d-excess values lower than -9‰ were discarded, as they lie outside the normal distribution of regular samples.

4-10/13. ...profile of density and physical properties of the ice

ANS.4-10/13. Corrected as suggested

10-13: add info of the drill, handling, storage, and sampling for all cores. This can be summarized in a table.

ANS.4-10/13. This information is included in section 2.1, however additional antecedents on the handling of water samples will be included in the manuscript.

4-16. To which institution/facility in Viña del Mar? Please add information about sample melting and storage during melting (type of vial). As mentioned by the authors in page 20-21, secondary processes during storage and transport (and also melting of the sample) can perturb the isotopic signal.

ANS.4-16. As for the previous observation additional information of the protocol followed both in Chile and Germany will be included in section 2.1. This basically consist in melting the snow and firn samples at controlled conditions (4°C) in sealed bags overnight. At the next morning previous to running the isotope analysis, each sample was agitated for homogenization.

4-18. Indicate where the water stable isotopes were analysed (instrument, method, etc) and the accuracy for all the cores. This is depicted in Table 1 but it is not clearly stated in the text. Cite references to values shown in Table 1.

ANS.4-18. As before, additional information was added and table 1 referenced as suggested.

4-18. Please indicate where the firn and precipitation samples were collected at OH.

ANS.4-18. This information was added to 4-7

OBS. In Fig. 2, show source of the data. Please add the BE station in Figure 1.

ANS. The source of the data (LIMA-Landsat Image Mosaic of Antarctica) was added to the figure. Station BE (Bellingshausen) is located on the South Shetland Island, therefore out of the scope of this image. An insert to this figure will be added.

5-1: Please indicate the arrival point of the HYSPLIT trajectories. Was 1-backtrajectory estimated or a cluster of backtrajectories?

ANS.5-1. This information was added, corresponding to OH coodinates at an isobaric arrival point of 850 hPa (around 1500 m a.s.l.). The backward trajectory analysis corresponds to a cluster analysis and not single trajectories.

5-22. "for the whole region". The author's refer to the whole Antarctic Peninsula? If, yes, the authors might reconsider limit the interpretation to the nearby study area. It is unclear to me if the HYSPLIT backtrajectories were set to end at OH or to other sites in the Peninsula. Please clarify this.

ANS.5-22. This was corrected to express that our discussion correspond to the nearby region of the OH Station and Laclavere Plateau, and doesn't mean to extend this conclusions to the whole Antarctic Peninsula region, which clearly exceed the scope of this work.

Figure 3: it is not clear to me if cores OH-10, OH-9, and OH-6 were drilled at the exact point, how close were there drilled? This info cannot be inferred from Table 1 which shows the exact coordinates for all three cores. The core sites could be shown as a zoomed-in section in Fig. 1.

ANS. The location was intentionally selected at the same position for the cores OH-6, OH-9 and OH-10, but retrieved in different years. This will be explained on the revised

version of this manuscript.

6-25,26: this should be stated earlier in the text, e.g. in section 2.3 when figure 3 was shown.

ANS.6-25,26. This paragraph was moved as suggested.

6-27: Remove OH

ANS.6-27. Removed as suggested.

6-28: Please mention how many samples were rejected and why. This is mentioned earlier in the text but is not discussed.

ANS.6-28. This information was added to section 2.1. A total of 13 precipitation samples were discarded.

Figure 6: please add labels for the study site, and Bellingshausen Sea in the map.

ANS. Labels were added to the figure

7-6: please mention how the authors defined the seasons, e.g. DJF-! summer. Also, please explain better how you selected the set of samples representative of each season; as it is written in the text, it appears that the set for each season was selected upon the number of samples of arbitrary months which might cause bias, especially in section 3.1.1 where the seasonal regression slopes are discussed. Please consider using all samples available for each season or limit your discussion to the represented months but not to seasonal scales. Please discuss the annual precipitation distribution at the study site if available and put your results in that context. Also discuss how precipitation samples were taken, is a precipitation event identified as one precipitation sample or are the samples taken and identified on a daily (hourly) basis? Indicate in a figure the number of samples per month and also the volume per sample/event.

ANS.7-6. Done as suggested. Regarding the sample selection, we extended the explanation of our selection criteria. Related to precipitation sample acquisition, there was

already an explanation in the text referring to the daily scheme in which precipitations were measured. Figure 7 shows the precipitation sample distribution.

7-9: Please discuss how outliers were removed in section 3.1.1 (when first referred).

ANS.7-9. Done as suggested

7-11: please include the error and significance of the regression slopes.

ANS.7-11. Done as suggested

7-13: the authors must justify why they believe a particular month is representative of a season. If the authors have data for all months, why to assume only one month as representative of a season? This issue has been addressed in a previous comment (7-6). This is very important to clarify as the authors are attempting to link their results to climatic features.

ANS. 7-13. Answered in 7-6

7-16: An inverse behaviour between July and June or between July-June compared to MAM and SON? Please explain.

ANS.7-16. An specification was added to the text

7-22: do not use "weak", use instead: correlation coefficients for these comparisons are not significant (indicate level of confidence).

ANS.7-22. Done as suggested. Also the value of the rejected samples was wrongly stated in this section ("lower -2‰'), as showed before, the statistical outlier test points out to eliminate all samples deviating more than $2\sigma$ from the mean value (i.e.: <-9‰ and >12‰.

7:3.1.3 Please indicate geographical sites in a figure (in Fig. 6 for example).

ANS. Done as suggested

Figure 8 can be removed and the equation of the regression line can be given in section

3.1.3.

ANS. Figure 8. The authors consider that this figure is needed to give a better idea of the correlation for different values of d-excess. However, we do recognize that this figure is not key for the paper. We would still propose to keep this figure.

8-5: define GMWL and LMWL in the text (now is only defined in Fig. 9)

ANS. 8-5. This was done as suggested.

8-6: previously you defined the slope as "s", now it is mean slope as "m" but not defined, either use s again or define mean slope as "m".

ANS. 8-5. This was corrected throughout the paper to keep consistency using the "s" for slope.

8-9: Please remove the first sentence as it is unnecessary.

ANS. 8-9. We considered that this sentence works as a short introduction for the following section. We think it should be kept in this sense.

8-10: as mentioned in a previous comment, a discussion about the quality of the isotope raw data must be addressed earlier in the text.

ANS 8-10. We recognized this point and added a discussion on section 2.3

8-26: how the authors could explain the melt layers then if there is no signal of infiltration or connection with summer melt? Could the authors include the percentage of melt per m w.e.?

ANS. 8-26. This issue will be discussed at the end of section 4.1

8-31: higher than the annual, monthly mean?, please specify.

ANS.8-31. We mean monthly means, this was corrected in this section.

9-5: "monthly d18O-T relationship was considered to reflect seasonal behaviour" based

on? Please add a discussion to explain the authors' assumption.

ANS. 9-5. This phrase was eliminated from the text, as it was unnecessary and could lead to confusion of the following text.

5-9: what is the general trend? The data sets are too short as to describe or assume they represent a general trend.

ANS. 9-5. This was revised and corrected in the text

9-15: please indicate the basis to the -1.4C latitudinal correction. Also indicate lapse rates used.

ANS. 9-15. The correction procedure was now added to this section, the basis of doing so is previously exposed on the major comment 2.

9-16: Indicate the significance of the trend, this is important due to the short period covered (only 5 years).

ANS. 9.16. This information was added to the revised version of the text

Figure 10: indicate the resolution of the temp. data.

ANS. Added to the figure "monthly resolution"

9-20: indicate significance of the trend.

ANS. 9-20. Done as suggested

9-25: of which station? Please specify.

ANS. 9-20. OH (BE) station was added to this sentence.

10-30: replace "clear" with significant or not significant.

ANS.10-30. This sentence was modified, as it doesn't mean to point out an statistical trend, but rather a tendency of increase in density of the firn pack.

10-3.2.3 Is there any evidence of wind redistribution of snow that could be operating at lower elevations? How is the amplitude of the seasonal d18O cycles at different elevations? Is there any sign of melt at lower elevations?

ANS. 10-3.2.3. This matter was not discussed here in extent, since was already exposed by Fernandoy et al. (2012). A noticeable effect of melting is present bellow 700 m a.s.l., that it's location of OH-4 and bellow. Wind redistribution will play a role in some geographical singularities like depression or valley-like features. All cores were retrieved from geographical height in order to minimize this effect (Fernandoy et al., 2012). Nonetheless, these redistribution effects are much less important against the high accumulation rates for this region.

11-27: compositions

ANS. 11-27. Corrected in the text.

11-14: I would be cautious to extrapolate the results to the whole Peninsula region and rather specify that the result is valid for the study site. Reference to data from additional sites at the Peninsula is needed as to assure what the authors claim. The authors also need to address that the time extent of the cores prevent to robustly interpret their results into a climatic scenario. The results presented in the study are a snapshot representing and must be carefully put into a climatic context in order to avoid speculative interpretations.

ANS. 11.14. This observation was taken into account along with other previous similar observation referring the regional vs. local extension of our results. We will revise the text to make it clear, that our investigation shows the situation for a specific portion of the Antarctic Peninsula (i.e.: northern AP) and for a restricted time frame (2008-2015).

11-18-20: The authors need to show evidence of similar findings elsewhere in the Peninsula as to support their claim, otherwise the claim is highly speculative when extrapolating the study results to the whole Peninsula area.

ANS. 11-18-20. We will restrict our discussion concretely to our study region, since similar study are non-existent or scarce to rest of the Peninsula.

Figure 12: please correct x-axis label to "sea ice extent". Indicate the SIE data source. Indicate also the definition of the SIE index used.

ANS. 12. The axis of figure was modified accordingly, and the source and definition of SIE was added. SIE data was obtained from the NSIDC Sea ice index (https://nsidc.org/data/seaice_index/) and the definition of the sea ice extent is according to . Defining the edge of the sea ice, as the portion of sea surface covered by at least 15% of ice.

12-8: it is important to address if the trend is significant or not.

ANS. 12-8. As in previous observation, we don't mean to express a statistical trend of the isotope values, but rather a tendency of depleted values with higher altitudes. This will be corrected on the text.

12-20: "ice layers likely developed by wind ablation on wind–scouring processes at the plateau." Could the authors explain how wind could create ice layers?

ANS. 12-20. In this line we expressed our self not right. We actually don't mean to state that a proper ice layer could form purely from wind in a high accumulation region like the AP. We refer here to actual glazed (ice) "crust", only a few mm wide as shown by the stratigraphy of the firn cores. This wind crust could form a thin glazed surface due to sublimation and snow drift abrasion, and in some opportunities by solidification of super-cooled droplets flowing against ground surface irregularities (sastrugi-like). During our field work, we witness these processes during different years.  

References

Fernandoy, F., Meyer, H., and Tonelli, M.: Stable water isotopes of precipitation and firn cores from the northern Antarctic Peninsula region as a proxy for climate reconstruction, The Cryosphere, 6, 313-330, doi: 10.5194/tc-6-313-2012, 2012. Thompson,

[Figure]

R.: A Note on Restricted Maximum Likelihood Estimation with an Alternative Outlier Model, Journal of the Royal Statistical Society. Series B (Methodological), 47, 53-55, 1985.

[Figure]

The scatter plot shows two datasets with linear fits:

$OH_{TEMP} = -0.33 + 0.68 * ESP_{TEMP}$
$r = 0.96$

$OH_{TEMP} = -1.26 + 1.06 * BELL_{TEMP}$
$r = 0.97$

Legend:
- OH-BE$_{TEMP}$ (green)
- OH-ESP$_{TEMP}$ (blue)

Y-axis: Temperature (°C) - OH
X-axis: Temperature (°C)

**Fig. 1.** OH-BE (green dots) and OH-ESP (blue dots) monthly mean air temperature correlation between 1968 and 2015

---

## Author Comment (AC2) · 20 Apr 2017

Major comments

OBS.1. There is a need to be more explicit about what datasets were used and exactly how. I would also like to see a better justification of their inclusion over perhaps more suitable datasets. This relates to observations (e.g., why use Bellingshausen station observations over the more proximal O'Higgins?) and the gridded datasets (why use HadISST versus a higher resolution observed SST dataset, and why not use the sea ice data from this same gridded dataset as opposed to using the NSIDC sea ice extent index). Also, how sea ice extent was measured is not explained in the manuscript.

ANS. 1. We added to the manuscript an explanation, emphasizing which datasets were selected and how they were considered in this work. In relation to choosing Bellingshausen (BE) station over O'Higgins (OH) station, BE dataset was used instead of OH, since the latest has numerous gaps of validated data. (see: https://legacy.bas.ac.uk/met/READER/surface/O_Higgins.All.temperature.html). This is especially important for the most recent years. Some years (e.g.: 2015) even with 50% of the monthly data validated under 80% of daily temperature records. On the other hand, BE has an uninterrupted validated record from 1968 to the present. Both data sets (OH-BE) correlate (for validated months) with a r-value higher than 97% (p<0.01). This correlation is even higher than the correlation between the OH Station data and Esperanza (ESP) Station (96%), which is located less than 100 km away. However, ESP is located at the east coast of the Antarctic Peninsula and therefore, is partially influenced by continental conditions. The figure attached to this comment shows the linear regression and correlation for the OH-BE and OH-ESP datasets. In relation to the gridded datasets, we have corrected the text since we unintentionally made a mistake writing HadISST, instead of HadSST. For Sea Ice extension we did not use HadISST since one of its key limitations is that higher resolution and more homogenous data are available for the modern satellite period, 1979-present. Instead of HadISST, we used NSIDC because it provides a higher resolution grid (25km x 25km). The reviewer is right by noticing that the way in which sea ice extension was not detailed in the text. To measure the sea ice extent we considered as a starting point the location of O'Higgins station and as an end point, the sea ice front in the direction towards King George Island. This procedure will be added to the manuscript.

OBS.2. To estimate dexcess meteo from the air parcel source region, SST seems to be from a fixed region whereas relative humidity was determined based on the HYSPLIT trajectories. Is this correct? And if so, why not just pull SST time series from the same geographic area as the RH reanalysis data? The regressions between observed temperature and isotopes as presented in sections 3.1.1 and 3.1.2 are not clear. Adding these regression scatter plots to Figure 7 would help.

ANS. 2. The SST datasets were obtained from a fixed region based on two main reasons. The first one, because the continuity of the data in this quadrant during the time that we cover is more consistent than the neighbors, which in turn are rather limited and containing important gaps. The second reason is because almost all the air parcels that reached Laclavere Plateau, in the time interval studied, crossed through this region during a significant amount of time (one day or more). Thus, it might has likely exerted an imprint over the moisture parcel crossing this area. In contrast, out of this region, the pathways followed by the air parcels spread into different directions, characterized by data gaps and sometimes inconsistencies. Even though the SST data considered is originated from a single region, considering a rough value of the behavior of SST at this latitude is enough to estimate dexcess meteo. As we show in the text, this parameter is less dependent of SST values (mostly influenced by humidity). Therefore, dexcess meteo is highly dependent of relative humidity. The 3-day backward trajectories provided data along the whole path, those data sets were considered to give representativeness to dexcess meteo estimation. Linear regression were not added to the figure, but were discussed now in the text in section 3.1.1 and 3.2.2.

OBS.3. The regression between firn core derived dexcess and that derived from the gridded datasets (dexcess meteo) seems circular given that the gridded dataset-derived dexcess meteo was used to date the firn cores. There are multiple instances where the correlation between the core dexcess and the dexcess meteo is used to validate various parameters and interpretations of the core (including the dating), and I don't think this is supported because the firn cores were dated by peak matching with the dexcess meteo time series. If I am understanding this correctly, I believe the authors should revise the use of correlations between the two time series to support their analyses. I have documented some of these instances my comments below, but there are several other instances in the discussion that I have not mentioned.

ANS. 3. The reviewer is right by addressing this comment, we committed an unintentional error in the caption of Figure 8. As it was written at the end of in section 3.1.3 "From the relationship between monthly mean values of dexcess from precipitation samples and dexcess meteo constructed from meteorological parameters (rh and SST) of the high density precipitation pathways, a correlation coefficient of R= 0.86 (p<0.01) was obtained (Fig. 8)", the regression reflects the relation between dexcess from precipitations and dexcess meteo. The relation between both parameters is what we consider that validates the dating procedure, as precipitation dexcess is highly related with the constructed dexcess meteo. We will correct the figure's caption accordingly.

OBS.4. The lack of melt in the cores is surprising given warm summers in this region and the literature cited in the introduction.

ANS.4. We haven't neglected the effect of melting, however melt events are rather insignificant against the very high accumulation. During none of the field works (January – February) we have witnessed any major melt event at the highest point of the northern Antarctic Peninsula (i.e.: Laclavere Plateau). The stratigraphic profiles of the firn cores retrieved show different kind of ice layer/crust. We have attributed this to different phenomena like wind sublimation, precipitation of super-cooled humidity and in some cases possible melt layers. Nonetheless, we don't see any possible seasonality in the distribution of the ice layer and crusts.

Specific comments:

Line 4: Bromwich et al 2013 focuses on the central West Antarctic Ice Sheet air temperatures. I would suggest changing this reference to one that focuses specifically on Antarctic Peninsula air temperature trends.

ANS. Changed this citation to Carrasco (2013), and also indicated that this refers to the Antarctic Peninsula specifically.

Line 8: Change to "have recently lost mass"

ANS. Done as suggested

Line 9: Most modeling studies have shown that surface melt, though accelerated in places regionally, plays little direct role in the mass balance today. Only on the northernmost AP does it impact SMB, and only indirectly via ice shelf stability forcing, does it impact mass balance elsewhere – and today this is limited to the AP.

ANS. Added to the text that this refers mostly to AP

Line 12: Change "is losing" to lost

ANS. Done as suggested.

Line 13: Change "surpasses" to surpassed

ANS. Done as suggested

Line 14-15: Change to "This demonstrates how sensitive the coastal region of West Antarctica is to increased. . ."

ANS. Done as suggested

Line 16-17: I believe Trusel et al 2012 and Kuipers Munneke et al 2012 did not report significant positive trends in AP surface melt as whole. However, Abram et al 2013 and Trusel et al 2015 (Nature Geoscience) both note positive surface melt trends on the northeast Antarctic Peninsula from an ice core and from climate models and observations, respectively. I would suggest revising this sentence.

ANS. The lines were rephrased to clarify these points, as the reviewer make a valid point suggesting so.

Line 19: The first part of this sentence needs a citation indicating what studies show more wide-spread surface melt since the mid 20th century.

ANS. This was included and corrected according to the previous observation.

Line 31: Do the authors mean "tropical" ?

ANS. We refer to regular (pre-industrial period) patterns. We will include this information to this line.

Line 32-33: The "southern oscillation" is not another term for the SAM, but rather ENSO. Please revise.

ANS. This is correct. Southern Oscillation is misplaced; we will remove this term from cited line. Page 3

Line 5: I would suggest citing Orr et al. 2008 (J. Climate) in reference to summer airflow over the AP owing to westerly wind increases.

ANS. Two cites were included to this line showing the impact of summer airflow: (Orr et al., 2008; van Lipzig et al., 2008)

Line 15: Please change to "hampers accurately determining"

ANS. Done as suggested

Line 16: Please change to "Therefore, climate models are necessary to extend the scarce climate data both spatially and temporally."

ANS. Done as suggested

Line 8: Could you please expand upon what you mean by "improper storing"?

ANS. This was also brought up to our attention by referee 1 and clarified in the text

Line 28: Please indicate in the text how far Bellingshausen station is from the firn core sites and O'Higgins. Also, why were observations from O'Higgins not used?

ANS. Same as before. Referee 1 asked about this in his comments. We responded showing that OH data has long non-validated data periods. On the other hand, BE has an uninterrupted record since 1968. Although BE is located around 150 km NE from OH station, the correlation between both records (for valid period) is higher than 0.97 with a high statistical significance (p<0.01), which is higher than other station like

Esperanza (ESP) (r= 0.96) located less than 100km from OH, but on the east side of the AP. Linear regressions show that OH and BE data has a slope very close to 1, while OH-ESP is lower (s = 0.33) (with lower r). This probably reflects some influence from continental conditions. Therefore, we used BE to complement the OH data. Figure attached here.

Line 31-32: HadISST is actually on a 1 grid. Did you use a different version of the data product? Also, why was HadISST chosen over a more strictly observational SST dataset (e.g., AVHRR, AMSR-E) or the 0.25 NOAA OISST v2 product? Given the cores only go back to 2008, I would think that using observations would be the best route. The use of HadISST (and the actual resolution used) should be further justified.

ANS. This was explained before. Please see answer to Major comment 1.

Line 7: Did you use 1 day back trajectories or 3 day? If only 1 day as specified here, why on line 1 do you state 3 day? The methods here are a bit unclear. For example, did you calculate the RH only across the areas with >50pct parcel frequency (or some other threshold)? Also, could you reference Figure 6 here?

ANS. We first used 3 day backward trajectories to figure out the provenance and distribution of air parcels that reached the study site. As we noticed that there was a high density pathway in the region, we explored the conditions that prevail in the near surroundings (limited by 1 day backward trajectories). After analyzing both datasets, we determined that the area covered by 2 day backward trajectories had a high representativeness of the maritime region that surrounds Laclavere Plateau. This area is representative because geographically includes the region affected by westerly winds and sea ice front during winter time, both factors that exerts high influence on the air parcels that approach this region. Line 14-16: It is unclear what sea ice metric was used to define sea ice extent "around the API". Was total Antarctic sea ice extent provided by the NSIDC Sea Ice Index used? If so, that dataset is certainly not suitable

for the more regional/local analysis of this manuscript. This also raises the question of why not use the sea ice concentration data that are also part of the SST dataset (whether that is HadISST, if justified, or one of the higher resolution datasets)?

Line 18-20: This information is repetitive with the previous paragraph.

ANS. Modified in the text

Line 24: Please change "obtained" to "derived" or "estimated"

ANS. Done as suggested

Line 6: Change "has been proved" to "has been proven" or similar. Section 2.4 more broadly: Was a constant wave velocity chosen to convert two way time to depth? Certainly the firn here is quite heterogeneous given high surface melt rates. Also, was the surface actively melting during the January fieldwork?

ANS. Proved was replaced. Regarding section 2.4: Radar results were revised and re-analyzed. We now used a 2D model velocity based on the density profile obtained from the local cores for the upper snowpack and a density model for deeper snow/firn. During January, we were in the field for about 10 days. We experienced days of strong snow precipitation, wind drift and also sunny days, but we didn't see evidences of surface melting in this short period.

Line 6: Change "seasons'" to "season's"

ANS. Done as suggested

Line 10-11: Considering the "considerable differences" between daily and monthly mean isotopic values, could you please show standard deviation error bars on your monthly mean time series in figure 7?

ANS. Explained on major OBS. 2 this discussion was addressed in the text in section 3.1.1 Line 11-16: The regression analysis presented here is quite unclear. Is the regression slope derived from 3 points each for MAM and SON? Or, are you regressing daily values? Please consider revising the text here and adding a figure to show these regressions. This would be very helpful. Also, why are only fall and spring values being regressed (or were the other seasons regressed individually, but the results were insignificant?)? Please expand on this here.

ANS. The regression slopes presented were derived from d18O daily values from precipitation samples and mean air temperatures from those days. We will add a paragraph in the manuscript which will include the slope from other seasons and the standard error and significance of each slope. Additionally, Figure 7 will be modified to include an histogram to represent how precipitation samples are distributed in time , seasonal regressions will be discussed in the text in section 3.1.1. and 3.2.2.

Line 2: This correlation testing seems circular to me. The d excess (meteo) was used to date the ice cores by aligning the ice core d excess vs depth profiles with the d excess (meteo) vs time. So, we should clearly expect a high degree of correlation to result since these time series are already manually aligned

ANS. As explained on comment 3 this is not what we intended. Please see explanation above.

Line 8: This paragraph seems better suited for the methods section.

ANS. We consider that this section should be kept here, it is through that some part of the methodology is revised here, but still showing results of combined geophysics and stable water isotope information.

Line 9: Please change to "allowed us to derive"

ANS. Done as suggested

Line 13: Again, the methods need to be more clear about the time frame analyzed using the back trajectories. Here it is stated 2 days, but elsewhere it says 3 days and 1 day.

ANS. The answer related to this subject is written in the comment of Page 5 Line 7 Page 9

Lines 1-3 / Figure 10: The stated relationships in the text are quite difficult to see on the plot. Could you plot this instead as a scatter plot, or perhaps highlight these areas on the existing line plot? Only January is labeled on the plot, with one other tick at July (?), so it's hard to understand. Please consider revising Figure 10 to improve clarity.

ANS. The figure was modified showing now the period of positive/inverse relationship and the time (x) axis better labeled.

Line 5-7: The methods used here for extending or contracting the relationship is unclear. Please revise.

ANS. This paragraph intends to refer to the fact that at this latitudes, calendar seasons do not play a significant role as climatic seasonality extends beyond calendar time limits. Rather than using calendar seasons we turned to define three seasons: winter (JJAS), summer (DJF) and a transitional season (MAM-ON). There paragraph will be corrected in the text in order to better address this issue.

Line 15: Methods for determining a latitude temperature correction unclear. Please clarify.

ANS. Done, new information is in the text now.

Line 16-17: Figure 12b does not show the mean annual air temperature or a negative trend over time. Please revise figure citation or consider adding this information in a figure.

ANS. Figure citation was revised as suggested.

Line 17-20: Methods for determining temperature using the meteorological observations and sea ice extent (how was this measured?) is unclear. Please revise and consider adding a figure showing these monthly correlations.

ANS. To clarify this observation, we redefined our equation to take into account the seasonal behavior in these region. We separated our equation in two branches which are dependent on the presence of sea ice. In moths with presence of sea ice in this region (May-June-July-August-September) the temperature was obtained by Tlcl= (Tbe-1.4) + 1,13(Mmonth*SIE+Nmonth), where Mmonth and Nmonth are the slope and interception of the linear relation between lapse rates and SIE presented in Figure 12. On the other hand, during months without the presence of sea ice, the temperature over the Plateau was obtained by the expression Tlcl= (Tbe-1.4) + 1,13(H(x)), where H(x) is the monthly mean lapse rate value of a given month "x" measured in Bellingshausen Station between 1978-1996. This paragraph will be corrected in the manuscript

Line 28: Please show the linear regression showing the -0.008 per mil slope on Figure 13a as opposed to the linked dots. Please also show for Figure 13b.

ANS. Figures were revised and changed as suggested.

Line 6: Figure 14a only shows accumulation through 2014.

ANS. The most recent core analyzed goes to January 2015, therefore is not possible to have the accumulation for 2015. New cores from 2016 are yet to be analyzed will give more information about accumulation.

Line 11-13: Again, these datasets were aligned, so the correlation reporting is circular.

ANS. As explained on comment 3, please see explanation. After taking into account our correction, the text makes sense since both parameters are independent.

Line 15-16: This could be interesting – where did the anomalous humidity air parcel originate according to the back trajectory analysis?

ANS. These trajectories are coming mostly form eastern coast of the Antarctic Peninsula and the continent, therefore reflecting depleted isotope values.

Line 2: I would suggest revising the use of "natural" here when referring to SAM, given the anthropogenic forcing on SAM (ozone and GHG), which is appropriately acknowledged earlier in the manuscript.

ANS. To avoid confusion the word "natural" was eliminated and the phrase changed to "seasonal oscillations"

Line 17-19: I think this should be stated in reverse – that the isotopic composition is not altered by surface melt infiltration and percolation.

ANS. Corrected as suggested.

Line 20-21: I find the lack of ice layers in the firn cores due to surface melt refreezing to be unexpected. The mean monthly temperature at O'Higgins is often at or above 0°C during summer months. Do the lower elevation cores (300-600 m elevation) not have significant melt? And are smaller ice layers not present at the plateau cores? Even at -4-7°C mean monthly temperature (assumed using lapse rate for the 1100m plateau), I would expect melt each year. These layers may not coincide with summer seasons based on your age-depth scales, which is common due to melt percolation into deeper layers.

ANS. Certainly lower areas show clear sign of melting, this was widely discussed on our previous work by Fernandoy et al. (2012). We identified that melting is strong even, for the high accumulation of this region, at altitudes lower than 700 m a.s.l. As addressed before, during all of our field season (all of them between January and February), we haven't witnessed important melting events. Although we have seen the freezing of

super-cooled droplets in the irregular surface (sastrugi-like) and glazed surface due to ablation by strong wind, accompanied by snow drift.

Line 24: Should this say "altitude"?

ANS. We actually do mean latitude, referring to west flank from the AP divisory from Laclavere.

Line 1: Should this say "manually"?

ANS. To avoid confusion we will delete the word mainly.

Line 8: Should this say between 2008 and 2014? I don't believe any data before 2008 are presented.

ANS. Typeset error. Thanks.

Line 9: Change to "proxies".

ANS. Done as suggested

Line 3: I look forward to seeing longer records from this area!

ANS. We expect to continue the work in this area, further campaigns are planned for 2017/18. Figures

Figure 1 It would be helpful to have an inset of the Antarcitc Peninsula (or perhaps even just the northern Antarctic Peninsula). In particular, having Bellingshausen station located on this map would be helpful given that some of the meteorological data analyzed are from this site. Also, a box showing the area where SST data were extracted would be helpful.

ANS. A second panel was added to this figure to show the location of both Station and LCL. Figure 7

The small dots plotted differ from the legends and caption. Notably the "small orange dots" in (b) appear the same as the small dots in (a). Please revise this figure (also see comment from page 7 about adding error bars).

ANS. Revised. The figure was improved accordingly.

References

Carrasco, J. F.: Decadal Changes in the Near-Surface Air Temperature in the Western Side of the Antarctic Peninsula, Atmospheric and Climate Sciences, 03, 7, doi: 10.4236/acs.2013.33029, 2013. Fernandoy, F., Meyer, H., and Tonelli, M.: Stable water isotopes of precipitation and firn cores from the northern Antarctic Peninsula region as a proxy for climate reconstruction, The Cryosphere, 6, 313-330, doi: 10.5194/tc-6-313-2012, 2012. Orr, A., Marshall, G. J., Hunt, J. C. R., Sommeria, J., Wang, C.-G., van Lipzig, N. P. M., Cresswell, D., and King, J. C.: Characteristics of Summer Airflow over the Antarctic Peninsula in Response to Recent Strengthening of Westerly Circumpolar Winds, Journal of the Atmospheric Sciences, 65, 1396-1413, 10.1175/2007JAS2498.1, 2008. van Lipzig, N. P. M., Marshall, G. J., Orr, A., and King, J. C.: The Relationship between the Southern Hemisphere Annular Mode and Antarctic Peninsula Summer Temperatures: Analysis of a High-Resolution Model Climatology, Journal of Climate, 21, 1649-1668, 10.1175/2007JCLI1695.1, 2008.

[Figure]

$OH_{TEMP} = -0.33 + 0.68 * ESP_{TEMP}$
r = 0.96

$OH_{TEMP} = -1.26 + 1.06 * BELL_{TEMP}$
r = 0.97

Temperature (°C) - OH

Temperature (°C)

- OH-BE$_{TEMP}$
- OH-ESP$_{TEMP}$

**Fig. 1.** OH-BE (green dots) and OH-ESP (blue dots) monthly mean air temperature correlation between 1968 and 2015

---

## Referee Report (RR1)

Comments on *New regional insights into the stable water isotope signal at the northern Antarctic Peninsula as tools for climate studies,* Fernandoy et al.

The manuscript presents valuable glacio-chemical data for the northern area of the Antarctic Peninsula. The authors present water stable isotope data sets of precipitation and firn cores collected near O'Higgins station. The authors use an innovative method to obtain the time scale of the firn cores which, the authors claim, cannot be dated by traditional methods such as annual cycles counting in the d18O profile. The authors then discuss the isotope-temperature relationship at different seasons and conclude that an isotope-temperature relationship cannot be valid for all seasons, but rather depends on seasonal variability of oceanic conditions.

This is the second time revising this manuscript. Many of the comments in the first revision have been addressed and the manuscript has improved accordingly. However, I still find relevant points that were not addressed by the authors in the final corrected version of the manuscript. These should be addressed before the paper is considered for publication.

**General comments**

1. In section 3.1, please indicate how the precipitation samples were collected, type of sample collector, sample bottles, handling and storage, etc.
2. RC. In section 3.2: please indicate the basis to the -1.4C latitudinal correction. It is not clear for me how the authors got the -1.4 C factor, this should be clear in the manuscript. The authors added a figure in the first response letter, but it is still not clear to me how they obtained the -1.4 factor.
3. When calculating the R values, the authors indicate alpha, however, they do not indicate if the R values were corrected considering the degrees of freedom of the system. That should be included.
4. The authors write: "The latter indication is well supported by the high accumulation rate in the region that does not allow a prolonged exposition of the freshly fallen snow to the atmosphere. Furthermore, the absence of significant infiltration and percolation associated with melting and refreezing events and the lack of a relationship between ice layers and seasons as well as with the stable water isotope record implies that the isotopic composition is not altered by surface melt infiltration and percolation". The authors should show the melt percentage per depth, ice layer distribution and thickness in a plot. If they have the data, I do not see the problem on doing it. Including these data in the manuscript will improve and give support to this part of the discussion. As it is, I am still not convinced by the authors' claims, basically because they do not show the data. If the melt layers are "few" why not to simply indicate how many per mwe the cores have? And then calculate melt percentage. In the previous revision of this manuscript, I've asked: "how the authors could explain the melt layers then if there is no signal of infiltration or connection with summer melt? Could the authors include the percentage of melt per m w.e.?" The authors response was: ANS. 12-20. In this line we expressed our

self not right. We actually don´t mean to state that a proper ice layer could form purely from wind in a high accumulation region like the AP. We refer here to actual glazed (ice) "crust", only a few mm wide as shown by the stratigraphy of the firn cores. This wind crust could form a thin glazed surface due to sublimation and snow drift abrasion, and in some opportunities by solidification of super-cooled droplets flowing against ground surface irregularities (sastrugi-like). During our field work, we witness these processes during different years. However, in the authors response + corrections uploaded on 2017-09-11, I still see the phrase: "Thus, this reassures that post–depositional processes in the LCL region are negligible in the time period analyzed and that ice layers likely developed by wind ablation on wind–scouring processes at the plateau." Therefore, the authors did not include the clarification in the paper.

5. (In 10. 31.) Please define what are "elevated values". A value higher than the mean could be simply a number 1*10^-8 above the mean, but that would not make sense to interpret as a result of a geophysical process, at least not in the manuscript context. Please define higher (or lower) values in terms of how many standard deviations the value is above or below the mean. Please revise the discussion from section 4.2.2 on, considering the above.

**Minor comments**

6. 30-34. Please indicate in the text how you have assessed that there is a significant difference in the Sdev of the cores.

7.20-23. Please re-write this sentence: "By studying the progression of these variations, the frequency of the second mode showed the highest frequency in the second interval, where the Sdev reaches an equilibrium. Thus, the final signal is only defined by a set of low frequencies.", e.g. By studying the progression of these variations, the frequency of the second mode showed the highest value in the second interval, where the Sdev stabilizes?. Thus, the final signal is only defined by a set of low frequencies.

In table 2 please write the mean, max and min values according to the sdev significant digits. Please revise this along the text and tables.

8. 11. The discarding of outliers should be described here, not in the following section (4.1.2.)

8. 24. Please indicate the standard error in the slope of the linear regression when using the MAM and SON datasets.

9. 8. "and show variability". This sound ambiguous, please elaborate on this.

10. 26-29. If the authors have the number and thickness of ice layers, and layer densities, please include the percentage of melt per m w.e. Without this calculation, this

paragraph does not support itself. This was asked in the previous revision and the authors response was that they will include this discussion, however they didn't calculate melt percentage, nor justified why they didn´t do it. In the previous revision, I asked the authors: "how the authors could explain the melt layers then if there is no signal of infiltration or connection with summer melt?" this has not been answered yet. Consequently, the phrase "(10. 26-27) The melt layers do not show evidence of infiltration" remains obscure. This must be clarified.

12. 29. Please indicate if the trend is significant and at what confidence level.
13. 10. "During MAM a clear decrease of $\delta 18O$ with height (-2.4‰ km-1 from sea level at OH up to LCL) is found, whereas during JJA no decreasing $\delta 18O$ trend is obtained from sea level to 1130 m a.s.l. (Fig. 13b).". Please define the significance of the trend, otherwise, the sentence is ambiguous.

14. 14. Here I am confused because the editor had already asked to correct the terms "inverse and direct relationships", however, I still find them in the text, despite the fact the authors claim, in the response to the editor, to have revised them.

2. 3. Warming of
2. 12. At an estimated rate of

2.13. Please indicate the rate of positive mass balance in East Antarctica

4.6-7. Please cite the ice-core studies that are available.

4.16. Eastern side

4.30. The authors mention "improper storing" without describing what is proper storing. See also the general comments.

5.4. Please indicate the temperature at which the cores were kept in the commercial freezer.

In Fig. 2, please indicate the resolution of the meteorological data.

8. 6.  In the last →in the previous

8. 10. Was constructed → was obtained

10. 17. "an even higher"→a higher

10. 20. "more fresh and less compacted". This is ambiguous. If the authors have layer density, please indicate it, e.g. consist of snow layers with density between xx and xx, and firn with densities between xx and xx.

---

## Author Response (AR2)

[revised manuscript text omitted]
 showed to bewere considerably noisy, with but values do not fluctuateting far fromclosely around the mean. Although Tthere was a significant difference inbetween the Standard Deviation (Sdev) values of oxygen isotopes between the cores from lower altitudes cores (OH-4, $\delta^{18}$O Sdev = 1.2) and versus the onescores from higher altitudes cores (OH-10, $\delta^{18}$O Sdev = 2.57) (Table 1), there

25   were no clearr detachments detachments within each signal. Furthermore, tThe patterns described of the isotope–depth profiles described lacked of patterns which could be attributed by their owndo not correspond to seasonal cycles. Despite several attempts were carried out to achieve a chronology by annual layer counting, the noise and the lack of a consistent behavior in the isotope signals inhibited this procedureanalysis. Difficulties resulted emerged from using conventional dating methods methodology with these firn cores led to the searchus to search for other types of methodsways that couldto provide

[revised manuscript text omitted]

Editor Decision: Reconsider after major revisions (26 May 2017) by Benjamin Smith

Comments to the Author:

Editor's comment:

5 Both reviewers recommended a large number of edits to the manuscript, and raised a few major points requiring more significant revisions. The authors made almost all of the recommended major revisions, and all of the minor revisions, but the revised paper is still very difficult to read because of non-standard use of English expressions, and because the analysis in the methods section (section 3.x) is not well motivated. I would recommend a thorough re-write of the paper with careful attention to how the reader is to understand the choices the authors made in their analysis, and detailed editing. Some of my

10 own comments on the revised manuscript are below.

Ans. This was completely reviewed and rewritten as suggested by the editor

Section 1. Reviewer 1 asked for more information on the glaciological setting, and the authors said that they would provide it. This is not apparent in the revised manuscript.

Section 2.2:

Both reviewers questioned the choice of station BE for the temperature and precip records, and the authors gave them a plot and a rationale for the choice in their response to the reviews. This information needs to be conveyed in some way in this section of the paper.

20 Ans. Additional information was added to the introduction section as suggested

Section 2.3:

This section needs to be carefully checked for English grammar and word choice. It is at present difficult to interpret, and leaves considerable room for misinterpretation by readers.

The FFT/IFFT procedure appears to be important in dating the cores, but the description of the algorithm does not explain how the analysis was performed. A more detailed technical discussion, including range of frequencies chosen, is needed here.

Ans.: Done as suggested the whole section was reviewed and rewriten

30 Section 2.4:

It is not clear how the GPR results relate to the other analysis performed here. It does not appear that the authors make much use of these data, and I suggest removing this section from the paper to save space.

Ans.: As suggested, this section was eliminated and will later revisited in a future publication

3.1.1 and 3.1.2

These sections also needs rewriting for clarity. The authors use the term "direct relation" and "inverse relation" to mean (I think) "positive correlation" and "negative correlation." They also use a very complex sentence structure without many clues to the reader as to the significance of different parts of their analysis. More motivation and more standard use of English idiom would help here.

When presenting statistical significance values, the authors should make clear how many independent variables they believe they have. Is every measurement statistically independent from every other?

Ans.: The section was reviewed and reformulated for better clarity of our ideas and evidence.

3.2.1.

Reviewer 2 suggested that using the d_excess_meteo to date the core should result in strong correlations between d_excess_meteo and d_excess_core, so the correlation between the two should not be surprising. The authors claimed that this resulted from a misstatement in the original manuscript, but the revised MS appears to make the same claim. If I am misreading this, the text needs to be clarified to prevent this misunderstanding.

Ans.: Done as suggested by the editor. The section was modified to better express our discussion.

3.2.2

It is not clear from the discussion what the 2-year pattern is. The sentence that should explain it has two clauses, and it is difficult to understand which of the two expresses the nature of the signal.

This section gives a fairly confusing analysis of the data. Are the authors using three different D18-O-vs-temperature functions for three different seasons? What are these functions? Are they from the accumulation –sampling analysis? If they're derived from the core, how do we know that they express something significant about the climate? It seems that by allowing themselves to select three different D18-O-vs-temperature relationships for two different sets of years, the authors have too many degrees of freedom in interpreting their data. With this many adjustable parameters they are bound to find correlations among the results. Am I just misreading this section? Again, the analysis needs more motivation.

Ans.: Done as suggested, the section was revised and clarified the points mentioned by the editor.

The justification for using BE should be presented in the data section, rather than here.

Ans.: The information was now added to the introduction

Section 4.1:

P. 13 Line 26: "mean $\delta$18O and $\delta$D values show a decreasing tendency with increasing height from sea level" : what does "tendency" mean here. Correlation?

Ans.: This sentance was reformulated as suggested by the editor

We checked again the text through-out to revise the use of English, this was done by North-American native speaker together with the autors.

---

## Author Response (AR3)

**Dear editor Dr. Benjamin Bell**

In response to the last comments: We highly appreciate your commitment to the quality of this manuscript. We addressed all your comments in this new version, that we consider has strongly improved.

**Observation 1:**

First, I would like you to see my previous comments about the use of "direct relationship" and "inverse relationship." These terms are not clear-- I think you mean positive linear correlation and negative linear correlation, but I'm not entirely sure. This comes up often in the paper, and is consistently confusing.

**Answer from the authors:**

The terminology was revised and changed along the text now referring to positive or negative correlations depending on the results of each analysis.

**- Observation 2:**

Second, please provide the number of degrees of freedom when you provide a P value for a statistical correlation. The P values you state are consistently small (with two or three exceptions) even though the correlations are weak, but I am not sure the number of degrees of freedom necessarily justifies the interpretation that the correlations are unlikely to have come about by random chance. If two low-frequency-content signals are compared, they may show a strong linear correlation simply because the low-frequency signals have similar periods. In these cases, assuming that the number of degrees of freedom is equal to the number of data points is incorrect, and results in spuriously small P values.

**Answer from the authors:**

For all relevant correlation along the text, a t-statistic test of each correlation (Pearson's) was performed and significance described, in order to screen possible false significance of p values. All statistically significant correlations were found to be statistically significant at alpha 0.05.

**Observation 3:**

Third, and perhaps most important: Please provide motivation to the statistical correlations examined in sections 2 and 3. You frequently describe analyses you have performed without explaining what they were intended to accomplish, which makes the discussion in section 4 very difficult to relate to the analysis performed in section 3.

**Answer from the authors:**

Section 2, 3 and 4 were revised and partially rewritten in order to improve the clarity of the statistical analysis done and it meaning better explained. Section 4 was also revised to better explain the results achieved.

**- Observation 4:**

**P3 ~30 – the glaciological background should go in its own section.**

**Answer from the authors:**

This was done as suggested and included in the new section 2

**- Observation 4:**

P6 26: You need to specify that this is now the reanalysis data you are talking about (e.g. "For every precipitation event registered in the (NCEP?) data")

**Answer from the authors:**

This was modified by "for each day when a precipitation sample was collected at OH"

**Observation 5:**

P9, top- Need to specify whether the "precipitation samples" are modeled or are actual samples from a different study.

**Answer from the authors:**

Done as suggested and changed to "precipitation samples gathered at OH"

**- Observation 6:**

P9 line 20: Need to evaluate the significance of the correlation between  $d_{excess\ meteo}$  and  $d_{excess\ core}$  in light of the fact that  $d_{excess\ meteo}$  was use to date the core. What is the probability that one time series, stretched to match another, shows a 75% correlation with the series it is stretched to match? How many degrees of freedom are involved?

**Answer from the authors:**

Similar to other correlation a t-statisc test was carried out for this correlation, furthermore  $d_{excess \ core}$  was calculated from a composite from different firn cores and not directly used in the  $d_{excess \ meteo}$  and  $d_{excess \ core}$  correlation applied for the dating of each single core.

**Observation 7:**

P9 line ~29: What standard deviation is in question here? The amplitude of the oscillations in the cores? The difference between the cores?

**Answer from the authors:**

This was clarified, referring to the Sdev of the  $\delta^{18}O(\delta D)$  values from the 5cm samples.

**- Observation 8:**

P 10, Line 16-20: Where do you describe the bounds on the seasons you have defined this way (after sometimes stretching seasons to include groups of similar data?) How and where do you use these classifications?

**Answer from the authors:**

This is now addressed in section 4.2.2 of the manuscript.

[revised manuscript text omitted]

Seeing aAs calendar seasonsal behavior\_in these latitudes doesnot not play a significant rolefollow higher latitudes patterns (i.e. DJF, MAM, JJA, SON)<del>,</del> sinceand\_seasonality, which largely often extends beyond calendar limits, depends on the sea

- 10 ice cover during winter, often extends beyond calendar limits. Large sea ice extent (SIE) leads to a delayed on-set of spring conditions. In this case winter-like conditions will be extended beyond August. Restricted sea ice extent on the contrary will lead to earlier spring-like conditions (before August). Depending on this conditions, we defined three seasons with their correspondent corresponding δ18O—T relationship. These seasons are: (1) an austral transitional season which considers the months from March to May and October–November (MAM-ON) (using precipitation datasets from March 2008–2009 and
- 15 October 2014 for the  $\delta^{18}$ O-T relationship during that season, *s*=0.69), (2) an austral winter season which considers the months from June to September (JJAS) (using precipitation datasets from June 2008 for the  $\delta^{18}$ O-T relationship during that season, *s*=0.35), and (3) an austral summer season which considers months from December to February (DJF) (considers precipitation datasets from December 2008 for the  $\delta^{18}$ O-T relationship during that season, *s*=1.17) (See-compare  $\delta^{18}$ O-T relationship functions in section 34.1.1). Despite this main seasonal classification for the use of the  $\delta^{18}$ O-T relationship,
- 20 some particular seasons showed variable behavior when compared to the mean seasonal behavior in the time span covered in this study. In those cases, the seasonal behavior was extended or contracted beyond the boundaries of the main season classification depending on the SIE sea ice extent. Large sea ice extent during winter will leads to a delayed on set of spring conditions. In this case winter like conditions will be extended beyond August. Restricted sea ice extent on the contrary will lead to earlier spring like conditions (before August).
- 25

**4.2.3 Air temperature trends at Laclavere Plateau**

Additionally, Using the relationship between the linear correlation between meteorological data: monthly lapse rates in BE, winter SAM index and SIE sea ice extent (SIE) from OH is represented in (Fig. 11), - Considering latitude corrected air

30 temperatures from BE, lapse rates from BE and SIE from OH (SIEOH), a mean annual air temperature of  $-7.5^{\circ}$ C with a trend of  $-0.18^{\circ}$ C year-1 (statistically not significant at *p*=0,05) was estimated on LCL for the time period 2009 2014. aA monthly temperature mean estimate\_, derived from the linear correlation between meteorological data and the monthly lapse rate\_SIE relationshipthat, can be expressed by the equation  $T_{LCL} = (T_{BE}-1.4)+1.13$  (Mmonth\*SIEOH+Nmonth), during the months when sea ice is developed (from May to September) and where  $M_{month}$  and  $N_{Month}$  represent the slope and intercept of the monthly lapse rate–SIE relationship, respectively. During the months when there is no sea ice (from October to April) the monthly temperature can be expressed by the equation  $T_{LCL} = (T_{BE}-1.4)+1.13 * H(t)$ , where H(t) is the monthly mean lapse rate value of the month t measured in BE between 1978-1996. Considering this variables, a mean annual air temperature of -7.5°C with

- 5 a trend of -0.18°C year-1 (statistically not significant at p=0.05) was estimated on LCL for the time period 2009-2014. AdditionallyOn the other hand, considering the  $\delta^{18}$ O time series data and the isotope–T relationship, a mean annual air temperature of -6.5°C with a trend of -0.33°C year-1 (statistically not significant at p=0.05) was estimated on LCL for the years 2009-2014. The comparison between monthly mean temperature on LCL, estimated using the  $\delta^{18}$ O signal from firn cores and TLCL estimated using the coupled effect of the latitude–corrected temperature record from BE, SIEOH and lapse
- 10 rates from BE, reveals a correlation coefficient of R=0.7 (p<0.01; t-statistics test statistically significant at alpha 0.05). Both signals show a synchronous behavior, also with respect to the air temperature record at OH station. No statistically significant direct correlation was observed between coastal stations (OH and BE) temperature records and the stable water isotope composition of firn cores.

Comparing the 818O signal from OH 6 with data from precipitation samples at OH and with two other cores from the western

15 side of the AP (OH 4 and OH 5) during a common period (March 2008 – August 2008), a δ18O decrease of -0.085‰ km-1 was found with increasing distance from the coast (Fig. 12a). The same data set was used to study the δ18O altitude relationship. The δ18O seasonal means show an altitude dependence dependency through whichthat yields seasonal δ18O 
[revised manuscript text omitted]

5

---

## Author Response (AR4)

**Observations by the Editor**

**Obs. 1:**

Methods: The methods are not always clearly described. In particular, in section 3.3, the Fourier transform procedure does not make sense. The IFFT is usually used to convert from the frequency domain to the space domain, not the other way around. The expression "The signal obtained from each step of the FFT decomposition" does not mean anything to me, and the statement that "By studying the progression of these variations, the frequency of the second mode showed the highest frequency where the Sdev reaches an equilibrium" needs to be rewritten, as it does not correspond to anything I understand about signal processing.

**Response:**

Sections 3 and 4 were rewritten and reformulated accordingly and the focus of the analysis was modified. With these changes we aim to emphasize that the analysis using FFT aimed to extract low frequency oscillations from a noisy signal obtained from the isotope record. The filtered signal was then compared with meteorological data retrieved from re-analysis data using a linear regression analysis. The linear regression model was fitted to the whole data set and in this way the signal was transformed from the space domain (depth) to the time domain (age). Thereafter temperature, accumulation rate and environmental conditions trends were reconstructed.

**Obs. 2:**

I think what is going on is that the authors looked at the FFT spectrum of the signal and decided what parts of it were meaningful, then edited out the higher-frequency, less meaningful portion and used the IFFT to produce a filtered signal containing only low-frequency information. They then matched the peaks and troughs of the two d signal to date the core. What is not clear is whether only a single rescaling value was adequate to match the core and the meterological signal, or whether different stretching values were used to move each peak in the core inline with the meterological signal. I, and the referees, assumed the latter, based on which the correlations drawn between the core and the meterological signal are meaningless. Under this understanding, the authors' explanation that the correlation was between the composite core and the meterological records is not helpful, because the composite core would suffer the same, or nearly the same, spurious correlation with the meterological data. If only a single stretching value was used, then the correlations between the two signals may be significant. Someone familiar with standard English terminology for Fourier analysis needs to read and revise this section carefully.

**Response:**

This section has been rewritten to make it clear that the stretching value we found via the comparison with the modeled d-excess has been applied to the complete dataset, we have not stretched specific sections with different values. The correlations can therefore be calculated using the standard method and the number of degrees of freedom is n-2. All alpha values are 0.05 and

we have therefore removed them from the text. Instead we indicate the degrees of freedom (n-2) for all correlations.

**Obs. 3:**

Results: As in previous versions, the motivation for the correlations presented in section 4 were not at all easy to follow in the revised manuscript. Each time the authors compare two variables and calculate a correlation value between them, the need to explain WHY they performed this correlation, WHAT they expected to learn from the correlation, and what the statistical test indicates.

**Response:**

Section 4 has been carefully reviewed and rewritten in order to make the objective of each analysis clear. This new version explicitly states for each comparison, what is the specific aim and result obtained from each analysis.

**Obs. 4:**

The alpha values now included in the manuscript do not generally add clarity to the results: much more helpful would be to list the number of degrees of freedom in the correlations that was used in calculating the p values. Presenting the correlation results in line with the text generally makes the paper much harder to read. In particular, the long paragraph in 4.1.1 could be improved by presenting the s values and their significance statistics in a separate table, along with the monthly correlations.

**Response:**

As explained in the response to Obs. 2, we thoroughly revised and reformulated Section 4. The number of degrees of freedom were added to each calculation. All slope, significance level, standard errors and further statistical data is presented in Table 4 and only strictly necessary information is shown in the text.

**Obs. 5:**

Section 4.1.1, line 25: This sentence: "Thus, the time series of monthly averages shows a positive correlation between both parameters" did not seem to follow from the other material in the paragraph, which is largely about the daily correlations (the s values).

**Response:**

This sentence has been removed. This sentence was no longer needed, and unfortunately we failed to eliminate it in the last version.

**Obs. 6:**

General considerations: The use of the terms "inverse/direct behavior" and "inverse/direct relationship" remains common in the manuscript. I recommended, and the reviewers agreed, that this expression should not be used, and that "positive/negative correlation" should be used instead.

**Response:**

We regret that we didn't correct this properly. We have now reviewed this terminology to only use positive/negative correlation.

**Obs. 7:**

The authors need to break the discussion section into shorter paragraphs, with a topic sentence for each to indicate what point they are trying to make, and why. The very long paragraphs used here are difficult to untangle, and the line of the argument gets lost.

**Response:**

The entire paper has been reworked to improve its structure, coherence and to better direct the reader towards an improved understanding of our results and discussion. We firmly believe that this was achieved in this updated version.

**Non-public comments to the Author:**

Dear Dr. Fernandoy, After the second round of revisions, a manuscript like this should be very close to ready for publication. I do not see that this is the case here. I think the referees have made a good effort at reviewing the manuscript again, but do not have the patience to provide detailed comments on the results and discussion sections of the paper, except to echo some of the points that I have made about them. Before I would be willing to send a revised manuscript out for more reviews, I would like to see extensive revisions to the results and discussion sections so that the reader can easily follow why each step in the correlation analysis was performed, and what the results mean. I would also like to see extensive editing for English, following my recommendations against the use of the terms 'direct/inverse relationship.' I also would like to see an acknowledgement of the potential for spurious correlation between the stretched meterological record and the core, and a good justification for why the correlation is potentially significant (if it is.). While this may seem a minor point, it is something that will be important to anyone interested in trying the same method for their own core.

**Response:**

Dear Dr. Smith, Firstly on behalf of the co-authors, I would like to apologize for the frustration caused whilst editing this manuscript. In review of the publication, we realized that the methods

needed to be rewritten to put the results and discussion sections into context. Subsequent to clarifying the methods used, we rewrote sections of the results and discussion to clarify the outcomes and related interpretations. For this purpose, we invited two new co-authors with extensive scientific and publishing experience. Dr. Shelley MacDonell is widely re-known glaciologist and hydrologist, adding that she is a native English Speaker. Dr. Fabrice Lambert is a statistics and glacio-chemistry expert. Both have been directly involved in editing the whole manuscript, adding a deeper insight into the statistical treatment and improving the scientific integrity of the manuscript, as well as improving language and terminology issues found in the previous versions. Within this process, we focused on using a more direct approach, which is related to extensive English editing. We hope that the new version clarifies the approach we have taken and that within the new version, the impact of our results will be more apparent. We appreciate the time that you and the reviewers have taken in revising this manuscript, and regret the time it has taken to improve the manuscript, hopefully to the point of now being publishable.

**Observations by Reviewer 1**

Comments on New regional insights into the stable water isotope signal at the northern Antarctic Peninsula as tools for climate studies, Fernandoy et al. The manuscript presents valuable glaciochemical data for the northern area of the Antarctic Peninsula. The authors present water stable isotope data sets of precipitation and firn cores collected near O'Higgins station. The authors use an innovative method to obtain the time scale of the firn cores which, the authors claim, cannot be dated by traditional methods such as annual cycles counting in the d180 profile. The authors then discuss the isotope-temperature relationship at different seasons and conclude that an isotope-temperature relationship cannot be valid for all seasons, but rather depends on seasonal variability of oceanic conditions. This is the second time revising this manuscript. Many of the comments in the first revision have been addressed and the manuscript has improved accordingly. However, I still find relevant points that were not addressed by the authors in the final corrected version of the manuscript. These should be addressed before the paper is considered for publication.

**Response:**

Firstly, we regret overlooking comments made on the previous version of this paper. In this iteration, we have carefully considered each of the reviewer's comments, and have edited the document as appropriate. We would like to thank the reviewer for their detailed review of this publication.

**General comments**

**Obs. 1:**

In section 3.1, please indicate how the precipitation samples were collected, type of sample collector, sample bottles, handling and storage, etc.

**Response:**

This information has been included in Section 3.1.

**Obs. 2:**

RC. In section 3.2: please indicate the basis to the -1.4C latitudinal correction. It is not clear for me how the authors got the -1.4 C factor, this should be clear in the manuscript. The authors added a figure in the first response letter, but it is still not clear to me how they obtained the -1.4 factor.

**Response:**

This correction is based on the difference of mean temperature between the two automatic weather stations (Bellingshausen and O'Higgins). They have a high correlation (R = 0.97), and show similar temperature oscillations at a daily scale. O'Higgins Station is on average 1.4°C colder than Bellingshausen Station, and is located around 150 km south of Bellingshausen. Therefore, this latitudinal difference is what we originally referred to as the "latitude correction". Due to the confusion in the name, we have changed it to be "temperature correction". We added the following explication to the text: "The temperature record from OH contains several large data gaps, and so the available data from 1968 to 2015 were compared with those measured at BE to evaluate the possibility of lapsing data from BE to the site due to the data continuity available (uninterrupted record since 1968). The BE and OH data are highly correlated (R=0.97, p<0.01), and so a correction of -1.4°C was applied to the BE data based on linear regression analysis."

**Obs. 3:**

When calculating the R values, the authors indicate alpha, however, they do not indicate if the R values were corrected considering the degrees of freedom of the system. That should be included.

**Response:**

See response to observation 2 from the Editor. To summarize: the stretching value estimated from the comparison with modeled d-excess has been applied to the complete dataset, we have not stretched specific sections with different values. The correlations were calculated using the standard method and so the number of degrees of freedom is n-2. All alpha values are 0.05 and we have therefore removed them from the text. Instead we indicate the degrees of freedom (n-2) for all correlations in this section.

**Obs. 4:**

The authors write: "The latter indication is well supported by the high accumulation rate in the region that does not allow a prolonged exposition of the freshly fallen snow to the atmosphere. Furthermore, the absence of significant infiltration and percolation associated with melting and refreezing events and the lack of a relationship between ice layers and seasons as well as with the stable water isotope record implies that the isotopic composition is not altered by surface melt infiltration and percolation". The authors should show the melt percentage per depth, ice layer

distribution and thickness in a plot. If they have the data, I do not see the problem on doing it. Including these data in the manuscript will improve and give support to this part of the discussion. As it is, I am still not convinced by the authors' claims, basically because they do not show the data. If the melt layers are "few" why not to simply indicate how many per mwe the cores have? And then calculate melt percentage. In the previous revision of this manuscript, I've asked: "how the authors could explain the melt layers then if there is no signal of infiltration or connection with summer melt? Could the authors include the percentage of melt per m w.e.?" The authors response was: ANS. 12-20. In this line we expressed our self not right. We actually don't mean to state that a proper ice layer could form purely from wind in a high accumulation region like the AP. We refer here to actual glazed (ice) "crust", only a few mm wide as shown by the stratigraphy of the firn cores. This wind crust could form a thin glazed surface due to sublimation and snow drift abrasion, and in some opportunities by solidification of super-cooled droplets flowing against ground surface irregularities (sastrugi-like). During our field work, we witness these processes during different years. However, in the authors response + corrections uploaded on 2017-09-11, I still see the phrase: "Thus, this reassures that post-depositional processes in the LCL region are negligible in the time period analyzed and that ice layers likely developed by wind ablation on wind-scouring processes at the plateau." Therefore, the authors did not include the clarification in the paper.

**Response:**

In this reviewed version we include percentage melt, as we agree with the reviewer that this is necessary information to show that melting is not an important with relation to the high accumulation of this region. Nonetheless, we think a new figure is not absolutely necessary, since the melt events are very restricted and not regularly distributed (i.e.: Seasonal distribution). This manuscript has already several figures and the authors consider that, this figure in particular doesn't contribute vital information to the discussion. Melt percentage were calculated taking in account only, regular and thicker ice layer. Thinner (<1 mm to 1 cm) and irregular glazed crusts are not taken in account in this calculation, as they don't represent melt, but wind-scouring and droplets solidification.

We will add our previous explanation to the processed, as we recognized that we failed to make it clear as the reviewer correctly points that out. Explanation and correction where added to section 4.2.1 (p. 10, l. 296-304) and section 5.1 (p. 14, l. 439-444)

**Obs. 5:**

(In 10. 31.) Please define what are "elevated values". A value higher than the mean could be simply a number 1\*10^-8 above the mean, but that would not make sense to interpret as a result of a geophysical process, at least not in the manuscript context. Please define higher (or lower) values in terms of how many standard deviations the value is above or below the mean. Please revise the discussion from section 4.2.2 on, considering the above.

**Response:**

To make this section of the discussion clearer, we remove the term "elevated values" and have modified this phrase to read: "This pattern is characterized by positive monthly mean  $\delta$ 180 standardized anomaly values (z = observation - mean \* Std. dev.-1) between May and November in the years 2008 (z = 0.6), 2010, 2012 and 2014 (z > 1). Additionally they exhibit a negative correlation to temperatures at BE (z = -0.7 for 2010, and < -1 for 2008, 2012 and 2014). Between June and July in the following years 2009, 2011 and 2013,  $\delta$ 180 values (z < -1) are lower than the mean and exhibit a positive correlation to temperature at BE (z < -1) (Fig. 9)". This enables improved understanding of the periodical pattern identified, and a better introduction to Figure 9. The discussion in section 4.2.2 is still consistent with the previous statement.

**Minor comments**

**6. 30-34.** Please indicate in the text how you have assessed that there is a significant difference in the Sdev of the cores.

**Response:**

This is now included: "there was a significant difference in the standard deviation (Sdev > 1.0) of high resolution (5 cm) oxygen isotopes values ".

**7.20-23.** Please re-write this sentence: "By studying the progression of these variations, the frequency of the second mode showed the highest frequency in the second interval, where the Sdev reaches an equilibrium. Thus, the final signal is only defined by a set of low frequencies.", e.g. By studying the progression of these variations, the frequency of the second mode showed the highest value in the second interval, where the Sdev stabilizes?. Thus, the final signal is only defined by a set of low frequencies.

In table 2 please write the mean, max and min values according to the sdev significant digits. Please revise this along the text and tables.

**Response:**

The first paragraph and the whole section have been revised. Significant digits of values within all tables and within the text have been corrected.

8. 11. The discarding of outliers should be described here, not in the following section (4.1.2.)

**Response:**

In the revised version, this section was moved from Section 4.1.2. to Section 4.1.1.

**8. 24.** Please indicate the standard error in the slope of the linear regression when using the MAM and SON datasets.

**Response:**

A new table (Table 3) including information of the linear regression was now included.

**9.8.** "and show variability". This sound ambiguous, please elaborate on this.

**Response:**

The whole paragraph has been now rewritten and this phrase has been removed.

**10. 26-29.** If the authors have the number and thickness of ice layers, and layer densities, please include the percentage of melt per m w.e. Without this calculation, this paragraph does not support itself. This was asked in the previous revision and the authors response was that they will include this discussion, however they didn't calculate melt percentage, nor justified why they didn't do it. In the previous revision, I asked the authors: "how the authors could explain the melt layers then if there is no signal of infiltration or connection with summer melt?" this has not been answered yet. Consequently, the phrase "(10. 26-27) The melt layers do not show evidence of infiltration" remains obscure. This must be clarified.

**Response:**

This information has been included in Sections 4.1.2 and 5.1. Our observations are based on visual inspection and density profiles of the cores. We defined two different types of high density layers (ice layers and crusts), which were defined based on morphology, and continuous, regular layers thicker than 1 cm are associated to melt events. The thickest layer identified is approximately 5cm, and has defined borders. This implies that these events were not extended in time. Other high density structures which are relatively thin (less than 1 cm) and irregular were associated to other phenomena such as wind-scour ablation and super-cooled droplet solidification. These observations are now stated in the text.

**12. 29.** Please indicate if the trend is significant and at what confidence level.

**Response:**

The information is now included in this section and stated as: "A marked decrease was observed in the accumulation during JJA and SON between 2008 and 2015, which could be in part responsible for the overall decreasing rates. However, this trend was found to be statically non–significant (p>0.1)"

**13. 10.** "During MAM a clear decrease of  $\delta$ 180 with height (-2.4‰ km-1 from sea level at OH up to LCL) is found, whereas during JJA no decreasing  $\delta$ 180 trend is obtained from sea level to 1130 m a.s.l. (Fig. 13b).". Please define the significance of the trend, otherwise, the sentence is ambiguous.

**Response:**

This was now included as: "Between OH (0 m a.s.l.) and LCL (1130 m a.s.l.) during MAM, a clear decrease of  $\delta$ 180 with height is observed (-2.4‰ km-1 with R=0.97 at p level<0.05), whereas during JJA no significant decreasing  $\delta$ 180 trend is observed (Fig. 14b)"

**14. 14.** Here I am confused because the editor had already asked to correct the terms "inverse and direct relationships", however, I still find them in the text, despite the fact the authors claim, in the response to the editor, to have revised them.

**Response:**

We regret this oversight in the previous version of the text. We have reviewed the text carefully, and made the respective modifications.

2.3. Warming of

**Response:**

Corrected as suggested

2. 12. At an estimated rate of

**Response:**

Corrected as suggested

2.13. Please indicate the rate of positive mass balance in East Antarctica

**Response:**

Information from Harig and Simmons (2015) has been included, and the section now reads: The glaciers of the AP have lost ice mass at a rate of approximately 27 ( $\pm$ 2) Gt y-1 between 2002 and 2014. This mass loss combined with the mass loss over the West Antarctic ice sheet (121( $\pm$ 8) Gt y-1), surpassed the mean positive mass balance of +62 ( $\pm$ 4) Gt y-1 observed in East Antarctica, of which most of the positive balance relates to the Dronning Maud Laud region whereas the mass balance of the rest of the EAIS is at equilibrium (Harig and Simons, 2015).

**4.6-7.** Please cite the ice-core studies that are available.

**Response:**

The following references are now included: Aristarain et al, 2004; Simoes et al, 2014; Goodwin et al, 2015; Fernandoy et al, 2010, Dalla Rosa, 2013

4.16. Eastern side

**Response:**

Corrected as suggested

**4.30.** The authors mention "improper storing" without describing what is proper storing. See also the general comments.

**Response:**

Additional information is now included: "The overwintering crew at O'Higgins Station collected daily precipitation samples from pluviometers installed at the meteorological observation site. Each daily sample comprised of a filling a narrow neck HDPE type bottle with a 30 ml composite sample of the precipitation (both liquid and solid) that fell in the previous 24 hours. The bottles were tightly closed and stored frozen year–long to ensure correct storage and to facilitate the subsequent transport to the laboratory at the end of each year. From these samples, approximately 6% (13 samples) were discarded from the analysis due to improper storage causing leakage from the bottles. Improper storage was assessed using a statistical outlier test (modified Thompson tau technique) which indicated unusual values of stable water isotope analyses."

**5.4.** Please indicate the temperature at which the cores were kept in the commercial freezer.

**Response:**

The information was now included: "and later transported and stored at -20°C in a commercial cold store in Viña del Mar, Chile."

In Fig. 2, please indicate the resolution of the meteorological data.

**Response:**

The information is now included; all data sets correspond to monthly means from the meteorological observatory at Bellingshausen Station.

8.6. In the last - in the previous

**Response:**

This was rephrased to read: "across the Bellingshausen Sea during the 24 hours before the air parcels reach the AP (Fig. 5)."

8. 10. Was constructed - was obtained

**Response:**

Corrected as suggested

10. 17. "an even higher" - a higher

**Response:**

Corrected as suggested

**10. 20.** "more fresh and less compacted". This is ambiguous. If the authors have layer density, please indicate it, e.g. consist of snow layers with density between xx and xx, and firn with densities between xx and xx.

**Response:**

This information is now included and the section reads: For the overlapping time interval in OH-9 and OH-10 (February 2012 to January 2014), we only considered data from OH-9, as these samples consist of fresher snow layers with density between 350 and approximately 410 kg m-3 (at core depth between 0 and 1 m), and firn with densities between approximately 410 and 530 kg m-3 (at core depth between 1 and 7 m) than the corresponding interval in OH-10.

**Observations by Reviewer 2**

I thank the authors for addressing many of my concerns in their updated manuscript. It is much improved from the previous version.

**Obs. 1:**

One main concern I still have is dating the core by peak matching dexcess\_core with dexcess\_meteo. The authors describe this on page 10 of the revised manuscript around line 10 as "peak-valley fitting". My understanding is that the authors then present the correlation between the peak-matched datasets as a validation of the dating method. My original point is that this seems circular – we should expect the dexcess in the core and the observations to match since they've been aligned. I also agree with the Editor's comment about artificially increased correlations due to smoothing of the time series. There's a need to account for reduced degrees of freedom in testing the significance. While the authors now present t-statistics in their revised manuscript, it's not clear that these were calculated using reduced degrees of freedom (not just n-2) as the editor suggests doing. Perhaps the method presented by Bretherton et al 1999 (J. Climate) would be appropriate here, where an effective sample size (n\*) is calculated by accounting for lag-1 serial correlation in time series, and from which to effective degrees of freedom, and then t and p values are calculated.

**Response:**

As exposed to the observation to the Editor and Reviewer 1 comments, in this version, we improved the description of statistical treatment, that we clearly didn't correctly expressed in the previous version of this manuscript. We regret, that we didn't achieved that before and probably caused some confusion to the reviewer. We apologize for this in advance.

As expressed to the Editor in his first observation:

Sections 3 and 4 were rewritten and reformulated accordingly and the focus of the analysis was modified. With these changes we aim to emphasize that the analysis using FFT aimed to extract low frequency oscillations from a noisy signal obtained from the isotope record. The filtered signal was then compared with meteorological data retrieved from re-analysis data using a linear regression analysis. The linear regression model was fitted to the whole data set and in this way

the signal was transformed from the space domain (depth) to the time domain (age). Thereafter temperature, accumulation rate and environmental conditions trends were reconstructed. Moreover, in response to the Editor's observations to sections 3 and 4:

This section has been rewritten to make it clear that the stretching value we found via the comparison with the modeled d-excess has been applied to the complete dataset, we have not stretched specific sections with different values. The correlations can therefore be calculated using the standard method and the number of degrees of freedom is n-2. All alpha values are 0.05 and we have therefore removed them from the text. Instead we indicate the degrees of freedom (n-2) for all correlations.

**Obs. 2:**

The data on figure 4 (the old figure 7) has changed since the last version, substantially in some cases. I am curious why this is. The data now appear to show better correlations.

**Response:**

The reviewer is right in this point, we unfortunately used an unfiltered data-set to produce the mean monthly values for the first version of the figure. This issue was later corrected, taking out all outlier data points and producing the correct monthly mean values. This didn't affect any of the statics carried out for this data, since the data-set used for the mathematical treatment was corrected filtered. Therefore, the new version of the figure, offers a better (visual) correlation of the data. The time axis (x-axis) was modified accordingly.

**New insights into the use of stable water isotopes at the northern Antarctic Peninsula as a tool for regional climate studies New regional insights into the use of Regional\_stable water isotopes**

signal at the northern Antarctic Peninsula as atools for regionalpolar climate studies

Francisco Fernandoy1, Dieter Tetzner2, Hanno Meyer3, Guisella Gacitúa4, Kirstin Hoffmann3, Ulrike Falk5, Fabrice Lambert6, Shelley MacDonell7

1Facultad de Ingenieria, Universidad Andres Bello, Viña del Mar, 2531015, Chile
 2Center for Climate and Resilience Research, Universidad de Chile, Santiago, 8370361, Chile
 3Alfred Wegener Institute Helmholtz Centre for Polar and Marine Research, Research Unit Potsdam, Telegrafenberg A43, 14473 Potsdam, Germany.

4Programa GAIA-Antártica, Universidad de Magallanes, Punta Arenas, 6210427, Chile

5Climate Lab, Geography Department, University Bremen, 28334 Bremen, Germany

6PDepartment of Physical Geography, Pontificia Universidad Católica de Chile

15 7Centro de Estudios Avanzados en Zonas Áridas (CEAZA), La Serena, Chile....

5

Correspondence to: Francisco Fernandoy (francisco.fernandoy@unab.cl)

Abstract. Due to recent atmospheric and oceanic warming, The-the Antarctic Peninsula is one of the most challenging regions of Antarctica to understand both local- and regional-scale climate signals from a climatological perspective, owing to

- 20 the recent atmospheric and oceanic warming. The sSteep topography and a lack of long-term and in situ meteorological observations complicate the extrapolation of existing climate models to the sub-regional scale. Therefore, new techniques must be developed to better understand processes operating in the region. For example, iIsotope signals are traditionally related mainly to atmospheric conditions, however, but a detailed analysis of individual components can give new insight into oceanic and atmospheric processes. This paper aims to use new isotopic records collected from snow and firn cores in
- 25 conjunction with existing meteorological and oceanic datasets to determine changes at the climatic scale in the northern extent of the Antarctic Peninsula.

In particular, Here, we present new evidence from the northern Antarctic Peninsula to demonstrate how stable water isotopes of firn cores and recent precipitation samples can reveal climatic processes related to nearby oceanic and atmospheric

30 conditions. Aa noticeable discernablediscernible aeffecteffect of the sea ice cover on local temperatures and the expression of atmospheric climatic modes, in particular the Southern Annular Mode (SAM), is demonstrated.

In years with a large sea ice extension in winter (negative SAM anomaly), an inversion layer in the lower troposphere develops at the coastal zone. Therefore, an isotope-temperature relationship ( $\delta$ -T) valid for all seasons-periods cannot be

- 35 concludedobtained, and instead,- The the  $\delta$ -T- relationship rather-depends on the seasonal variability of oceanic conditions. Comparatively, Transitional-transitional seasons (autumn and spring) are both stable seasons with anhave a consistent isotope-temperature gradient of +0.69‰ °C-1. The As shown by firn core analysis, firn stable isotope composition reveals that the near-surface temperature at in the most-northern-most portion of the Antarctic Peninsula shows a decreasing trend (-0.33°C y-1) between 2008 and 2014. MoreoverIn addition, the deuterium excess (*dexcess*) has been is demonstrated to be a
- 40 reliable indicator of seasonal oceanic conditions, and therefore suitable to improve a firn age model based on seasonal dexcess variability. The annual accumulation rate in this region is highly variable, ranging between 1060 kg m-2 y-1 and 2470 kg m-2 y-1 from 2008 to 2014. The combination of isotopic and meteorological data in areas where data exist is-a key for-to

reconstructing climatic conditions with a high temporal resolution in polar regions Polar Regions where no direct observations exist.

45

**1** Introduction**

West Antarctica, and especially the Antarctic Peninsula (AP), hasve been in the scope of received increasing attention from the scientific community due to the notable effects of the present warming on the atmosphere, cryosphere, biosphere

- 50 and ocean. The increase of air temperatures along the West Antarctic Peninsula coast (Carrasco, 2013) displays signs of a shifting climate system since the early 20th century (Thomas et al., 2009). Recently, rapid warming of both atmosphere and ocean is has causing caused ice shelf instability of ice shelves in West Antarctica, especially in some regions of the AP (Pritchard et al., 2012). Instability leading to The collapse of ice shelves triggersice shelf collapse has triggered an accelerated accelerated ice mass flow and discharge from land-based glaciers into the ocean, as the ice shelves' buttressing
- 55 function gets is lost. Several grounded tributary glaciers on AP and in West Antarctica recently loose mass to the oceans, and often at aAccelerated rates due to this phenomenonof ice mass loss (Pritchard and Vaughan, 2007; Rignot et al., 2005; Pritchard et al., 2012), which in combination with increased surface snow melt, has contributed to a negative surface mass balance especially in the northern part of the AP region (Harig and Simons, 2015; Seehaus et al., 2015; Dutrieux et al., 2014; Shepherd et al., 2012).
- 60 The glaciers of -the AP have lost ice mass at an estimated rate of approximately<del>round</del> 27 (±2) Gt y-1 between 2002 and 2014. This mass loss, which 
[revised manuscript text omitted]

- 140 overwintering crew at O'Higgins Station gather thecollected daily precipitation samples from pluviometers installed in a daily base directly fromat the meteorological observation siteory. Each daily sample comprised of a filling a narrow neck HDPE type bottle with a 30 ml composite sample of the precipitation (both liquid and solid) that fell in the previous 24 hours totalizing a daily composite sampling and extracting 30ml sample, that is saved in narrow neck type HDPE bottles. The bottles arewere tightly closed (tighten) and keptstored frozen year-long to ensure propercorrect storage and forto facilitate
- 145 the subsequent-later transported to the laboratory at the end of each year. From these samples, approximatelyround 6% (13 samples) were discarded from the analysis due to improper storage causing leakage from the bottlesing, likely due a leak of the bottles. Improper storage This-was assessed using a statistical outlier test (modified Thompson tau technique) which indicated by unusual values of stable water isotope analyses, and were discriminated using a statistical outlier test (modified Thompson tau technique).
- 150

[revised manuscript text omitted]

- 220 Fthe raw datasets obtained from stable water isotope analysis in firn cores (Section 3.1) produced low oscillation variance in the isotope–depth profiles profile. The–Whilst the measured isotope signals were considerably–noisy, but–the values do not fluctuate far from the each core's mean. This low variance, added to the fact that the There was a significant difference in the Standard Deviation (Sdev) values of oxygen isotopes between cores from lower altitudes (OH 4,  $\delta^{48}$ O Sdev = 1.2) versus cores from higher altitudes (OH 10,  $\delta^{48}$ O Sdev = 2.57) (Table 1). Furthermore, the patterns described in the isotope–depth
- 225 profiles do not correspond to seasonal cycles, means that dating each core using traditional annual layer counting is complicated. Despite several attempts to achieve a chronology by annual layer counting, the noise and the lack of consistent behavior in the isotope signals inhibited this interpretation. Difficulties from using conventional dating methodology for these firn cores led us to search for other ways to define the time scale of our signals.
- We firstly analysed the *dexcess* data, because *dexcess* is related to seasonal oceanic conditions, and therefore displays an annual signal in this region (Fernandoy et al., 2010). Whilst the stable water isotope results did not display a regular pattern, the *dexcess* is characterized by values displayed a noisy, low frequency signal oscillation., and so we used a Fast Fourier Transform (FFT) approach to filter the *dexcess* results to extract a seasonal signal. Low frequency data was dated by comparing the filtered, measured dataset with a filtered, theoretical dataset calculated based on atmospheric and oceanic conditions, specifically relative humidity and sea surface temperature, obtained from reanalysis data (GDAS). We use this low-
- 235 frequency periodic signal to date the core (see below).

We calculated a Ttheoretical  $d_{excess}$  values at our site was calculated using the relationship between rh and SST computed by Uemura et al. (2008):  $d_{excess\ meteo}$  = -0.42 \* rh + 0.45 \* SST + 37.9. The suitability of using this relationship for the Antarctic Peninsula region was assessed by comparing the  $d_{excess}$  measurements of the daily precipitation samples taken at the OH

- 240 station with the corresponding theoretical values. For each day that a precipitation sample was collected at OH, 3--days air parcel backward trajectories were calculated using the HYSPLIT model. We identified Ffrequent air parcel paths-were studied and calculated monthly mean values of *rh* and SST from re-analysis data (GDAS) along these paths-were calculated. We found a very good agreement between the measured and our theoretical  $d_{excessesses}$  values, with a correlation of R=0.86 (p<0.01)XXX.
- 245 Based on the previous significant correlation found between the measured and theoretical dexcess-values, This high correlation allows us to directly compare a direct comparison between a synthetic dexcess meteo time series and the observational dexcess record obtained from each firn core-could be made. For the method to be successful, the resultant depth-age model should maximize the common variability between the two time series.

- 260
- 265 At the same time, tThe dexcess signal obtained from stable isotope analysis of firn cores wasis measured represented into a signal with respect to the depth (i.e.: in the space domain). To extract the low frequency seasonal signal we first computed the Fast Fourrier Transform (FFT) of the dexcess data, which identifies all the frequencies in the record. The second lowest frequency was the one withselected from the FFT process, as it had the highest power The second lowest frequency peak was the onethat with the highest power. We therefore reconstructed the low-frequency dexcess signal by calculating the Inverse Fast Fourrier Transform (IFFT) from the lowest two identified frequency peaks. The dexcess isotope signal, was
- 270 Inverse Fast Fourier Transform (IFFT) non-the lowest two identified inequency peaks. The devees isotope signal, was filtered using the Inverse Fast Fourier Transform (IFFT) in order to transform the space domain (depth) into a frequency representation of the isotope signal. To determine the best frequency of this mode, we analyzed the variation between the original signal and the signal obtained from each step of the Fast Fourier Transform (FFT) decomposition. By studying the progression of these variations, the frequency of the second mode showed the highest frequency, where the Sdev reaches an equilibrium. Thus, the final signal is only defined by a set of low frequencies.
- The theoretical  $d_{excess}$  relationship was then used to calculate the behavior of  $d_{excess}$  for the length of the reanalysis record (2006 2014). We applied the same procedure to the monthly means of the synthetic  $d_{excess meteo}$  time series, thus obtaining two low-frequency signals that should show the same seasonal variability Thereafter, the same procedure was applied to time series constructed with monthly means of  $d_{excess meteo}$  described previously. The strong similarities between the two signals,
- 280 due to their dependency on the same variables (i.e.: environmental condition of the moisture source region). We then chose a linear depth—age model that visually matched the variability in the low—frequency observational dexcess data with the variability in the low-frequency synthetic dexcess meteo data. In this way, enabled the transformation from the depth domain of the dexcess signal (derived from measured stable isotope values) to time. This was made possible by using the common oscillation patterns in both profiles (by their IFFT), as time markers.- A single a-linear stretching factor was calculated using
- 285 that relationship and later applied to the wholecomplete firn cores datasets. The results of the dating procedure are shown in Figure 3. We used the same depth—age model to put the firn core δ18O records on a time axis for further analysis using monthly means. After following this procedure, dexcess can be represented by an age model as shown in Fig. 3. Subsequently, monthly means of the firn cores isotope signal were calculated to generate time series for further analysis. Once the dexcess signals from firn cores were represented as time series, the same was done for δ18O records, by considering the time depth
- 290 constraints defined (Fig. 3).

**4. Results**

**4.1 Precipitation samples**

[revised manuscript text omitted]

spring or SON (September October November), considering the October 2014 dataset to be representative of the SON behavior: -0.77 (standard error 2.08, p < 0.01) and -0.61 (standard error 2.88, p = 0.03), respectively. If only these two datasets (MAM and SON) are taken into account together, a  $\delta^{18}O-T$  relationship is defined by the new-linear regression-can

- 335 be defined as: δ18O= 0.79\* Tdaily-7.76 (R= 0.74, p<0.01, t-statistic test statistically significant at alpha 0.05). Thus, the time series of monthly averages shows a positive correlation between both parameters. An inverse behavior of s was identified during July 2008 and June 2014, compared to MAM and SON. Following the same procedure For austral Summer, the δ18O-T relationship for, December 2008 austral summer, or DJF (December January February), represented by December 2008, can be expressed as: δ18O= 1.17\* Tdaeily-8.19 (R=0.81, p=0.01);- was represented by the sample set of December 2008 and constrained by the sample set of December 2008.
- 340 austral winter or JJA (June-July-August) represented was represented by the sample set of June 2008as:, with and the sample set of June 2008as:, with and the sample set of June 2008as:, with and the sample of the sample set of June 2008as:, with and the set of the sample set of June 2008as:, with and the set of the sample set of June 2008as:, with and the set of the sample set of June 2008as:, with and the set of June 2008as:, with and set of June 2008as:, with and

**345 4.1.2 Deuterium excess – Temperature relationship**

350

370

Deuterium excess (dexcess) was calculated for each precipitation sample from stable water isotope data obtained at OH (see Table 2 for descriptive statistics). To evaluate the proximity to the original evaporation source for each precipitation sample, we examined both the isotope-temperature relationship as well as the near surface temperature and  $d_{excess}$  relationship using linear regression analysis. Together with isotope temperature relationship, the relationship of near surface temperature and the secondary parameter  $d_{excess}$ , was explored using linear trends to understand the proximity of the evaporation source. From

- stable water isotope information obtained from precipitation samples gathered at OH,  $d_{excess}$  values were calculated for each sample. Table 2 shows  $d_{excess}$  basic statistics for the dataset. DAs for the evaluation of the isotope-temperature relationship (Section 4.1.1), d-Values of  $d_{excess}$  lower than 9 ‰ (see Section 3.1) were filtered out in order to avoid disturbances in the model, as the quality of these samples may have been compromised during storage and transport. Daily  $d_{excess}$  values for
- 355 December 2008, March 2008 and 2009, June 2008 and October 2014the same months as specified in section 4.1.1 were compared with daily mean temperatures, similar as for 848O T. Chowever, correlation coefficients for these comparisons are were not significant (Table 3)R>-0.42 and R<0.09, p>0.1, for negative and positive correlation period, respectively) and show variability. However, for the 2008-09 datasets Bbythe dexcess-T correlation comparing for monthly temperature averages (calculated from daily events as outlined in Section 4.1.1) to the datasets of 2008 2009 and 2014, the dexcess T correlation was found to be significant (Table 3; Fig. 4b), and the associated defining a linear regression waas calculated to be: coefficient improved (Fig. 4b). For the 2008-2009 dataset, we obtained a correlation coefficient of R= -0.77 (p<0.01; T= -1 \* dexcess += -0.60 \* T\_monthly +2.12 (R=-0.77, p<0.01).53; t-statistic test statistically significant at alpha 0.05). For the 2014 dataset the correlation is not significant; defining a linear correlation: dexcess = 0.65 \* T\_monthly +0.94 (R=0.33, p>0.05)., we obtained an R=0.33 (p>0.05; T=0.17 \* dexcess 3.62).

**365 4.1.3 Moisture source of precipitation**

Three\_-day air parcel backward trajectories from precipitation events exhibit a wide distribution, probably explaining in part the variability of the isotope-temperature relationship presented in Sections 4.1.1 and 4.1.2. Most of the pathways originateing spatially in both-the Southern Pacific Ocean and the Amundsen-Bellingshausen Seas. The trajectories are mainly-primarily derived from the Bellingshausen Sea, the Bransfield Straight and the Drake Passage, Tierra del Fuego and South America's southern tip. MoreoverIn addition, some trajectories (<15%) originate from AP's eastern side. Precipitation trajectories show ann pattern almost elliptically distributed pattern pattern with a N40°W orientation, <del>where and</del> most follow pathways bounded by the latitudes between 60°S and 67°S. The correlation between monthly mean values of  $d_{excess}$  (from precipitation samples) and  $d_{excess\ meteo}$  (constructed from the meteorological parameters rh and SST of the high density precipitation pathways) had, showed a significant correlation coefficient of R= 0.86 (p<0.01; t-statistic-test statistically-significant at alpha 0.05) (Fig. 76), demonstrating that oceanic conditions control most of the precipitation

variability as the results of the previous sections pointed out.

**4.2 Firn core samples from the AP**

375

Table 1 shows the stable isotope results and basie\_descriptive statistics for firn cores retrieved at the northern AP. The co-isotopic relationship δD-δ18O of for each single firn core retrieved from LCL is analogous-related to the global meteoric
water line (GMWL) and the local meteoric water line (LMWL) (Rozanski et al., 1993), with a mean slope of s= 7.91 and an intercept of 3.64 (Fig. 78). These values are very close to those of the LMWL, although with a slightly higher intercept.

**4.2.1 Age model based on stable water isotopes**

Stable water isotope results from each firm cores allow to derive the derivation of individual depth profiles  $\partial f_0 \delta D$ ,  $\delta^{18}O$  and  $d_{\text{excess}}$  for each firm core. Lowest noise values and the clearest seasonal patterns were found in  $d_{\text{excess}}$  profiles ( $d_{\text{excess}}$  core) (Fig. 385 3), similar to findings published by Fernandoy et al. (2012). In Ssection 4.1.3 it was shown how that -Fernandoy et al. (2012). Due to the, and there was a high correlation between  $d_{excess}$  from precipitation samples and  $d_{excess meteo}$  at sea level; are significantly correlated, which-and therefore suggestings a nearby moisture source of the moisture-precipitating in this area (see section 4.1.3), the correlation between dexcess meteo and dexcess core was evaluated. The significant correlation points outindicates that, meant that moisture source the This was done to determine if the conditions of a coastal and oceanic-390 proximal moisture sourceorigin, that will be were \_also represented and preserved in the dexcess core record, and so and could to be used later as a chronology chronological marker. The  $d_{excess core}$  signals were first filtered for their high frequency oscillation patterns and then the remnant signals were compared with the high frequency filtered dexcess meteo monthly means (See Section 3.3). The intercomparison between dexcess meteo and dexcess core illustrates resulted in a close similarity between them. Main peak-valley fitting between both signals leads to a monthly mean  $d_{excess core}$  signal represented on a defined 395 depth-time scale (Fig. 8). The comparison between time series of monthly mean  $d_{excess core}$  and  $d_{excess meteo}$  data reveals correlation coefficients of R $\ge$ 0.67 (p<0.01;- degree of freedom (df) > 21 for all single cores, see table 4) t-statistic test statistically significant at alpha 0.05) for all firn cores analyzed and obtained from 2006 to 2015. Table 34 summarizes correlation coefficients, statistical significances and time intervals for each firn core. From the firn cores retrieved from LCL, single time series was constructed а 400 -and then compared to the  $d_{excess meteo}$  time series in order to analyze the isotopic signal for the whole time interval. A n even higher correlation coefficient of R= 0.75 (p<0.01; df= 81t-statistics test statistically significant at alpha 0.05) was obtained between the two signals ( $d_{excess meteo}$  and  $d_{excess core}$ ). For the overlapping time interval in OH-9 and OH-10 (February 2012 to January 2014), we only considered data from OH-9, as these samples consist of more recentfreshfresher snow layers with density between 350 and around approximately 410 kg m-3 (at core depth between 0 and 1 m), and firn with densities between aroundapproximately 410 and 530 kg m-3 (at core depth between 1 and 7 m) and less compacted firm than the corresponding 405 interval in OH-10. This in turn helps to avoid attenuation of the isotopic signal. Although we only considered OH-9 data for the overlapping time interval, we studied the changes in the standard deviation of the isotopic signal ( $\delta^{18}$ O and  $\delta$ D) from both firn cores in the common time span. The standard deviation shows a decrease of 16% after one year of deposition in core

410 signal decreases by 18%.

During visual firn core logging and density determination measurement, thin and scarce elevated density melt layers were identified. Melt layers were characterized by regular lateral extension and thickness 10 mm or higher. A 50 mm thick layer of around 50 mm of thickness was observed in the core OH-10 at depth 8.5m, showing which was the maximum observed

OH-10 with respect to the same time interval in OH-9; and for 2 to 3 years after the deposition, the standard deviation of the

thickness-observed. Melt add uprepresented a total of 5% of the accumulated water equivalent column of the cores OH-6,

- 415 OH-9 and OH-10. The melt layers do not show clear evidence of infiltration nor have a clear pattern of distribution with depth (i.e. an association with summer layers, as there are more or less homogenously distributed along the cores). Other firm high density structures, such as7 thinner crust like layers (<10 mm), do not correspond to melt events and are mostly related to wind scour processes and liquid precipitation-of liquid droplets (mean width of ~1 cm). The melt layers do not show evidence of infiltration nor have a clear pattern of distribution with depth (i.e. the an association with summer layers). Melt
- 420 and crust layers do not show a clear seasonal pattern with relation to While analyzing the melt layers in relation to their time equivalent with depth, no clear pattern associated with a season was noted. Around 70% of melt layers counted have a width <10 mm.

**4.2.2 Seasonal temperature reconstruction from stable water isotopes**

The age model developed using the  $d_{excess\ core}$  oscillation was later applied to construct a  $\delta^{18}$ O time series (Fig. 3). From this time series a periodical two2-year pattern was identified. This pattern is characterized by elevated values,a positive monthly mean  $\delta^{18}$ O standardized anomaly-higher than the  $\delta^{18}$ O values (z = observation - mean \* Std. dev.-1) monthly mean values between May and November in the years 2008 (z = 0.6), 2010, 2012 and 2014 (z > 1), Additionally, theywhich exhibit an inverse relationshipnegative correlation to temperatures at BE (z = -0.7 for 2010, and < -1 for 2008, 2012 and 2014). Between June and July in the following-years 2009, 2011 and 2013,  $\delta^{18}$ O values (z < -1) are lower than the mean and exhibit

- 430 a direct relationshippositive correlation to temperature at BE (z < -1) (Fig. 99). Therefore, the two-2-year periodical pattern mentioned above is represented by even numbered years with austral Wwinter  $\delta^{18}$ O values higher than the mean, followed by odd numbered years with austral winter  $\delta^{18}$ O values lower than the mean. Monthly mean  $\delta^{18}$ O values were transformed to their temperature equivalent using the  $\delta^{18}$ O-T relationship obtained in Ssection 4.1.1 from precipitation samples (Fig. 1010), in order to investigate their seasonal behavior.
- 435 Calendar seasons in these latitudes does not follow regular patterns (i.e. DJF, MAM, JJA, SON), as seasonality largely depends on the sea ice cover during winter, often extending beyond their-calendar limits. Large sea ice extent (SIE) leads to a delayed on-set of spring conditions. In this case winter-like conditions will be extended beyond August. Restricted sea ice extent on the contrary will lead to earlier Sspring-like conditions (before August). Depending on this-such conditions, we defined three seasons with their corresponding  $\delta^{18}O-T$  relationship. These seasons are: (1) an austral transitional season 440 which considers the months from March to May and October-November (MAM-ON) (using precipitation datasets from Table 3, March 2008–2009 and October 2014 for the  $\delta^{18}$ O–T relationship-during that season, s=0.69), (2) an austral Wwinter season which considers the months from June to September (JJAS) (using precipitation datasets from June 2008 for the  $\delta^{18}$ O-T relationship, Table 3-during that season, s=0.35), and (3) an austral S-summer season which considers months from December to February (DJF) (considers precipitation datasets from December 2008 for the  $\delta^{18}O-T$  relationship-during that season, s=1.17). All (compare of this section refers to the  $\delta^{18}$ O T relationship functions in section 4.1.1). Despite tThis 445 main-basic seasonal-classification of the seasons to for the use of describe the  $\delta^{18}O-T$  relationship does not explain all of the variability observed in the data, some particular seasons showed variable behavior when compared to the mean seasonal behavior in the time span covered in this study. In those cases, the seasonal behavior was extended or contracted beyond the boundaries of the main season classification depending on the SIE.

450

**4.2.3 Air temperature trends at Laclavere Plateau**

In the AP region, lapse rates change seasonally primarily due to variations in the presence and extent of sea ice cover which impacts thermal stability on the lower atmosphere. MTo reconstruct the near surface temperature from isotope data retrieved from firm cores and meteorological data from near stations, conditions of thermal stability of the lower atmosphere was 455 explored throughout different seasons. In the same way as for seasonality of the isotope temperature relationship, mean seasonal lapse rates obtained in this region show a clear seasonal dependency, with the highest rates during DJF (-5.31 °C km-1), alikesimilar values during MAM and SON (-4.43°C km-1, and -4.06°C km-1 respectively) and the lowest rates during JJA (-2.73° km-1) (Fig. 11).

Monthly near surface temperatures at LCL were estimated using Therefore,

- 460 Uua linear regression analysis based on sing the linear correlation between meteorological data: monthly lapse rates in BE, winter SAM index and SIE from OH (Fig. 142), The resultant equation is: a monthly temperature mean estimate that can be expressed by the equation  $T_{LCL} = (T_{BE} - 1.4) + 1.13$  ( $M_{month} * SIE_{OH} + N_{month}$ ), during the months when sea ice is developed (from May to September) and where Mmonth and NMonth represent the slope and intercept of the monthly lapse rate-SIE relationship, respectively. During the months when there is no sea ice (from October to April) the monthly temperature ean be expressed
- 465 by the equation is calculated from  $T_{LCL} = (T_{BE} - 1.4) + 1.13 * H(t)$ , where H(t) is the monthly mean lapse rate value of the month t measured in BE between 1978-1996. Considering theise variables, a mean annual air temperature of -7.5°C with a trend of  $-0.18^{\circ}$ C year-1 (statistically not significant at p=0.05) was estimated on-for LCL for the time period-2009-2014. On the other hand, Comparatively, considering if only the  $\delta^{18}O$  time series data and the isotope-T relationship are considered, a mean annual air temperature of -6.5°C with a trend of -0.33°C year-1 (statistically not significant at p=0.05) was is estimated on for
- 470 LCL for the years 2009-2014 same period. The comparison correlation between monthly mean temperature on 
[revised manuscript text omitted]

910

---

## Author Response (AR5)

**Answer to the Editor:**

Dear Dr. Smith, we certainly appreciate your input in this draft again and we are pleased that this draft is almost suitable for publication.
Regarding your last observations:

**Obs 1.**

My question is whether you looked at a FFT spectrum, and identified two peaks, each including a number of frequencies, or if you only used the two lowest-frequency bins in the FFT (i.e. the 1-cycle-per-record and the two-cycle-per-record bins). If the former, you should state the cutoff frequency (or wavelength) that you applied. If the latter, something is wrong, because any two signals reconstructed from only two Fourier components are likely to have good correlation.

**Response:**

In our case, we did the former. We looked at the FFT spectrum in order to identify the first two amplitude peaks. By following this procedure, we included a number of frequencies within the lowest range obtained from the FFT analysis. The cutoff frequencies used for each core were:

$3.14 < OH\text{-}4 <= 25.16$ $cm^{-1}$
$4.69 < OH\text{-}5 <= 37.56$ $cm^{-1}$
$4.81 < OH\text{-}6 <= 43.27$ $cm^{-1}$
$4.31 < OH\text{-}9 <= 43.10$ $cm^{-1}$
$5.26 < OH\text{-}10 <= 78.95$ $cm^{-1}$

This information is now included in Table 4 and the paragraph describing this in section 3.3 was modified to include this information.

**Obs 2.**

In either case, the number of degrees of freedom in the filtered signals cannot be larger than the number of Fourier components included in the spectrum input to the IFFT, so this is an important number to quote in the text.

**Response:**

To compute de IFFT the whole spectrum was integrated up to the second peak, so that the interval was covered by: 0<freq<=cut off freq. This means that the peak around 0 is not included and the second peak developed was effectively included (see Figure below). For example, in OH-9, the second frequency peak was at 43.1, so to build the new low-frequency signal, all frequencies/components were included until reaching the cutoff frequency of 43.1, without considering freq=0. This has been better described in the replacement paragraph, and with the addition of Table 4.

[Figure]

We now indicate for each range of frequencies used for the IDFT the number of frequencies contained in that range.

**Response to the editor:**

In addition, we proofread the document again and have corrected some minor mistakes in the text.

[revised manuscript text omitted]

---

## Author Response (AR6)

**Answer to the Editor:**

Dear Dr. Smith, in behalf of all authors we would like to thank your time an effort working on this manuscript. We really appreciate your patience and input improving the draft during this time. We are very happy to hear that the draft is now ready for publication.

Here, we did some minor modification to some of the figures to unify the term dexcess in Figures 3 and 4, and some other minor details in Figure 1 and 15.

[revised manuscript text omitted]